# SliderQuant: Accurate Post-Training Quantization for LLMs

**Shigeng Wang**[1,2]*, **Chao Li**[1]*, **Yangyuxuan Kang**[1], **Jiawei Fan**[1], **Zhonghong Ou**[2], **Anbang Yao**[1]†
[1]Intel Labs China,  [2]BUPT
{shigeng.wang,chao3.li,yangyuxuan.kang,jiawei.fan,anbang.yao}@intel.com

## Abstract

In this paper, we address post-training quantization (PTQ) for large language models (LLMs) from an overlooked perspective: given a pre-trained high-precision LLM, the predominant sequential quantization framework treats different layers equally, but this may be not optimal in challenging low bit-width settings. Based on three major categories of sequential PTQ methods, we empirically study the quantization impact of different layers on model accuracy, and observe that: (1) shallow/deep layers are usually more sensitive to quantization than intermediate layers; (2) among shallow/deep layers, the most sensitive one is the first/last layer, which exhibits significantly larger quantization error than others. These empirical observations imply that the quantization design for different layers of LLMs is required on multiple levels instead of a single level shared to all layers. Motivated by this, we propose a new PTQ framework termed **Slid**ing-lay**er Quant**ization (SliderQuant) that relies on a simple adaptive sliding quantization concept facilitated by few learnable parameters. The base component of SliderQuant is called inter-layer sliding quantization, which incorporates three types of novel sliding window designs tailored for addressing the varying quantization sensitivity of shallow, intermediate and deep layers. The other component is called intra-layer sliding quantization that leverages an incremental strategy to quantize each window. As a result, SliderQuant has a strong ability to reduce quantization errors across layers. Extensive experiments on basic language generation, zero-shot commonsense reasoning and challenging math and code tasks with various LLMs (including Llama/Llama2/Llama3/Qwen2.5 model families, DeepSeek-R1 distilled models and large MoE models) show that our method outperforms existing PTQ methods (including the latest PTQ methods using rotation transformations) for both weight-only quantization and weight-activation quantization under diverse bit width settings. Code is available at https://github.com/deep-optimization/SliderQuant.

## 1 Introduction

Transformer-based large language models (LLMs) (Vaswani et al., 2017; Brown et al., 2020; Achiam et al., 2023; Anil et al., 2023; Liu et al., 2024a; Jaech et al., 2024; Guo et al., 2025) have demonstrated extraordinary performance on a wide range of natural language processing tasks. However, deploying them in real-world scenarios poses a great challenge due to their huge model sizes. Post-training quantization (PTQ) is a practically appealing way to reduce memory and computation demands of LLMs at inference. It approximates pre-trained high-precision models with low-precision replacements conditioned on a small number of calibration samples, without the need of the expensive retraining pipeline used in quantization-aware training (Zafrir et al., 2019; Ma et al., 2024a; Xu et al., 2024). Because of this, PTQ research has gained increasing attention in the LLM community.

Existing PTQ methods for LLMs generally use a sequential quantization framework: splitting a pre-trained LLM into the same-sized disjoint parts, and then quantizing them from the first to the last part separately. Early seminal works, such as LLM.int8() (Dettmers et al., 2022), ZeroQuant (Yao et al., 2022), GPTQ (Frantar et al., 2023) and SmoothQuant (Xiao et al., 2023), assume that different layers of a pre-trained LLM are independent to each other, and use the layer-wise quantization.

---

*This work was done when Shigeng Wang was an intern at Intel Labs China. These authors contributed equally to the writing of the paper led by Anbang Yao who conceived the project. † Corresponding author.

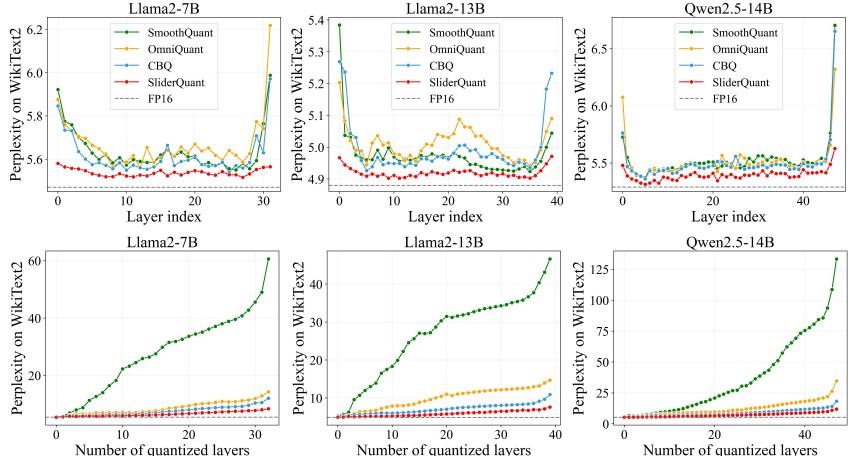

Figure 1: Illustrations on the quantization impact of different layers to model accuracy: (1) quantizing a single layer (the first row) and (2) quantizing the first $l$ layers (the second row) of Llama2-7B, Llama2-13B and Qwen2.5-14B. Here, we select 3 representative layer-wise, block-wise and multi-block-wise quantization methods, SmoothQuant, OmniQuant and CBQ, and examine them under 4-bit weight-activation (W4A4) quantization on WikiText2. *In the Appendix, Figure C provides more illustrations on Llama3-8B, Qwen2.5-7B and Qwen2.5-32B, showing similar observations.*

This simple framework has been popularly adopted by many subsequent works (Lin et al., 2024b; Ashkboos et al., 2024b; Lin et al., 2024a; Dettmers et al., 2024; Duanmu et al., 2024). Instead, OmniQuant (Shao et al., 2024) and FlatQuant (Sun et al., 2025) employ the block-wise quantization in which layers within each attention block are quantized simultaneously. To explore longer-distance dependencies than the block-wise quantization, QLLM (Liu et al., 2024b) and CBQ (Ding et al., 2025) utilize the multi-block-wise quantization based on a fixed-size sliding window, resembling the insights of prior works (Li et al., 2021; Zheng et al., 2022) tailored for quantizing convolutional neural networks in computer vision. These methods have significantly advanced the post-training quantization research for LLMs. However, in formulation, they typically treat different layers of a pre-trained LLM equally, no matter using the layer-wise or block-wise or multi-block-wise quantization. Such an assumption appears reasonable under moderate quantization settings, e.g., 8-bit weight quantization, as the total quantization error tends to be small. However, we conjecture it may be not optimal under challenging quantization settings, e.g., 4-bit weight-activation quantization.

For the sequential quantization framework described above, some previous works (Nagel et al., 2020; Li et al., 2021; Frantar et al., 2023; Shao et al., 2024; Ding et al., 2025) have established a solid theoretical foundation based on second-order Taylor expansion to get a better approximation solution for post-training quantization. This paper moves one step further: we revisit the predominant sequential quantization framework via questioning whether different layers of currently prevailing LLMs have similar quantization impacts on model accuracy. To explore this question, we select three representative layer-wise, block-wise and multi-block-wise quantization methods, SmoothQuant, OmniQuant and CBQ, and examine them on a lot of popular LLMs in the challenging 4-bit weight-activation quantization regime. Of particular interest, we have observed several properties. Firstly, for each of our tested LLMs, intermediate layers usually have smaller quantization impacts on model accuracy compared to shallow/deep layers. This implies that shallow/deep layers are more sensitive to quantization than intermediate layers which are relatively easy to quantize. Secondly, among shallow/deep layers, the first/last layer has the largest quantization impact on model accuracy, exhibiting significantly larger quantization error than others. This implies that the first and last layers are greatly important in the quantization process, as they are responsible for the very basic feature extraction and the final feature abstraction, respectively. Thirdly, as more layers are sequentially quantized, the quantization impact on model accuracy will be magnified gradually. However, SmoothQuant, OmniQuant and CBQ show unsatisfactory abilities to suppress this issue, suffering from the underlying premise that all layers are treated equally in their layer-wise, block-wise and multi-block-wise quantization frameworks. Figure 1 illustrates these properties on Llama2-7B, Llama2-13B (Touvron et al., 2023b) and Qwen2.5-14B (Yang et al., 2024).

These empirical properties highlight two ingredients that are essential to formulate an improved sequential quantization framework adopting a fixed bit-width: (1) special attention during quantization is required on shallow and deep layers, particularly the first and last layers; (2) the quantization synergy of successive layers is required to reduce quantization errors across layers. Driven by this, we

present a new sequential quantization framework, **Slid**ing-lay**er Quant**ization (SliderQuant) shown in Figure 2, which relies on a simple adaptive sliding quantization concept. In principle, by adopting a sliding window, the layers of a pre-trained LLM are sequentially divided into overlapping windows with the same size first, and then the quantization is performed window by window facilitated by few learnable quantization parameters. The overlaps of consecutive windows establish a basic quantization synergy path to reduce quantization errors across layers. However, simply using a fixed-size sliding window (Duanmu et al., 2024; Ding et al., 2025) still has a large gap to endow SliderQuant with the desired two ingredients, as shallow, intermediate and deep layers will be quantized with the same window size and moving interval per step and treated equally important. We fill this gap by presenting two novel sliding quantization components. Our base component, inter-layer sliding quantization, incorporates three types of sliding window designs tailored for adaptively quantizing shallow, intermediate and deep layers with a smart optimization relay across them. Specifically, it first allocates a progressively expanded sliding window along shallow layers, a fixed-size sliding window along intermediate layers and a progressively contracted sliding window along deep layers, and then performs the sliding quantization progressively. With these three types of sliding window designs, our inter-layer sliding quantization component can leverage the aforementioned empirical properties about the varying layer sensitivity to quantization. To exploit these empirical properties further, we present another complementary component, called intra-layer sliding quantization. It extends the progressively expanded sliding design within each window of inter-layer sliding quantization component, by which all layers in each window are jointly quantized in an incremental manner. Coupling these two components in this way forms a neat implementation of SliderQuant.

Overall, SliderQuant is a flexible PTQ framework which can be used for both weight-only and weight-activation quantization. Our experimental results show that SliderQuant outperforms existing PTQ methods across a broad range of quantization settings (W4A16, W3A16, W2A16, W4A4), model families (Llama, Llama2, Llama3, Qwen2.5, Qwen3) and model sizes (7B, 8B, 13B, 14B, 32B, 65B, 70B) on 2 basic language generation and 6 commonsense reasoning benchmarks. Incorporating advanced techniques (e.g., rotation transformations) into SliderQuant further improves its performance. In addition, we validate the effectiveness of our SliderQuant on the advanced MoE model Qwen3-30B-A3B. Notably, we also apply SliderQuant to the recently popular DeepSeek-R1 (Guo et al., 2025) distilled models with strong chain-of-thought reasoning abilities, achieving near-lossless accuracy under 4-bit weight-only quantization on 5 challenging math and code tasks.

## 2 RELATED WORK

Many PTQ works focus on weight-only quantization. Early works, such as Q-BERT (Shen et al., 2020) and GOBO (Zadeh et al., 2020), use a mixed-precision decomposition scheme in which the weight outliers (i.e., a small fraction of weights causing large quantization errors) are retained in high-precision format while the other weights are quantized into low-precision format. They consider small language models like BERT (Devlin et al., 2019). GPTQ (Frantar et al., 2023) formulates an efficient weight-only quantization approach using approximate second-order Hessian matrices. Compared to the popular round-to-nearest (RTN) quantization method (Dettmers et al., 2022), GPTQ shows much better performance when quantizing weights to 3-bit/4-bit. QuIP (Chee et al., 2024) introduces an incoherence-driven scheme to promote GPTQ, especially for the 2-bit quantization of weights, and an improved variant is further presented in Tseng et al. (2024). SpQR (Dettmers et al., 2024) and AWQ (Lin et al., 2024b) extend the mixed-precision decomposition scheme through designing more effective strategies to identify weight outliers, achieving superior performance to GPTQ. To avoid the inefficiency of the mixed-precision implementation on hardware systems (Cai et al., 2020), AWQ searches channel-wise factors to scale down weight outliers, enabling full-weight quantization. Other works, including but not limited to Shen et al. (2020), Tang et al. (2023), Park et al. (2024) and Kim et al. (2024), try to advance the weight-only PTQ research from other aspects.

Compared to weight-only quantization, weight-activation quantization can bring more significant reductions in both compute and storage costs at inference, but it is more challenging due to its higher sensitivity to quantization error. LLM.int8() (Dettmers et al., 2022), a pioneering work for weight-activation quantization, uses a mixed-precision scheme that quantizes all weights and most of activations into INT8 format, but isolates activation outliers (i.e., a small fraction of activations that have large magnitudes than the others) into FP16 format. To suppress activation outliers, the authors of Wei et al. (2022) employ the non-scaling layer normalization and the token-wise clipping, making activations to be more friendly for 8-bit quantization. Unlike LLM.int8() using the vanilla vector-

wise quantization with RTN, ZeroQuant (Yao et al., 2022) applies a fine-grained INT-8 quantization scheme consisting of group-wise quantization for weights and token-wise quantization for activations. Based on a linear equivalent transformation, SmoothQuant (Xiao et al., 2023) uses per-channel smoothing factors to scale down activation outliers and scale up the corresponding weights, mitigating the quantization difficulty from activations to weights which are easier to quantize. SmoothQuant calculates channel-wise smoothing factors over a randomly sampled calibration set in an offline manner, and is tailored for 8-bit weight-activation quantization. Instead, OmniQuant (Shao et al., 2024) dynamically learns activation-smoothing factors and weight-clipping thresholds, and considers more diverse bit-width settings down to 4-bit. QUIK (Ashkboos et al., 2024a) addresses 4-bit weight-activation quantization by extending the mixed-precision scheme. QLLM (Liu et al., 2024b) formulates a channel disassembly and channel assembly scheme facilitated by the low-rank adaptation (Hu et al., 2022) to suppress outliers in some channels. However, this scheme modifies LLM architectures, and thus introduces extra inference-time costs. QuaRot (Ashkboos et al., 2024b) uses random rotation transformations to remove outliers from the hidden state. Some subsequent works improve QuaRot by making rotation transformations learnable (Liu et al., 2025; Sun et al., 2025) or combining rotation and permutation transformations (Lin et al., 2024a). Similar to QLLM, these rotation-based methods also introduce extra computational costs at inference, as rotation transformations added to some layers are not absorbable due to non-linear operations.

## 3 METHOD

In this section, we describe the formulation of our SliderQuant and detail its implementation.

### 3.1 BASIC CONCEPT: FIXED-SIZE SLIDING QUANTIZATION

Given a pre-trained high-precision LLM having $L$ layers, we start with the vanilla sliding quantization (Duanmu et al., 2024; Ding et al., 2025). It uses a fixed-size sliding window $\{s, i\}$ moving along the layer direction of the given model and performs the sequential quantization in a window-wise manner, where $s$ denotes the window size and $i$ denotes the moving interval per step. The overlap between two consecutive windows is $s - i$. Let $\mathbf{W} = \{\mathbf{W}_1, ...\mathbf{W}_s\}$ be the pre-trained weight matrix set for $s$ layers in the current window and let $\mathbf{X}$ be its input feature corresponding to a small set of $c$ calibration samples. Then, for weight-only quantization, the optimization objective is defined as

$$\underset{\hat{\mathbf{W}}}{argmin} \, ||\mathcal{F}(\mathbf{W}, \mathbf{X}) - \mathcal{F}(\hat{\mathbf{W}}, \mathbf{X})||_2^2, \tag{1}$$

where $\mathcal{F}(\cdot, \cdot)$ denotes the output feature of the current window, and $\hat{\mathbf{W}} = \{\hat{\mathbf{W}}_1, ...\hat{\mathbf{W}}_s\}$ denotes the low-precision weight matrix set needs to be determined. For weight-activation quantization, the low-precision input feature $\hat{X}$ is obtained from the quantization of $X$ beforehand, and then its optimization objective can be defined by simply replacing $\mathcal{F}(\hat{\mathbf{W}}, \mathbf{X})$ in Eq. 1 by $\mathcal{F}(\hat{\mathbf{W}}, \hat{\mathbf{X}})$.

According to the above definition, when the window size $s$ is one layer or one attention block or multiple attention blocks and the moving interval per step $i$ is equal to the window size $s$ (i.e., there is no overlap between two consecutive windows), we will get layer-wise, block-wise and multi-block-wise quantization frameworks popularly used in existing PTQ works (Yao et al., 2022; Frantar et al., 2023; Xiao et al., 2023; Shao et al., 2024; Liu et al., 2024b). That is, they are special cases of fixed-sized sliding quantization. For sliding PTQ methods including ours, a larger window size leads to increased memory cost but enjoys much better accuracy (see Table 1) compared to layer-wise PTQ methods like GPTQ (the most efficient PTQ method tailored for weight-only quantization).

### 3.2 SLIDERQUANT

Recall that our empirical observations underlie two ingredients that are crucial to improve the sequential quantization framework. Firstly, special attention during quantization is required on shallow and deep layers, particularly the first and last layers, as they are more sensitive to quantization than intermediate layers. Secondly, the quantization synergy of successive layers is required to reduce quantization errors across layers. For fixed-size sliding quantization (*we use it as the baseline sliding quantization design in our ablations*), the existence of an overlap $s - i \geq 1$ between any two consecutive windows builds a basic synergy path to reduce quantization errors across layers. However, when using a fixed-size sliding window, all layers of a pre-trained LLM will be quantized with the same window size and moving interval per step. That is, shallow, intermediate and deep layers are still treated equally to a great extent, leading to a large gap to have the desired quantization

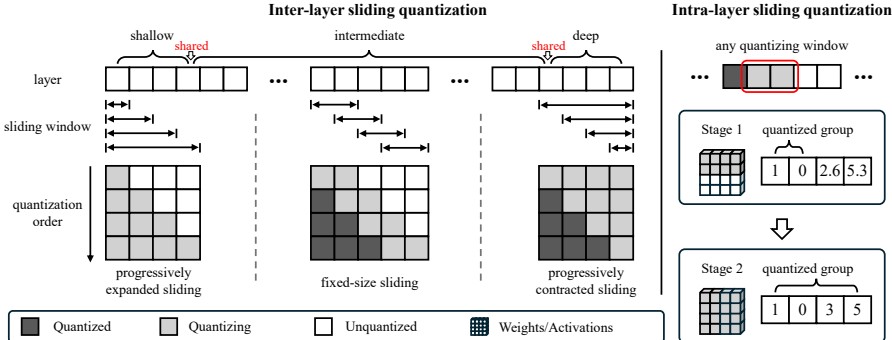

Figure 2: Overview of SliderQuant consisting of two components based on a simple adaptive sliding quantization concept. The base component of SliderQuant, inter-layer sliding quantization, has three sliding window designs along shallow, intermediate and deep layers of any pre-trained high-precision LLM, which are tailored for addressing their varying layer sensitivity to quantization. To establish a smooth sliding quantization relay from shallow to intermediate layers then from intermediate to deep layers, we set one overlapped layer between shallow and intermediate layers and one overlapped layer between intermediate and deep layers. Besides, this also makes each intermediate layer have an even quantization frequency. The other component of SliderQuant, intra-layer sliding quantization, is applied within the current window of inter-layer sliding quantization component, by which all layers in the current window are jointly quantized in an incremental manner.

design. We present SliderQuant to fill this gap via designing a more adaptive sliding quantization framework, which consists of two novel sliding quantization components, as shown in Figure 2.

**Inter-Layer Sliding Quantization.** Our base component, inter-layer sliding quantization, incorporates three types of sliding window designs tailored for adaptively quantizing shallow, intermediate and deep layers with a smart optimization relay across them. For $L_s$ shallow layers, a progressively expanded sliding window (PESW) is designed, which starts from quantizing the first layer with the window size of 1, then takes the first layer as the anchor layer and gradually increases the window size by 1 per step until including all shallow layers. With PESW, the first layer is always involved in the quantization process with every expanded sliding window, building dense local to global synergies to ease the quantization of shallow layers. Reversely, a progressively contracted sliding window (PCSW) is designed for $L_d$ deep layers, which starts from quantizing all deep layers, and then gradually decreases the window size by 1 per step until to only include the last layer. With PCSW, the last layer is always used as the anchor layer, and it is involved in the quantization process with every contracted sliding window, building dense global to local synergies to ease the quantization of deep layers. For $L_i$ intermediate layers, we adopt a fixed-size sliding window $\{s = 2, i = 1\}$ (FSSW) and an even optimization frequency across layers. Specifically, we set one overlapped layer between shallow and intermediate layers, and also set one overlapped layer between intermediate and deep layers. Alternatively, when the window size of fixed-size sliding quantization at intermediate layers is larger than 2 (i.e., $s > 2$), we can easily ensure an even optimization frequency by changing the number of overlapped layers between intermediate layers and shallow/deep layers, e.g., 2 overlapped layers when $s = 3$. Sequentially applying these three types of sliding window designs establishes the desired optimization relay across all layers of any pre-trained LLM, flexibly leveraging our identified empirical properties about the varying layer sensitivity to quantization. *According to the ablative experiments, we set $L_s = 4, L_d = 4$ as default for an accuracy-efficiency trade-off.*

**Intra-Layer Sliding Quantization.** To exploit our identified empirical properties further, another complementary component, called intra-layer sliding quantization, is presented. This component is applied into the current sliding window of our base component. Specifically, it extends the progressively expanded sliding design to each window of inter-layer sliding quantization, where its $s$ layers parallelly apply the progressively expanded sliding by a ratio $\gamma$ along weight and activation dimensions. As a result, the joint quantization of all $s$ layers is completed incrementally in $N = 1/\gamma$ sliding stages. *We set $\gamma = 0.5, N = 2$ as default, as illustrated in the right part of Figure 2.* In the first stage, the first half of weight/activation matrices in the current window of inter-layer sliding quantization is quantized jointly. In the second stage, the whole of weight/activation matrices (including the first half) in the current window of inter-layer sliding quantization is quantized jointly. As a result, intra-layer sliding quantization builds a local to global parameter synergy across layers within the current sliding window of inter-layer sliding quantization to reduce quantization error. Coupling these two components in this way also leads to a neat formulation of SliderQuant.

**Learnable Parameters and Quantizer.** SliderQuant builds a new sequential quantization framework by a flexible schedule of multiple sliding window designs, enabling dense synergies among shallow/intermediate/deep layers and a smart quantization relay across them. Next, we describe how SliderQuant quantizes weights and activations at each layer in a sliding window. It is well known that effectively removing outliers in weights and activations at each layer of a pre-trained LLM is important to reduce the total quantization error. Prior works (Xiao et al., 2023; Shao et al., 2024; Liu et al., 2024b; Ding et al., 2025) popularly use channel scaling (CS) (Meller et al., 2019) and low-rank adaptation (LoRA) (Hu et al., 2022; Dettmers et al., 2023) to handle this issue. Inspired by them, we simply combine CS and LoRA in our method. Let $\mathbf{W}_i \in \mathbb{R}^{n \times m}$ be the weight matrix of the $i^{th}$ layer in a sliding window, and let $\mathbf{X}_i \in \mathbb{R}^{k \times n}$ be its input feature corresponding to a small set of $c$ calibration samples ($c = 128$ as default). Then, the quantization process is defined as

$$\tilde{\mathbf{X}}_i = \mathbf{X}_i \oslash \alpha_i, \quad \tilde{\mathbf{W}}_i = \mathbf{W}_i \odot \alpha_i + \mathbf{A}_i \mathbf{B}_i, \quad \mathbf{X}_{i+1} = \text{quantizer}(\tilde{\mathbf{X}}_i) \cdot \text{quantizer}(\tilde{\mathbf{W}}_i), \quad (2)$$

where $\alpha_i \in \mathbb{R}^n$ denotes a learnable channel-wise scaling vector to scale $\mathbf{X}_i$ and reversely scale $\mathbf{W}_i$, $\mathbf{A}_i \in \mathbb{R}^{n \times r}$ and $\mathbf{B}_i \in \mathbb{R}^{r \times m}$ ( *we set $r = 4$ in experiments*) are two low-rank matrices to get a refined weight matrix $\tilde{\mathbf{W}}_i \in \mathbb{R}^{n \times m}$ for quantization, $\oslash$ and $\odot$ denote element-wise division and multiplication operations, respectively. With the refined weight matrices $\tilde{W}$ and input $\tilde{X}$ defined by Eq. 2, the corresponding quantized weight matrices $\hat{W} = \text{quantizer}(\tilde{W})$ and input $\hat{X} = \text{quantizer}(\tilde{X})$ are obtained to minimize the mean square error defined in Eq. 1. In implementation, we use a uniform quantizer for both weights and activations, for simplicity and fair performance comparisons with existing post-training quantization methods. *We put its definition in the Appendix B.1.*

## 4 EXPERIMENTS

**Models.** We first select widely used Llama (Touvron et al., 2023a), Llama2 (Touvron et al., 2023b), Llama3 (Dubey et al., 2024) and Qwen2.5 (Yang et al., 2024) families for experiments. To further explore the potential of SliderQuant, we evaluate it on more advanced LLMs, including a Mixture of Experts (MoE) model Qwen3-30B-A3B (Yang et al., 2025) and the recently popular DeepSeek-R1 (Guo et al., 2025) distilled models with chain-of-thought reasoning abilities.

**Evaluations.** For most models (including the Llama series, Qwen2.5, and Qwen3), following the commonly adopted settings in post-training quantization research, we evaluate their two fundamental capabilities: basic language generation and commonsense reasoning. For language generation, we report perplexity on WikiText2 (Merity et al., 2017) and C4 (Raffel et al., 2020). For commonsense reasoning, we report zero-shot accuracy on 6 widely adopted benchmarks: PIQA (Bisk et al., 2020), ARC (Clark et al., 2018), HellaSwag (HS) (Zellers et al., 2019), Winogrande (WG) (Sakaguchi et al., 2021), BoolQ (Clark et al., 2019), and MMLU (Hendrycks et al., 2020). For the DeepSeek-R1 distilled models with chain-of-thought reasoning abilities, we evaluate their performance on challenging mathematical reasoning (MATH-500 (Hendrycks et al., 2021), AIME-2024 (Jia, 2024), and GSM8K (Cobbe et al., 2021)) and code generation benchmarks (HumanEval+ and MBPP+ (Liu et al., 2023)). To ensure fair and reproducible results, we adopt standard evaluations: LM Evaluation Harness (Gao et al., 2024) for commonsense reasoning, OpenCompass (Contributors, 2023) for mathematical reasoning, and EvalPlus (Liu et al., 2023) for code generation.

**Counterpart Methods.** Recent top-performing PTQ methods usually introduce architectural modifications, some of which are not absorable at inference. For a fair comparison, we categorize existing PTQ methods into two distinct groups based on whether they introduce extra inference-time costs. Notably, as a flexible quantization framework, SliderQuant maintains compatibility with both paradigms. For quantization without extra inference-time costs, our experiments cover both weight-only and weight-activation quantization: for weight-only quantization, we compare SliderQuant with round-to-nearest quantization (RTN), GPTQ (Frantar et al., 2023), AWQ (Lin et al., 2024b), and QuIP (Chee et al., 2024); for weight-activation quantization, we compare SliderQuant with RTN, SmoothQuant (Xiao et al., 2023), OmniQuant (Shao et al., 2024), AffineQuant (Ma et al., 2024b), and CBQ (Ding et al., 2025). For quantization with extra inference-time costs, we implement a variant of SliderQuant named SliderQuant+ for a fair comparison, which additionally incorporates rotation transformations. We compare it with QLLM (Liu et al., 2024b), Atom (Zhao et al., 2024), DuQuant (Lin et al., 2024a), QuaRot (Ashkboos et al., 2024b), SpinQuant (Liu et al., 2025), and FlatQuant (Sun et al., 2025). *Implementation details are provided in the Appendix B.*

Table 1: Results comparison of different quantization methods without extra inference-time costs on the language generation tasks. The metric is perplexity. Best results are shown in bold.

| #Bits | Method | Llama2-7B | | Llama2-13B | | Llama2-70B | | Llama3-8B | | Qwen2.5-7B | | Qwen2.5-14B | |
|---|---|---|---|---|---|---|---|---|---|---|---|---|---|
| | | Wiki ↓ | C4 ↓ | Wiki ↓ | C4 ↓ | Wiki ↓ | C4 ↓ | Wiki ↓ | C4 ↓ | Wiki ↓ | C4 ↓ | Wiki ↓ | C4 ↓ |
| W16A16 | - | 5.47 | 6.97 | 4.88 | 6.46 | 3.33 | 5.54 | 6.13 | 8.93 | 7.73 | 11.55 | 5.30 | 9.11 |
| W4A16 | RTN | 6.11 | 7.71 | 5.20 | 6.83 | 3.67 | 5.79 | 8.29 | 11.85 | 10.39 | 14.83 | 6.78 | 10.35 |
| | AWQ | 6.15 | 7.68 | 5.12 | 6.74 | 3.60 | 5.70 | 8.09 | 11.23 | 8.54 | 12.78 | 6.43 | 9.89 |
| | GPTQ | 5.83 | 7.37 | 5.13 | 6.70 | 3.58 | 5.67 | 8.01 | 11.34 | 8.64 | 12.98 | 6.45 | 10.01 |
| | OmniQuant | 5.74 | 7.35 | 5.02 | 6.65 | 3.47 | 5.65 | 7.28 | 10.59 | 8.23 | 12.25 | 5.94 | 9.67 |
| | CBQ | 5.67 | 7.23 | 5.02 | 6.67 | 3.46 | 5.64 | 6.93 | 10.27 | 7.92 | 11.77 | 5.83 | 9.54 |
| | SliderQuant | **5.61** | **7.19** | **5.00** | **6.54** | **3.41** | **5.60** | **6.79** | **9.94** | **7.81** | **11.59** | **5.80** | **9.53** |
| W2A16 | RTN | 3.8e4 | 4.8e4 | 5.6e4 | 7.2e4 | 2.0e4 | 2.4e4 | 2.4e6 | 2.5e6 | 6.9e4 | 6.9e4 | 6.0e6 | 4.4e6 |
| | AWQ | 2.2e5 | 1.7e5 | 1.2e5 | 9.4e4 | 9.1e1 | 5.1e1 | 5.6e5 | 3.1e5 | 1.5e2 | 2.7e2 | 1.4e3 | 2.7e3 |
| | GPTQ | 7.7e3 | NAN | 2.1e3 | 3.2e2 | 77.95 | 48.82 | 7.8e5 | 9.7e5 | 1.2e2 | 3.1e2 | 1.2e3 | 1.3e3 |
| | QuIP | 55.00 | - | 13.75 | - | 6.96 | - | 1.2e3 | - | - | - | - | - |
| | OmniQuant | 37.37 | 90.64 | 17.21 | 26.76 | 7.81 | 12.28 | 2.8e5 | 3.9e5 | 56.45 | 89.13 | 67.84 | 89.56 |
| | CBQ | 12.10 | 18.91 | 9.32 | 21.93 | 7.23 | 11.34 | 91.83 | 404.31 | 18.65 | 37.10 | 13.65 | 25.55 |
| | SliderQuant | **9.59** | **13.83** | **7.71** | **11.21** | **6.53** | **9.59** | **27.59** | **56.98** | **17.15** | **31.08** | **12.68** | **21.91** |
| W4A4 | RTN | 5.3e2 | 5.4e2 | 5.8e2 | 5.3e2 | 8.9e4 | 9.9e4 | 2.3e2 | 2.0e2 | 3.6e5 | 3.7e5 | 4.0e3 | 3.0e3 |
| | SmoothQuant | 83.12 | 77.27 | 46.62 | 43.19 | 33.40 | 43.28 | 2.0e2 | 1.5e2 | 1.3e2 | 2.9e2 | 1.3e2 | 1.4e2 |
| | OmniQuant | 14.26 | 18.02 | 12.30 | 14.55 | 11.54 | 13.72 | 1.5e2 | 1.4e2 | 93.73 | 2.9e2 | 34.70 | 61.75 |
| | AffineQuant | 12.69 | 15.76 | 11.45 | 13.97 | - | - | 2.1e3 | 3.5e3 | - | - | - | - |
| | CBQ | 12.73 | 14.45 | 8.48 | 11.71 | 7.56 | 11.04 | 35.97 | 32.64 | 35.00 | 72.09 | 18.20 | 27.96 |
| | SliderQuant | **8.34** | **11.10** | **7.62** | **10.26** | **6.87** | **9.67** | **15.47** | **21.74** | **13.81** | **21.52** | **11.00** | **16.60** |

Table 2: Results comparison of different quantization methods without extra inference-time costs on the zero-shot commonsense reasoning tasks. The metric is accuracy (%).

| Model | #Bits | Method | PIQA ↑ | ARC-e ↑ | ARC-c ↑ | HS ↑ | WG ↑ | BoolQ ↑ | MMLU ↑ | Avg ↑ |
|---|---|---|---|---|---|---|---|---|---|---|
| Llama2-13B | W16A16 | - | 80.41 | 77.40 | 49.15 | 79.37 | 72.14 | 80.55 | 52.77 | 70.26 |
| | W4A4 | SmoothQuant | 61.10 | 44.87 | 27.47 | 41.03 | 50.67 | 58.50 | 21.14 | 43.54 |
| | W4A4 | OmniQuant | 69.21 | 57.37 | 34.56 | 61.95 | 56.91 | 65.44 | 23.56 | 52.71 |
| | W4A4 | CBQ | 71.00 | 61.57 | 35.84 | 65.15 | 57.93 | 66.39 | 24.78 | 54.67 |
| | W4A4 | SliderQuant | **71.65** | **62.88** | **37.80** | **66.02** | **60.77** | **71.22** | **27.04** | **56.77** |
| Qwen2.5-14B | W16A16 | - | 82.10 | 79.59 | 58.87 | 82.95 | 75.61 | 85.26 | 77.58 | 77.42 |
| | W4A4 | SmoothQuant | 54.57 | 35.14 | 24.66 | 35.29 | 51.46 | 56.45 | 24.90 | 40.35 |
| | W4A4 | OmniQuant | 59.45 | 48.56 | 30.16 | 61.42 | 54.23 | 58.34 | 27.68 | 48.55 |
| | W4A4 | CBQ | 67.52 | 60.86 | 36.18 | 60.12 | 58.09 | 60.15 | 31.61 | 53.50 |
| | W4A4 | SliderQuant | **71.16** | **66.96** | **40.96** | **63.38** | **62.51** | **64.25** | **43.50** | **58.96** |

## 4.1 MAIN RESULTS

**Quantization without Extra Inference-Time Costs.** We first report the results for quantization methods without extra inference-time costs. As shown in Table 1, SliderQuant consistently achieves lower perplexity on WikiText2 and C4 than counterpart PTQ methods across a broad range of quantization settings, model families and model sizes. Under the extremely low-bit configuration of W4A4, SliderQuant gets more prominent performance. These results demonstrate the robustness and effectiveness of SliderQuant in preserving generation quality even under aggressive quantization. In Table 2, we provide the results comparison on 6 commonsense QA benchmarks. We can observe that SliderQuant consistently surpasses counterpart PTQ methods. These results validate the effectiveness of SliderQuant in preserving various capabilities of LLMs under low-bit quantization.

**Quantization with Extra Inference-Time Costs.** The comparative results of quantization methods with extra inference-time costs are shown in Table 3. We can see that SliderQuant+ achieves the best results on average across different models and benchmarks. While SliderQuant demonstrates strong quantization capabilities, incorporating rotation transformations further enhances its effectiveness, enabling it to handle more precision-sensitive scenarios. Combining the results in Table 1, Table 2 and Table 3, we demonstrate the generalizability of our sliding-layer quantization framework, which consistently gets superior performance compared to the state-of-the-art PTQ methods with or without extra inference-time costs. *Experimental details are provided in the Appendix C.*

**Extension to Mixture of Experts Architectures.** As shown in Table 4, when applying SliderQuant to Qwen3-30B-A3B, an advanced MoE architecture, we can observe a similar performance improvement trend as on dense LLMs. The results further validate the generalizability of SliderQuant.

**Challenging Math and Code Tasks with Reasoning Language Models.** Most existing PTQ works are limited to evaluating LLMs on basic language generation and commonsense reasoning tasks. However, for real-world applications, it is also crucial to assess their complex reasoning abilities on more challenging tasks. In the experiments, we apply SliderQuant to the state-of-the-art DeepSeek-

Table 3: Results comparison of different quantization methods with extra inference-time costs. SliderQuant+ denotes SliderQuant using rotation transformations.

| Model | #Bits | Methods | WikiText2 ↓ | C4 ↓ | PIQA ↑ | ARC-e ↑ | ARC-c ↑ | HS ↑ | WG ↑ | Avg ↑ |
|---|---|---|---|---|---|---|---|---|---|---|
| Llama2-7B | W16A16 | - | 5.47 | 6.97 | 78.84 | 74.62 | 46.42 | 75.90 | 69.46 | 69.05 |
| | W4A4 | QLLM | 11.75 | 13.26 | 67.68 | 44.40 | 30.89 | 58.45 | 56.59 | 51.60 |
| | W4A4 | Atom | 6.96 | 9.12 | 69.75 | 47.35 | 34.22 | 63.21 | 56.51 | 54.21 |
| | W4A4 | DuQuant | 6.08 | 7.79 | 75.68 | 50.00 | 37.46 | 69.74 | 63.93 | 59.36 |
| | W4A4 | QuaRot | 6.10 | 8.69 | 76.77 | 69.87 | 40.87 | 72.16 | 63.77 | 64.69 |
| | W4A4 | SpinQuant | 5.96 | 8.28 | 76.17 | 69.28 | 41.72 | 72.90 | 66.06 | 65.23 |
| | W4A4 | FlatQuant | 5.79 | 7.79 | 77.26 | 72.05 | 43.26 | 73.64 | 69.53 | 67.15 |
| | W4A4 | SliderQuant+ | **5.71** | **7.68** | **77.97** | **73.15** | **43.35** | **73.71** | **69.74** | **67.58** |
| Llama2-13B | W16A16 | - | 4.88 | 5.46 | 80.41 | 77.40 | 49.15 | 79.37 | 72.14 | 71.69 |
| | W4A4 | QLLM | 9.09 | 11.13 | 70.46 | 48.48 | 34.39 | 62.80 | 55.41 | 54.31 |
| | W4A4 | Atom | 6.96 | 9.12 | 71.16 | 50.89 | 37.88 | 67.51 | 58.40 | 57.17 |
| | W4A4 | DuQuant | 5.33 | **7.02** | 77.26 | 56.23 | 42.15 | 73.68 | 65.43 | 62.95 |
| | W4A4 | QuaRot | 6.10 | 8.67 | 77.69 | 69.95 | 42.83 | 73.54 | 67.88 | 66.38 |
| | W4A4 | SpinQuant | 5.44 | 8.11 | 78.40 | 72.43 | 43.69 | 75.52 | 68.90 | 67.79 |
| | W4A4 | FlatQuant | 5.12 | 7.09 | 79.38 | 76.64 | 48.04 | 77.59 | 70.24 | 70.38 |
| | W4A4 | SliderQuant+ | **5.07** | 7.04 | **79.96** | **77.27** | **48.95** | **77.96** | **71.98** | **71.22** |
| Llama-2-70B | W16A16 | - | 3.32 | 5.71 | 82.70 | 81.02 | 57.17 | 83.81 | 77.98 | 76.54 |
| | W4A4 | QuaRot | 3.79 | 6.12 | 81.83 | 79.76 | 55.46 | 81.58 | 76.09 | 74.94 |
| | W4A4 | SpinQuant | 3.70 | 6.07 | 82.37 | 79.04 | 55.38 | 82.57 | 78.22 | 75.52 |
| | W4A4 | FlatQuant | 3.55 | 5.91 | 82.75 | 80.30 | 56.14 | 83.01 | 77.90 | 76.02 |
| | W4A4 | SliderQuant+ | **3.50** | **5.87** | **82.75** | **81.23** | **56.57** | **83.12** | **77.93** | **76.32** |
| Llama3-8B | W16A16 | - | 6.13 | 8.93 | 80.79 | 77.69 | 53.41 | 79.13 | 72.77 | 72.76 |
| | W4A4 | Atom | 22.14 | 31.83 | 62.95 | 49.45 | 30.12 | 53.75 | 56.04 | 50.46 |
| | W4A4 | DuQuant | 8.06 | 11.29 | 76.22 | 70.41 | 43.69 | 73.87 | 67.80 | 66.40 |
| | W4A4 | QuaRot | 8.16 | 13.38 | 75.14 | 68.01 | 43.34 | 72.94 | 65.82 | 65.05 |
| | W4A4 | SpinQuant | 7.39 | 12.19 | 77.37 | 74.20 | 47.27 | 74.55 | 68.51 | 68.38 |
| | W4A4 | FlatQuant | 6.98 | 11.13 | 79.16 | 75.80 | 50.00 | 76.80 | 72.69 | 70.89 |
| | W4A4 | SliderQuant+ | **6.87** | **11.04** | **79.22** | **77.53** | **50.60** | **77.31** | **72.82** | **71.50** |
| Qwen2.5-7B-Instruct | W16A16 | - | 8.36 | 14.37 | 80.20 | 75.80 | 51.37 | 79.57 | 69.93 | 71.37 |
| | W4A4 | FlatQuant | 8.46 | 13.94 | 76.93 | 77.69 | 51.71 | 78.42 | 69.53 | 70.86 |
| | W4A4 | SliderQuant+ | **8.00** | **13.38** | **79.56** | **79.05** | **52.27** | **78.66** | **69.88** | **71.88** |

Table 4: Exploration of applying SliderQuant to the Mixture of Experts (MoE) model Qwen3-30B-A3B. We take OmniQuant as the baseline using its public code for a comparative evaluation.

| #Bits | Methods | WikiText2 ↓ | C4 ↓ | PIQA ↑ | ARC-e ↑ | ARC-c ↑ | HS ↑ | WG ↑ | BoolQ ↑ | MMLU ↑ | Avg ↑ |
|---|---|---|---|---|---|---|---|---|---|---|---|
| W16A16 | - | 8.71 | 12.08 | 80.14 | 79.25 | 56.23 | 77.66 | 69.93 | 88.56 | 77.74 | 75.64 |
| W4A16 | OmniQuant | 9.25 | 12.54 | 79.43 | 77.23 | 52.22 | 76.36 | 69.46 | 87.68 | 76.21 | 74.08 |
| W4A16 | SliderQuant | **9.04** | **12.47** | **79.87** | **77.44** | **55.20** | **76.75** | **70.96** | **87.9**2 | **76.83** | **75.00** |
| W3A16 | OmniQuant | 10.27 | 13.52 | 79.00 | 76.73 | 52.73 | 74.77 | 67.25 | 86.94 | 73.63 | 73.01 |
| W3A16 | SliderQuant | **9.92** | **13.32** | **80.20** | **79.00** | **54.61** | **74.80** | **70.48** | **87.61** | **74.63** | **74.48** |
| W2A16 | OmniQuant | 33.25 | 27.12 | 70.78 | 54.97 | 34.04 | 56.59 | 54.46 | 69.63 | 32.67 | 53.31 |
| W2A16 | SliderQuant | **23.84** | **21.60** | **72.36** | **67.51** | **40.53** | **60.79** | **62.04** | **77.61** | **47.19** | **61.15** |
| W4A4 | OmniQuant | 52.43 | 41.98 | 67.41 | 55.64 | 33.28 | 52.34 | 52.09 | 64.86 | 25.17 | 50.11 |
| W4A4 | SliderQuant | **15.49** | **17.77** | **75.52** | **70.79** | **46.67** | **68.82** | **63.14** | **82.26** | **60.80** | **66.86** |

Table 5: Exploration of applying SliderQuant to the DeepSeek-R1 distilled models with chain-of-thought reasoning abilities on the challenging mathematical reasoning and code generation tasks.

| Model | #Bits | Method | Mathematical Reasoning(pass@1) | | | Code Generation(pass@1) | | Avg ↑ |
|---|---|---|---|---|---|---|---|---|
| | | | MATH-500 ↑ | AIME-2024 ↑ | GSM8K ↑ | HumanEval+ ↑ | MBPP+ ↑ | |
| DeepSeek-R1-Distill-Qwen-14B | W16A16 | - | 95.00 | 73.33 | 91.50 | 73.17 | 61.11 | 78.82 |
| | W2A16 | OmniQuant | 0.00 | 0.00 | 2.20 | 0.00 | 0.00 | 0.44 |
| | W2A16 | SliderQuant | **29.40** | **10.00** | **54.28** | **12.80** | **21.16** | **25.53** |
| DeepSeek-R1-Distill-Qwen-14B | W4A16 | OmniQuant | 91.60 | 50.00 | 90.29 | 70.12 | 55.03 | 71.41 |
| | W4A16 | SliderQuant | **94.60** | **70.00** | **91.35** | **72.56** | **60.32** | **77.77** |
| DeepSeek-R1-Distill-Qwen-32B | W16A16 | - | 94.60 | 76.67 | 93.02 | 81.71 | 69.84 | 83.17 |
| | W2A16 | OmniQuant | 13.40 | 0.00 | 26.83 | 0.00 | 0.00 | 8.05 |
| | W2A16 | SliderQuant | **58.60** | **16.67** | **73.69** | **12.80** | **21.16** | **36.59** |
| DeepSeek-R1-Distill-Qwen-32B | W4A16 | OmniQuant | 93.00 | 56.66 | 92.64 | 75.00 | 65.61 | 76.58 |
| | W4A16 | SliderQuant | **94.40** | **76.67** | **92.94** | **80.49** | **69.05** | **82.71** |

R1 distilled models, exploring its performance on challenging mathematical reasoning and code generation benchmarks. The results are shown in Table 5. Under the W4A16 setting, SliderQuant remains near-lossless relative to FP16 on both 14B and 32B models, while consistently outperforming OmniQuant. Under the more aggressive W2A16 setting, SliderQuant achieves substantially higher accuracy than OmniQuant across all benchmarks and model scales. These results indicate that SliderQuant preserves reasoning fidelity at 4-bit setting and remains robust even at 2-bit setting.

Table 6: Results comparison of SliderQuant with mixed-precision quantization methods on Llama2-13B. In the table, (1) for LLM-MQ/QUIK, 0.5% in FP16 or 5% in FP16 indicates the ratio of weight or weight-activation outliers at each layer stored in FP16; (2) for 2.1-bit weight quantization by SliderQuant, we simply use 4-bit quantization to the first and last layers while quantizing all other layers to 2-bit, and SliderQuant+ denotes our best version of SliderQuant using rotation transformations.

| #Bits | Method | WikiText2 ↓ | PIQA ↑ | ARC-e ↑ | HS ↑ | WG ↑ | Avg ↑ |
|---|---|---|---|---|---|---|---|
| W16A16 | - | 4.88 | 80.41 | 77.40 | 79.37 | 72.14 | 77.33 |
| W4A16 (0.5% in FP16) | LLM-MQ | 8.03 | 79.49 | 58.50 | 76.31 | 69.30 | 70.90 |
| W4A16 | SliderQuant | **5.00** | **80.41** | **76.73** | **78.30** | **72.14** | **76.90** |
| W3.4A16 (0.5% in FP16) | LLM-MQ | 8.61 | 79.49 | 58.12 | 74.77 | 69.61 | 70.50 |
| W3A16 | SliderQuant | **5.27** | **79.11** | **74.16** | **76.78** | **69.76** | **74.95** |
| W2.2A16 (0.5% in FP16) | LLM-MQ | 10.80 | 76.77 | 55.26 | 70.83 | 67.09 | 67.49 |
| W2.1A16 | SliderQuant | **7.64** | **77.21** | **62.58** | **71.05** | **67.51** | **69.59** |
| W2A16 (0.5% in FP16) | LLM-MQ | 12.17 | 75.84 | 54.29 | 68.32 | 65.51 | 65.99 |
| W2A16 | SliderQuant | 7.71 | 73.56 | 67.47 | 64.75 | 63.22 | 67.25 |
| W2A16 (0.5% in FP16) | SliderQuant | **7.64** | **76.13** | **67.51** | **69.25** | **66.12** | **69.75** |
| W4.9A4 (5% in FP16) | QUIK | 5.28 | 79.22 | 74.92 | **78.36** | 71.90 | 76.10 |
| W4A4 | SliderQuant+ | **5.07** | **79.96** | **77.27** | 77.96 | **71.98** | **76.79** |

Table 7: Results comparison of SliderQuant with the mixed-precision quantization method SpQR on Llama-7B and Llama-13B.

| Model | #Bits | Method | WikiText2 ↓ | C4 ↓ | PIQA ↑ | ARC-e ↑ | ARC-c ↑ | HS ↑ | WG ↑ | Avg ↑ |
|---|---|---|---|---|---|---|---|---|---|---|
| Llama-7B | W16A16 | - | 5.68 | 7.08 | 79.43 | 73.15 | 45.05 | 76.16 | 70.24 | 68.81 |
| | W3.45A16 | SpQR | 5.87 | 7.28 | 78.13 | 65.87 | 38.05 | 55.27 | 67.48 | 60.96 |
| | W3A16 | SliderQuant | **5.82** | **7.13** | 77.42 | 69.70 | 40.36 | 71.74 | 67.96 | 65.44 |
| Llama-13B | W16A16 | - | 5.09 | 5.62 | 80.41 | 74.71 | 47.95 | 79.08 | 73.09 | 71.05 |
| | W3.45A16 | SpQR | 5.22 | 6.72 | 78.73 | 73.27 | 42.75 | 58.22 | 68.90 | 64.37 |
| | W3A16 | SliderQuant | **5.21** | **6.78** | 79.33 | 73.15 | 45.73 | 76.19 | 70.72 | 69.02 |

**Comparison with Mixed-Precision Methods.** In Table 6 and Table 7, we compare SliderQuant with the state-of-the-art mixed-precision quantization methods including LLM-MQ (Li et al., 2023) and SpQR (Dettmers et al., 2024) for weight-only quantization, and QUIK for weight-activation quantization. Across all settings, SliderQuant consistently achieves higher accuracy, even with lower bit widths. Notably, SliderQuant surpasses LLM-MQ and QUIK with higher bit widths preserving a fixed portion of FP16 outliers, and also yields clear gains over SpQR. These results show that our sliding-layer quantization mechanism is superior to existing mixed-precision quantization schemes.

## 4.2 ABLATION STUDIES

To have a better understanding of our proposed SliderQuant, we further conduct a lot of ablative experiments under both weight-only quantization and weight-activation quantization with Llama2-7B. *In the ablations, we use the fixed-size sliding quantization $\{s = 2, i = 1\}$ as the baseline.*

**Overall Design.** We first conduct experiments to study the two core components of SliderQuant, inter-layer sliding quantization (**Inter-S**) and intra-layer sliding quantization (**Intra-S**). The results are shown in Table 8. Recall that compared to the baseline fixed-size sliding quantization, Inter-S additionally introduces a progressively expanded sliding window (**PESW**) for shallow layers and a progressively contracted sliding window (**PCSW**) for deep layers. We can find that both PESW and PCSW significantly improve the performance, demonstrating the importance to consider the varying layer sensitivity to quantization of any pre-trained LLM. By extending the progressively expanded sliding design within each window, Intra-S further enhances the performance. Coupling Inter-S and Intra-S leads to the best results, confirming the effectiveness of SliderQuant's multi-level design.

**Effect of $L_s$ and $L_d$ in Inter-Layer Sliding Quantization.** After validating the effectiveness of Inter-S and Intra-S, we next study the effect of different settings for them independently. In Table 9, we provide results of Inter-S with different $L_s$ and $L_d$, namely the number of shallow layers and the number of deep layers in our design. For simplicity, we always keep $L_s$ and $L_d$ the same in the experiments. As $L_s$ and $L_d$ gradually increase from 2 to 6, the model performance steadily improves. While it also leads to larger memory and computational costs during quantization accordingly. Considering the trade-off between quantization performance and efficiency, we set $L_s = L_d = 4$ as default, as further scaling beyond this point tends to bring marginal performance improvements.

Table 8: Ablation of inter-layer sliding quantization (Inter-S) and intra-layer sliding quantization (Intra-S).

| Inter-S | | Intra-S | W4A4 | | W2A16 | |
| PESW | PCSW | | WikiText2 ↓ | C4 ↓ | WikiText2 ↓ | C4 ↓ |
|---|---|---|---|---|---|---|
| | | | 12.73 | 14.45 | 12.10 | 18.91 |
| | | ✓ | 10.34 | 13.46 | 10.71 | 16.10 |
| ✓ | | | 10.30 | 13.78 | 10.67 | 16.76 |
| | ✓ | | 9.84 | 13.31 | 10.92 | 17.31 |
| ✓ | ✓ | | 9.13 | 11.78 | 10.53 | 15.15 |
| ✓ | ✓ | ✓ | **8.34** | **11.10** | **9.59** | **13.83** |

Table 9: Ablation of inter-layer sliding quantization with different $L_s$ and $L_d$.

| $L_s$ | $L_d$ | W4A4 | | W2A16 | |
| | | WikiText2 ↓ | C4 ↓ | WikiText2 ↓ | C4 ↓ |
|---|---|---|---|---|---|
| 2 | 2 | 10.23 | 13.24 | 11.25 | 17.36 |
| 3 | 3 | 9.66 | 12.87 | 10.83 | 16.94 |
| 4 | 4 | 9.13 | 11.78 | 10.53 | 15.15 |
| 5 | 5 | 8.98 | 11.72 | 10.50 | 14.96 |
| 6 | 6 | **8.94** | **11.67** | **10.43** | **14.75** |

Table 10: Ablation of intra-layer sliding quantization with different $\gamma$. Note $N = 1/\gamma$.

| Ratio $\gamma$ | #Stage N | W4A4 | | W2A16 | |
| | | WikiText2 ↓ | C4 ↓ | WikiText2 ↓ | C4 ↓ |
|---|---|---|---|---|---|
| 1.0 | 1 | 12.73 | 14.45 | 12.10 | 18.91 |
| 0.5 | 2 | **10.34** | 13.46 | 10.71 | 16.10 |
| 0.33 | 3 | 10.56 | **13.30** | **10.67** | **15.87** |
| 0.25 | 4 | 11.32 | 14.10 | 10.83 | 16.45 |

Table 11: Ablation of fixed-size sliding quantization with larger $s$.

| Method | $s$ | $L_s \& L_d$ | W4A4 | | W2A16 | |
| | | | WikiText2 ↓ | C4 ↓ | WikiText2 ↓ | C4 ↓ |
|---|---|---|---|---|---|---|
| Baseline | 2 | - | 12.73 | 14.45 | 12.10 | 18.91 |
| | 3 | - | 11.18 | 13.94 | 11.48 | 17.23 |
| | 4 | - | 11.13 | 13.52 | 11.34 | 16.55 |
| Inter-S | 2 | 4 | **9.13** | **11.78** | **10.53** | **15.15** |

**Effect of $\gamma$ in Intra-Layer Sliding Quantization.** In this set of ablative experiments, we study the performance of Intra-S with different settings of the sliding ratio $\gamma$. For a given value of $\gamma$, all layers in each window of Inter-S are parallelly quantized along the weight/activation dimension incrementally, and the quantization process is completed in $N = 1/\gamma$ stages. As shown in Table 10, Intra-S with $N > 1$ always achieves better performance than the baseline ($N = 1$). For SliderQuant, we set $\gamma = 0.5, N = 2$ as default, considering the performance and the simplicity of implementation.

**Effect of $s$ in Fixed-Size Sliding Quantization.** To mitigate the performance differences potentially caused by varying window sizes, we increase the fixed window size $s$ for the baseline. As shown in Table 11, the performance of the baseline improves as $s$ increases from 2 to 4. Nevertheless, a substantial performance gap still remains compared to Inter-S which uses the same window size of 4 only in shallow and deep layers. This highlights the limitation of fixed-size sliding quantization, where a uniform strategy overlooks the varying quantization sensitivity across layers. These results further validate the necessity of our adaptive sliding window designs.

**Channel Scaling and LoRA.** Table 12 shows the effect of Channel Scaling (CS) and LoRA under W4A4 quantization. We observe that neither component alone achieves the best performance. Regardless of whether applied to the baseline or our SliderQuant, using both CS and LoRA together consistently yields better results than using either one individually. This demonstrates the complementary nature of the two techniques under our proposed sliding quantzation framework, well suppressing outliers in weights and activations at different layers.

Table 12: Effect of channel scaling (CS) and LoRA.

| #Bits | Methods | CS | LoRA | WikiText2 ↓ | C4 ↓ |
|---|---|---|---|---|---|
| W4A4 | Baseline | ✓ | | 20.41 | 29.67 |
| | | | ✓ | 12.73 | 14.45 |
| | | ✓ | ✓ | 13.92 | 18.44 |
| | SliderQuant | ✓ | | 9.18 | 12.21 |
| | | | ✓ | 12.95 | 17.27 |
| | | ✓ | ✓ | **8.34** | **11.10** |

**Additional Experiments and Analyses.** We include further implementation details and more experimental results *in the Appendix*, covering the following aspects: (1) the analysis of SliderQuant's quantization efficiency; (2) the ablation studies on the number of calibration samples and the group-wise quantization; (3) extended evaluation across diverse model families and quantization settings; (4) design details of SliderQuant with rotation transformations; (5) other results and visualizations; (6) other experiments and discussions conducted for the rebuttal.

## 5 CONCLUSION

In this paper, we propose SliderQuant, a new post-training quantization framework that explicitly accounts for the varying layer sensitivity to quantization of large language models. By coupling inter-layer and intra-layer sliding quantization components, SliderQuant reduces quantization errors across layers. Extensive experiments across various model families, quantization settings (e.g., W2A16, W4A4), and benchmarks demonstrate the effectiveness and generalizability of our method.

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

APPENDIX

# A    DATASETS USED IN EXPERIMENTS

**WikiText2** (Merity et al., 2017) is a popular language modeling benchmark consisting of over 2 million tokens from verified Wikipedia articles.

**C4** (Raffel et al., 2020)(Colossal Clean Crawled Corpus) is a large-scale dataset primarily used for language modeling tasks, comprising 156 billion clean tokens. It is sourced from cleaned web pages, originally from Common Crawl.

**PIQA** (Bisk et al., 2020) contains 16,000 training and 3,000 validation samples. It focuses on physical reasoning through multiple-choice questions, where models select the most appropriate solution from two options, with exactly one correct answer.

**ARC** (Clark et al., 2018) is a dataset of 7,787 genuine grade-school level, multiple-choice science questions, assembled to encourage research in advanced question-answering. The dataset is partitioned into a Challenge Set and an Easy Set, where the former contains only questions answered incorrectly by both a retrieval-based algorithm and a word co-occurrence algorithm.

**HellaSwag** (Zellers et al., 2019) contains 70,000 training and 10,000 validation samples. It focuses on commonsense reasoning by predicting the most plausible sentence continuation, sourced from crowdsourced captions and activity descriptions.

**Winogrande** (Sakaguchi et al., 2021) is a collection of 44,000 problems, which is formulated as a fill-in-a-blank task with binary options. The goal is to choose the right option for a given sentence which requires commonsense reasoning.

**BoolQ** (Clark et al., 2019) is a dataset comprising 15,942 naturally occurring yes/no questions paired with Wikipedia passages. Each example consists of a question, a passage, and a binary answer, aiming to evaluate reading comprehension and entailment-like reasoning.

**MMLU** (Hendrycks et al., 2020) is a benchmark designed to evaluate the multitask accuracy of language models across 57 diverse subjects, including elementary mathematics, U.S. history, computer science, law, and more. The dataset consists of multiple-choice questions and is intended to assess models' world knowledge and problem-solving abilities in zero-shot and few-shot settings.

**MATH-500** (Hendrycks et al., 2021) comprises 500 challenging competition-level mathematics problems sampled from the MATH dataset. These problems span various topics such as algebra, geometry, number theory, and probability, and are designed to test a model's ability to perform complex mathematical reasoning and generate step-by-step solutions.

**AIME-2024** (Jia, 2024) includes 30 problems from the 2024 American Invitational Mathematics Examination (AIME), a prestigious high school mathematics competition. The dataset serves as a benchmark for evaluating models' capabilities in solving advanced mathematical problems that require deep understanding and creative problem-solving skills.

**GSM8K** (Cobbe et al., 2021) is a dataset of 8,792 high-quality, linguistically diverse grade school math word problems created by human problem writers. The dataset is segmented into 7,473 training problems and 1,319 test problems, each requiring multi-step reasoning and basic arithmetic operations to solve.

**HumanEval+** (Liu et al., 2023) is an extension of the HumanEval dataset, consisting of 164 original programming problems designed to assess the functional correctness of code generated by language models. Each problem includes a function signature, a docstring specifying the intended functionality, and multiple test cases for evaluation.

**MBPP+** (Liu et al., 2023) is an augmented version of the Mostly Basic Programming Problems (MBPP) dataset, comprising approximately 378 crowd-sourced Python programming tasks. Each task includes a natural language description, a reference solution, and three test cases, aiming to evaluate models' abilities in basic programming and problem-solving.

# B  IMPLEMENTATION DETAILS OF SLIDERQUANT

## B.1  QUANTIZATION DETAILS

By default, we adopt a uniform quantizer for both weights and activations to maintain simplicity and efficiency, while ensuring fair comparison with other post-training quantization methods. Specifically, we apply per-channel quantization for weights and per-token quantization for activations, using the WikiText2 dataset for calibration. For particularly challenging tasks where quantization errors can have a larger impact, specifically the MoE models evaluated in Table 4 and the mathematical reasoning and code generation tasks in Table 5, we employ group-wise quantization with a group size of 128 to preserve performance, instead using the C4 dataset for calibration.

$$\text{quantizer}(\mathbf{Z}) = \text{clamp}\left(\left\lfloor \frac{\mathbf{Z}}{\alpha} \right\rceil - \beta, \ 0, \ 2^b - 1\right), \quad \alpha = \frac{\mathbf{Z}_{\max} - \mathbf{Z}_{\min}}{2^b - 1}, \quad \beta = \left\lfloor \frac{\mathbf{Z}_{\min}}{\alpha} \right\rceil, \quad \text{(A)}$$

where $\text{clamp}(x, Q_{min}, Q_{max})$ clips $x$ to the interval $[Q_{min}, Q_{max}]$; $\lfloor x \rceil$ denotes rounding to the nearest integer; $b$ is the bit-width; and $\mathbf{Z}_{\min}, \mathbf{Z}_{\max}$ are the minimum and maximum values of $\mathbf{Z}$, respectively.

## B.2  CHANNEL-WISE SCALING

Recall that, in our SliderQuant, we adopt popular used channel scaling and low-rank adaptation (LoRA) to effectively remove outliers in weights and activations at each layer of a pre-trained LLM. For the channel scaling, we simply follow the implementation of OmniQuant (Shao et al., 2024). For the Llama (Touvron et al., 2023a) and Llama2 (Touvron et al., 2023b) families, we introduce learnable scaling factors for the $Q_{proj}, K_{proj}, V_{proj}, O_{proj}, Up_{proj}$, and $Down_{proj}$ operators. For the Llama3 (Dubey et al., 2024) and Qwen2.5 (Yang et al., 2024) families, due to the presence of Group Query Attention, we align the dimensions of the Query and Key by replicating the scaling factor. All scaling factors are initialized to 1 and absorbed into the adjacent weights after quantization, introducing no additional inference overhead.

## B.3  LOW-RANK ADAPTATION

For every LLM tested in our experiments, we apply LoRA to all the linear layers to reduce quantization loss through learnable weight adjustments. The rank of LoRA is set to 4 (i.e., $r$=4), which introduces significantly fewer learnable parameters compared to full parameter fine-tuning. After quantization, the additional learnable parameters introduced by LoRA are absorbed into the model weights, resulting in no extra computational overhead during inference.

## B.4  HYPER-PARAMETER SETTINGS

For inter-layer sliding quantization, we set both the number of shallow layers $L_s$ and deep layers $L_d$ to 4 as default. For the remaining intermediate layers, we adopt a fixed-size sliding window with $s = 2$ and $i = 1$. For intra-layer sliding quantization, we set the sliding ratio $\gamma$ to 0.5. All the hyper-parameters are provided in Table A.

Table A: The detailed hyper-parameter settings of SliderQuant.

| Configuration | Setting |
|---|---|
| Calibration set | WikiText2 \| C4 |
| Number of calibration samples | 128 |
| Tokens per sample | 2048 |
| $L_s, L_d$ | 4, 4 |
| $s, i$ | 2, 1 |
| $\gamma$ | 0.5 |
| Rank of LoRA ($r$) | 4 |
| Quantization group size | channel-wise \| 128 |
| Batch size | 3 \| 1 |
| Optimizer | AdamW |
| Epochs (W2A16 \| others) | 60 \| 20 |
| Learning rate of scaling factor | 0.001 |
| Learning rate of LoRA | 0.0001 |
| Learning rate schedule | linear decay to zero |

## B.5 Quantization Hardware and Deployment Acceleration

Post-training quantization experiments for models up to 32B parameters are performed on a single NVIDIA RTX A6000 (48GB) GPU, while larger models such as the 65B and 70B variants require two A6000 GPUs. Since the A6000 serves as our primary experimental platform, its deployment results are reported in Table B. To provide a broader evaluation of hardware efficiency, we additionally benchmark the quantized models on other GPUs, including a consumer-grade NVIDIA RTX 4090 (24GB) and a data-center NVIDIA A100 (40GB), with results summarized in Tables C and D.

Importantly, our method does not rely on customized quantization kernels. All deployment tests are conducted with the widely adopted `llama.cpp` framework using a batch size of 1, a prompt length of 512 tokens, and a generation length of 128 tokens. While specialized implementations such as GPTQ (Frantar et al., 2023) or AWQ (Lin et al., 2024b) kernels may yield higher acceleration ratios, our results demonstrate that the proposed method can be directly integrated into mainstream frameworks without additional engineering efforts, ensuring practical applicability and fair comparability across hardware platforms.

Table B: Inference efficiency of quantized models on one NVIDIA RTX A6000 using `llama.cpp`. **W_Memory**: weight storage; **R_Memory**: peak runtime memory; **Tokens/s**: generated tokens per second.

| #Bits | Llama2-7B | | | Llama2-13B | | | Llama2-70B | | |
|---|---|---|---|---|---|---|---|---|---|
| | W_Memory | R_Memory | Tokens/s | W_Memory | R_Memory | Tokens/s | W_Memory | R_Memory | Tokens/s |
| FP16 | 12.55 GB | 12.67 GB | 45.89 | 24.24 GB | 24.34 GB | 24.71 | 128.48 GB | OOM | - |
| W4A16 | 3.59 GB | 3.91 GB | 116.56 | 6.91 GB | 7.64 GB | 68.66 | 36.55 GB | 36.81 GB | 15.42 |
| W3A16 | 2.41 GB | 3.00 GB | 125.66 | 4.62 GB | 4.99 GB | 74.15 | 24.76 GB | 25.04 GB | 17.90 |
| W2A16 | 1.73 GB | 2.34 GB | 135.80 | 3.29 GB | 3.68 GB | 80.21 | 17.03 GB | 17.62 GB | 19.62 |

Table C: Inference efficiency of quantized models on one NVIDIA RTX 4090 using `llama.cpp`. **W_Memory**: weight storage; **R_Memory**: peak runtime memory; **Tokens/s**: generated tokens per second.

| #Bits | Llama2-7B | | | Llama2-13B | | | Llama2-70B | | |
|---|---|---|---|---|---|---|---|---|---|
| | W_Memory | R_Memory | Tokens/s | W_Memory | R_Memory | Tokens/s | W_Memory | R_Memory | Tokens/s |
| FP16 | 12.55 GB | 13.24 GB | 62.22 | 24.24 GB | OOM | - | 128.48 GB | OOM | - |
| W4A16 | 3.59 GB | 4.26 GB | 156.09 | 6.91 GB | 7.50 GB | 92.99 | 36.55 GB | OOM | - |
| W3A16 | 2.41 GB | 3.18 GB | 188.06 | 4.62 GB | 5.32 GB | 117.20 | 24.76 GB | OOM | - |
| W2A16 | 1.73 GB | 2.42 GB | 226.73 | 3.29 GB | 3.71 GB | 143.51 | 17.03 GB | 17.80 GB | 38.67 |

Table D: Inference efficiency of quantized models on one NVIDIA A100-40GB using `llama.cpp`. **W_Memory**: weight storage; **R_Memory**: peak runtime memory; **Tokens/s**: generated tokens per second.

| #Bits | Llama2-7B | | | Llama2-13B | | | Llama2-70B | | |
|---|---|---|---|---|---|---|---|---|---|
| | W_Memory | R_Memory | Tokens/s | W_Memory | R_Memory | Tokens/s | W_Memory | R_Memory | Tokens/s |
| FP16 | 12.55 GB | 12.71 GB | 81.79 | 24.24 GB | 24.41 GB | 44.73 | 128.48 GB | OOM | - |
| W4A16 | 3.59 GB | 3.98 GB | 138.14 | 6.91 GB | 7.85 GB | 81.23 | 36.55 GB | 36.9 GB | 20.16 |
| W3A16 | 2.41 GB | 3.12 GB | 133.79 | 4.62 GB | 5.14 GB | 79.64 | 24.76 GB | 25.13 GB | 18.96 |
| W2A16 | 1.73 GB | 2.32 GB | 146.35 | 3.29 GB | 3.85 GB | 87.33 | 17.03 GB | 18.01 GB | 21.19 |

## C Implementation Details of SliderQuant with Rotation Transformations

Recall that in the Experiments section of the main paper, we apply our quantization framework SliderQuant with and without extra inference-time costs. In the default settings, all the additional parameters introduced by SliderQuant during quantization are merged into the original weights at inference. To further demonstrate the versatility of our approach, we also design a variant named SliderQuant+ that incorporates rotation transformations, corresponding to the results shown in Table 3 of the main paper. Our implementation is consistent with the QuaRot (Ashkboos et al., 2024b) codebase[1]. The detailed design is illustrated in Figure A. The non-absorbable Hadamard transformations are added after query projection ($Q_{proj}$), key projection ($K_{proj}$) and before the output projection $O_{proj}$ in the multi-head self-attention module. In the feed-forward network (FFN), the transformations are added before the down-projection ($Down_{proj}$).

---

[1]`https://github.com/spcl/QuaRot`

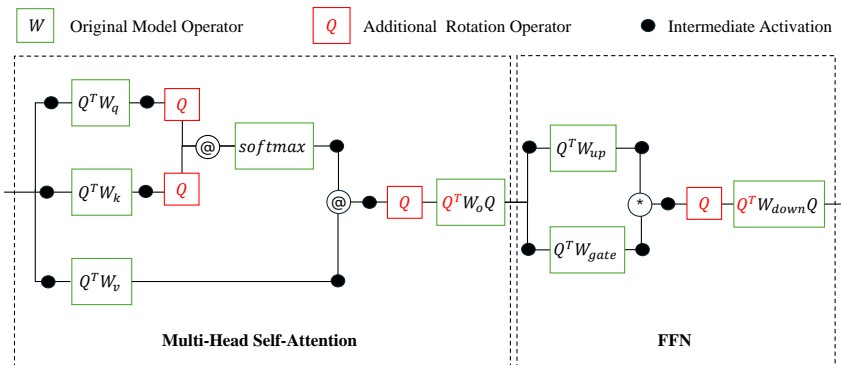

Figure A: Structural illustrations on additional rotation transformations added in SliderQuant+.

# D   QUANTIZATION EFFICIENCY OF SLIDERQUANT

To evaluate the quantization efficiency of SliderQuant, we measure its perplexity performance under different quantization time costs. As shown in Table E, SliderQuant achieves remarkably low perplexity after only 10 epochs. We adopt 20 epochs as the default setting to strike a balance between time overhead and performance. Figure B further illustrates the detailed relationship between quantization time and perplexity, demonstrating that our method converges rapidly. OmniQuant achieves 14.26|18.02 on WikiText2|C4 with the training time of 4.75 hours. Comparatively, with the training time less than 1 hour, SliderQuant achieves significantly better

Table E: The quantization efficiency of SliderQuant in terms of GPU hours. Experiments are conducted with Llama2-7B under W4A4 quantization on a single NVIDIA RTX A6000. The metric is perplexity.

| #Epoch | Wikitext2 ↓ | C4 ↓ | Time Cost (h) |
|--------|-------------|-------|---------------|
| 10 | 8.92 | 11.59 | 3.24 |
| 20 | 8.34 | 11.10 | 6.14 |
| 30 | 8.30 | 11.05 | 9.22 |
| 40 | 8.28 | 11.04 | 12.81 |

performance on both benchmarks (9.5|12.29 on WikiText2|C4). The results demonstrate the high quantization efficiency of SliderQuant. Even under extremely limited time budgets, the language models quantized with SliderQuant still maintains good performance. Extending the quantization duration can further improve performance.

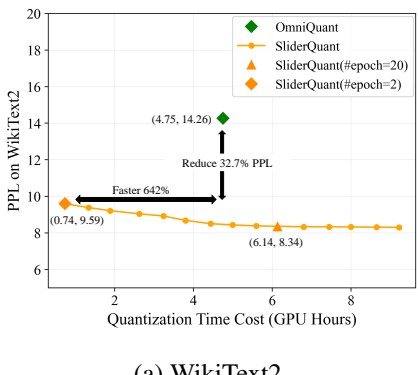

(a) WikiText2.

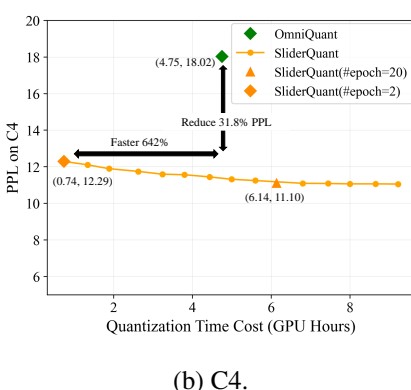

(b) C4.

Figure B: The perplexity of SliderQuant on Llama2-7B under W4A4 quantization with different quantization time costs. Experiments are conducted on a single NVIDIA RTX A6000.

# E   MORE ABLATION STUDIES

**The Number of Calibration Samples.** To ensure a fair comparison with prior methods, we use the calibration set with 128 samples as default. Here, we study the effect of the number of calibration samples to SliderQuant. The results are shown in Table F. Notably, even with only 32 samples, SliderQuant achieves the perplexity of 9.39|11.69 on Wikitext2|C4, demonstrating strong performance with a much smaller number of calibration samples. As the number of samples increases, perfor-

mance gradually improves, reaching the perplexity of 8.34|11.10 on Wikitext2|C4 with 128 samples. Further increasing the number of calibration samples leads to consistent performance improvement. These results highlight the efficiency and robustness of SliderQuant, which maintains competitive performance even under a small number of calibration samples.

**SliderQuant with Group-Wise Quantization.** To ensure inference efficiency, we use channel-wise quantization for SliderQuant in the default settings. In Table G, we provide the results of SliderQuant with group-wise quantization. Compared to channel-wise quantization, group-wise quantization leads to better performance but increased parameter storage and slower inference speed, due to the need to maintain more quantization parameters. We can find that the performance gradually improves as the group size decreases, demonstrating a trade-off between performance and efficiency. When inference cost is not a limiting factor, applying finer-grained group-wise quantization within the SliderQuant framework can lead to even better performance, showcasing its flexibility and strong quantization capability.

Table F: Effect of the number of calibration samples. We apply SliderQuant on Llama2-7B under W4A4 quantization.

| #Samples | Wikitext2 ↓ | C4 ↓ | Time Cost (h) |
|---|---|---|---|
| 32 | 9.38 | 11.69 | 1.87 h |
| 64 | 8.87 | 11.45 | 3.68 h |
| 128 | 8.34 | 11.10 | 6.14 h |
| 256 | 8.30 | 11.01 | 11.76 h |
| 512 | **8.26** | **10.93** | 23.16 h |

Table G: Ablation of SliderQuant with group-wise quantization on Llama2-7B under W2A16 quantization.

| Group Size | Wikitext2 ↓ | C4 ↓ |
|---|---|---|
| 32 | **8.20** | **10.90** |
| 64 | 8.78 | 11.45 |
| 128 | 9.15 | 12.30 |
| 256 | 9.23 | 13.11 |
| channel-wise | 9.59 | 13.83 |

# F    MORE RESULTS ACROSS DIVERSE LLM FAMILIES AND QUANTIZATION SETTINGS

**About the Results for Counterpart Methods.** To demonstrate the effectiveness of SliderQuant, we compare it with other counterpart post-training quantization methods (e.g., AWQ, GPTQ, OmniQuant). Here, we clarify the details about our reported results for the methods. For perplexity on WikiText2 and C4, we adopt the evaluation results of Llama and Llama2 models as reported in the respective papers of the counterpart methods. For models and settings not covered in prior work—such as Llama3, Qwen2.5, and certain quantization configurations (e.g., W2A16)—we evaluate the results using public code and train the models to the best of our ability. For downstream tasks, we follow a prioritized strategy: reported results in the original papers, followed by evaluation using official checkpoints, and finally, reproduction via open-source code when necessary. For a fair comparison, we implement the fixed-size sliding quantization method CBQ with the same sliding window setting as we used in SliderQuant for intermediate layers.

**The Results on Llama Model Family.** In the main paper, we report results on the Llama2, Llama3, Qwen2.5 model families. Here, we further provide results on the first version of Llama family, including Llama-7B, Llama-13B and Llama-65B, evaluated on both language generation and zero-shot commonsense reasoning tasks. As shown in Tables H and I, SliderQuant consistently achieves the best performance under various quantization settings, outperforming existing methods such as RTN, AWQ (Lin et al., 2024b), GPTQ (Frantar et al., 2023), SmoothQuant (Xiao et al., 2023), OmniQuant (Shao et al., 2024) and CBQ (Ding et al., 2025). Notably, in challenging low-bit configurations like W4A4 and W2A16, SliderQuant maintains strong generation quality and reasoning accuracy, demonstrating its robustness and precision-preserving capability.

**More Results with Weight-Activation Quantization.** In the main paper, we provide the results of Llama2-13B and Qwen2.5-14B with weight-activation quantization on zero-shot commonsense reasoning tasks. Here, we provide the additional results of other four models with different scales, including Llama2-7B, Llama2-70B, Qwen2.5-7B and Qwen2.5-32B. As shown in Table J, our SliderQuant consistently outperforms existing methods such as SmoothQuant, OmniQuant and CBQ across all tested models and tasks under W4A4 quantization. Even for smaller models like Llama2-7B and Qwen2.5-7B, which are generally more sensitive to quantization, SliderQuant achieves the best performance on average. For instance, on Llama2-7B, it reaches the average accuracy of 59.30%, notably higher than CBQ (56.26%) and OmniQuant (54.95%). Similarly, for the larger-

Table H: Results comparison of different quantization methods without extra inference-time costs on Llama model family for the language generation tasks. The metric is perplexity.

| #Bits | Method | Llama-7B | | Llama-13B | | Llama-65B | |
|---|---|---|---|---|---|---|---|
| | | Wiki ↓ | C4 ↓ | Wiki ↓ | C4 ↓ | Wiki ↓ | C4 ↓ |
| W16A16 | - | 5.68 | 7.08 | 5.09 | 6.61 | 3.53 | 5.62 |
| W4A16 | RTN | 6.43 | 7.93 | 5.55 | 6.98 | 3.87 | 5.85 |
| | AWQ | 6.08 | 7.52 | 5.34 | 6.98 | 3.76 | 5.77 |
| | GPTQ | 6.13 | 7.43 | 5.40 | 6.84 | 3.83 | 5.80 |
| | OmniQuant | 5.86 | 7.34 | 5.21 | 6.76 | 3.71 | 5.73 |
| | CBQ | 5.86 | 7.33 | 5.22 | 6.77 | 3.68 | 5.72 |
| | SliderQuant | **5.81** | **7.26** | **5.19** | **6.73** | **3.65** | **5.70** |
| W2A16 | RTN | 1.1e5 | 1.3e5 | 6.8e4 | 5.6e4 | 2.2e4 | 2.2e4 |
| | AWQ | 2.6e5 | 1.9e5 | 2.8e5 | 2.3e5 | 75.43 | 56.34 |
| | GPTQ | 2.1e3 | 690 | 5.5e3 | 2.5e3 | 55.91 | 40.58 |
| | OmniQuant | 15.47 | 24.89 | 13.21 | 18.31 | 7.58 | 10.77 |
| | CBQ | 9.65 | 13.45 | 7.96 | 11.66 | 6.56 | 9.34 |
| | SliderQuant | **9.00** | **12.91** | **7.51** | **10.43** | **5.95** | **8.36** |
| W4A4 | RTN | 2.7e2 | 4.0e2 | 2.4e3 | 1.8e3 | 3.7e4 | 8.9e3 |
| | SmoothQuant | 25.25 | 32.32 | 40.05 | 47.18 | 2.8e2 | 2.4e2 |
| | OmniQuant | 11.26 | 14.51 | 10.87 | 13.78 | 9.17 | 11.28 |
| | CBQ | 10.39 | 13.41 | 9.69 | 12.55 | 7.23 | 9.45 |
| | SliderQuant | **8.01** | **10.58** | **7.22** | **9.52** | **6.20** | **8.38** |

Table I: Results comparison of different quantization methods without extra inference-time costs on Llama model family for the zero-shot commonsense reasoning tasks. The metric is accuracy (%).

| Model | #Bits | Method | PIQA ↑ | ARC-e ↑ | ARC-c ↑ | HS ↑ | WG ↑ | BoolQ ↑ | Avg ↑ |
|---|---|---|---|---|---|---|---|---|---|
| Llama-7B | W16A16 | - | 79.43 | 73.15 | 45.05 | 76.16 | 70.24 | 75.17 | 69.87 |
| | W4A4 | SmoothQuant | 49.80 | 30.40 | 25.80 | 27.40 | 48.00 | 49.10 | 38.42 |
| | W4A4 | OmniQuant | 66.15 | 45.20 | 31.14 | 56.44 | 53.43 | 63.51 | 52.65 |
| | W4A4 | CBQ | 70.51 | 55.81 | 31.74 | 60.03 | 57.93 | 64.85 | 56.81 |
| | W4A4 | SliderQuant | **71.93** | **59.05** | **34.13** | **63.26** | **60.30** | **66.27** | **59.16** |
| Llama-13B | W16A16 | - | 80.41 | 74.71 | 47.95 | 79.08 | 73.09 | 77.92 | 72.19 |
| | W4A4 | SmoothQuant | 61.04 | 39.18 | 30.80 | 52.29 | 51.06 | 61.80 | 49.36 |
| | W4A4 | OmniQuant | 69.69 | 47.39 | 33.10 | 58.96 | 55.80 | 62.84 | 54.63 |
| | W4A4 | CBQ | 71.00 | 61.57 | 35.84 | 65.15 | 57.93 | 66.39 | 58.48 |
| | W4A4 | SliderQuant | **75.19** | **62.79** | **36.26** | **68.49** | **64.01** | **67.43** | **62.36** |
| Llama-65B | W16A16 | - | 82.37 | 79.76 | 55.38 | 84.13 | 76.95 | 84.92 | 77.25 |
| | W4A4 | SmoothQuant | 62.24 | 46.93 | 27.82 | 41.09 | 51.38 | 46.91 | 46.06 |
| | W4A4 | OmniQuant | 74.54 | 65.61 | 40.61 | 69.30 | 59.35 | 70.24 | 63.28 |
| | W4A4 | CBQ | 76.01 | 68.45 | 42.56 | 73.45 | 62.89 | 71.23 | 65.76 |
| | W4A4 | SliderQuant | **76.77** | **70.92** | **44.88** | **75.65** | **64.48** | **72.39** | **67.51** |

scale Llama2-70B, our method reaches 65.24%, outperforming CBQ (62.70%) and OmniQuant (59.93%) with clear margins.

**More Results with Weight-Only Quantization.** In the main paper, we provide the results with weight-only quantization on language generation tasks. In Table K, we further provide the results of SliderQuant with weight-only quantization on zero-shot commonsense reasoning tasks, showcasing the performance of SliderQuant under W4A16 and W2A16 quantization settings across five representative models selected from the Llama and Qwen model families. We can see that SliderQuant achieves nearly lossless performance under W4A16, with accuracies closely matching the full-precision (W16A16) counterparts. For example, on Qwen2.5-14B, SliderQuant in W4A16 achieves the average accuracy of 76.74%, compared to 77.40% for the FP16 counterpart. Under the more aggressive W2A16 setting, performance degradation becomes more noticeable but remains within an acceptable range.

As an additional complement to the results in the main paper, the comprehensive experiments across different model families (Llama, Llama2, Llama3, Qwen2.5), model scales (7B, 8B, 13B, 14B, 32B, 65B, 70B), quantization setting (W4A4,W2A16,W4A16), and benchmarks(language generation and zero-shot commonsense reasoning) further highlight the stable and superior performance of SliderQuant.

Table J: Results comparison of different quantization methods without extra inference-time costs on the zero-shot commonsense reasoning tasks under *weight-activation quantization*. The metric is accuracy (%).

| Model | #Bits | Method | PIQA ↑ | ARC-e ↑ | ARC-c ↑ | HS ↑ | WG ↑ | BoolQ ↑ | Avg ↑ |
|---|---|---|---|---|---|---|---|---|---|
| Llama2-7B | W16A16 | - | 78.84 | 74.62 | 46.42 | 75.90 | 69.46 | 78.01 | 70.54 |
| | W4A4 | SmoothQuant | 60.88 | 39.77 | 27.13 | 41.32 | 51.54 | 51.07 | 45.29 |
| | W4A4 | OmniQuant | 68.44 | 54.17 | 31.91 | 55.95 | 55.56 | 63.67 | 54.95 |
| | W4A4 | CBQ | 70.18 | 56.40 | 33.45 | 60.46 | 55.09 | 61.96 | 56.26 |
| | W4A4 | SliderQuant | **71.38** | **59.64** | **32.94** | **62.33** | **61.72** | **67.77** | **59.30** |
| Llama2-70B | W16A16 | - | 82.64 | 80.47 | 57.34 | 83.32 | 78.14 | 84.10 | 77.67 |
| | W4A4 | SmoothQuant | 61.37 | 46.21 | 31.23 | 52.65 | 50.91 | 57.28 | 49.94 |
| | W4A4 | OmniQuant | 71.71 | 59.55 | 37.20 | 66.63 | 58.17 | 66.30 | 59.93 |
| | W4A4 | CBQ | 73.24 | 62.45 | 37.30 | 69.84 | 61.23 | 72.13 | 62.70 |
| | W4A4 | SliderQuant | **75.79** | **64.14** | **37.71** | **73.02** | **65.75** | **75.02** | **65.24** |
| Qwen2.5-7B | W16A16 | - | 79.71 | 76.05 | 49.57 | 78.13 | 71.27 | 84.71 | 73.24 |
| | W4A4 | SmoothQuant | 50.44 | 25.97 | 26.02 | 25.92 | 53.20 | 38.81 | 36.73 |
| | W4A4 | OmniQuant | 53.32 | 33.84 | 24.40 | 33.75 | 52.80 | 38.78 | 39.48 |
| | W4A4 | CBQ | 62.57 | 48.99 | 31.06 | 47.31 | 54.54 | 53.67 | 49.69 |
| | W4A4 | SliderQuant | **66.92** | **59.30** | **33.62** | **55.06** | **59.43** | **62.54** | **56.15** |
| Qwen2.5-32B | W16A16 | - | 82.26 | 78.03 | 55.63 | 84.07 | 75.45 | 87.43 | 77.15 |
| | W4A4 | SmoothQuant | 61.59 | 48.15 | 32.34 | 49.91 | 52.01 | 50.55 | 49.09 |
| | W4A4 | OmniQuant | 71.16 | 61.24 | 39.93 | 65.48 | 59.27 | 65.26 | 60.39 |
| | W4A4 | CBQ | 71.44 | 63.47 | 36.86 | 59.41 | 63.93 | 69.63 | 60.79 |
| | W4A4 | SliderQuant | **72.09** | **68.10** | **40.70** | **62.12** | **63.30** | **70.14** | **62.74** |

Table K: Zero-shot commonsense reasoning results under *weight-only quantization* of SliderQuant. The evaluation metric is accuracy (%).

| Model | #Bits | Method | PIQA ↑ | ARC-e ↑ | ARC-c ↑ | HS ↑ | WG ↑ | BoolQ ↑ | Avg ↑ |
|---|---|---|---|---|---|---|---|---|---|
| Llama2-7B | W16A16 | - | 78.84 | 74.62 | 46.42 | 75.90 | 69.46 | 78.01 | 70.54 |
| | W4A16 | SliderQuant | 78.67 | 71.55 | 42.49 | 74.77 | 69.14 | 74.89 | 68.59 |
| | W2A16 | SliderQuant | 70.78 | 57.79 | 31.06 | 57.15 | 60.14 | 66.12 | 57.17 |
| Llama2-13B | W16A16 | - | 80.41 | 77.40 | 49.15 | 79.37 | 72.14 | 80.55 | 73.17 |
| | W4A16 | SliderQuant | 80.41 | 76.73 | 48.55 | 78.30 | 72.14 | 78.10 | 72.37 |
| | W2A16 | SliderQuant | 73.56 | 67.47 | 37.54 | 64.75 | 63.22 | 70.67 | 62.87 |
| Qwen2.5-7B | W16A16 | - | 79.71 | 76.05 | 49.57 | 78.13 | 71.27 | 84.71 | 73.24 |
| | W4A16 | SliderQuant | 79.22 | 75.47 | 49.20 | 76.97 | 71.27 | 83.15 | 72.55 |
| | W2A16 | SliderQuant | 68.50 | 60.31 | 34.56 | 53.02 | 59.91 | 65.05 | 56.89 |
| Qwen2.5-14B | W16A16 | - | 82.10 | 79.59 | 58.87 | 82.95 | 75.61 | 85.26 | 77.40 |
| | W4A16 | SliderQuant | 81.23 | 79.12 | 58.43 | 81.65 | 75.03 | 84.95 | 76.74 |
| | W2A16 | SliderQuant | 72.09 | 68.10 | 40.70 | 62.12 | 63.30 | 70.14 | 62.74 |
| Qwen2.5-32B | W16A16 | - | 82.26 | 81.03 | 58.45 | 84.13 | 77.61 | 87.49 | 78.50 |
| | W4A16 | SliderQuant | 81.88 | 80.39 | 57.59 | 83.17 | 77.27 | 86.74 | 77.84 |
| | W2A16 | SliderQuant | 74.59 | 72.01 | 46.42 | 65.12 | 66.30 | 56.85 | 63.55 |

# G  MORE EXPERIMENTS AND DISCUSSIONS FOR THE REBUTTAL

In this section, we provide more experiments and discussions conducted for the rebuttal.

## G.1  STUDY ON FOUR SLIDING WINDOW SCHEDULE KNOBS

Our SliderQuant has four schedule knobs including a progressively expanded sliding window (PESW) for $L_s$ shallow layers, a fixed-size sliding window withe the size of $s$ for $L_i$ intermediate layers and a progressively contracted sliding window (PCSW) for $L_d$ deep layers used in inter-layer sliding quantization (Inter-S), and an incremental quantization ratio $\gamma$ for intra-layer sliding quantization (Intra-S) within each window of Inter-S. In the Subsection E of the main paper, we used Llama2-7B to study their choices from the perspectives of their individual roles and combined roles, and chose $L_s = 4, s = 2, L_d = 4, \gamma = 0.5$ as the default setting of our SliderQuant applied to all different LLMs tested in our paper, for a simple implementation. Indeed, this default setting is not optimal even for Llama2-7B, let alone other LLMs. To better compare their choices, we additionally conducted multiple sets of ablative experiments with larger and recently released Qwen2.5-14B under W4A4 quantization. Detailed results are summarized in the below Table L, Table M, Table N and Table O, from which we can see that our SliderQuant: (1) with our default setting, four sched-

ule knobs are complementary to each other (see Table L); (2) for each individual knob, its default setting is usually not the best for both Llama2-7B and Qwen2.5-14B (see Table M and Table N); (3) combining it with PESW and PCSW is significantly better than merely using fixed-size sliding quantization to all layers, both on Llama2-7B and Qwen2.5-14B (see Table O). Although our SliderQuant with the default setting already achieves superior results to existing PTQ methods,

Table L: Ablation of inter-layer sliding quantization (Inter-S) and intra-layer sliding quantization (Intra-S) on Qwen2.5-14B under W4A4 quantization.

| Inter-S | | Intra-S | Wikitext2 ↓ | C4 ↓ |
|---|---|---|---|---|
| PESW | PCSW | | | |
| | | | 17.41 | 25.20 |
| | | ✓ | 16.01 | 23.71 |
| ✓ | | | 15.42 | 20.71 |
| | ✓ | | 14.08 | 21.48 |
| ✓ | ✓ | | 12.99 | 18.40 |
| ✓ | ✓ | ✓ | **11.00** | **16.60** |

Table M: Ablation of inter-layer sliding quantization with different choices of $L_s$ and $L_d$ on Qwen2.5-14B under W4A4 quantization.

| $L_s$ | $L_d$ | Wikitext2 ↓ | C4 ↓ |
|---|---|---|---|
| 2 | 2 | 14.95 | 20.30 |
| 3 | 3 | 13.83 | 19.07 |
| _4_ | _4_ | _12.99_ | _18.40_ |
| 5 | 5 | 12.45 | 18.12 |
| 6 | 6 | **12.22** | **18.01** |

Table N: Ablation of intra-layer sliding quantization with different choices of $\gamma$ on Qwen2.5-14B under W4A4 quantization.

| Ratio $\gamma$ | #Stage N | Wikitext2 ↓ | C4 ↓ |
|---|---|---|---|
| 1.0 | 1 | 17.41 | 25.20 |
| _0.5_ | _2_ | _16.01_ | _23.71_ |
| 0.33 | 3 | **15.67** | 23.03 |
| 0.25 | 4 | 16.07 | **22.71** |

Table O: Ablation of fixed-size sliding quantization with different choices of window size $s$ on Qwen2.5-14B under W4A4 quantization.

| Method | $s$ | $L_s \& L_d$ | Wikitext2 ↓ | C4 ↓ |
|---|---|---|---|---|
| | 2 | - | 17.41 | 25.20 |
| Baseline | 3 | - | 15.94 | 24.15 |
| | 4 | - | 14.75 | 21.41 |
| Inter-S | 2 | 4 | **12.99** | **18.40** |

these ablations indicate that there is still room to get improved quantization performance by choosing better settings of these four schedule knobs for different LLMs. *Underlined entries indicate our default hyperparameter settings and their corresponding experimental results.*

### G.2 ADAPTIVE SLIDING-WINDOW QUANTIZATION *vs* REPEATED OPTIMIZATION

In our SliderQuant, the first and the last layers are used as the anchor layer when quantizing shallow and deep layers, which means that they are quantized more times than the other layers. Naturally, a critical question is whether performance gain comes more from repeated optimization or our adaptive sliding quantization. To explore this question, we conducted an ablation to compare the roles of the sliding-window quantization design and the repeated optimization (quantization of all layers in each window by multiple times) on Llama2-7B and Qwen2.5-14B under W4A4 quantization. Detailed results are summarized in the below Table P, from which we can see that, on both Llama2-7B and Qwen2.5-14B: (1) for fixed-size sliding quantization with the window size of $s = 2$ or $s = 4$ applied to all model layers, performing multiple times of quantization of all layers in each window can improve the quantization performance, but the gain against its corresponding single time baseline is marginal ($< 1$ perplexity); (2) comparatively, retaining a fixed-size sliding window $s = 2$ for intermediate layers, by introducing a progressively expanded sliding window (PESW) for $L_s = 4$ shallow layers, a progressively contracted sliding window (PCSW) for $L_d = 4$ deep layers, our SliderQuant with this default setting achieves significant perplexity reductions (1.54 to 3.60 for Llama2-7B and 1.23 to 4.42 for Qwen2.5-14B on Wikitext2, with similar reductions observed on C4) over all fixed-sized sliding quantization baselines and its repeated variants. These experimental results indicate that our adaptive sliding quantization contributes significantly more to the final improvement of quantization performance compared to merely applying fixed-size sliding window to all layers of FP16 LLMs (even with repeated optimization), which well echoes the importance of our empirical observations and method's motivation.

### G.3 TRAINING TIME COST AND MEMORY OVERHEAD

We also conducted an ablation to compare the training time and the memory overhead of our SliderQuant (including our default version and a fast version), the fixed-size sliding quantization baseline (CBQ, which is closely related to our SliderQuant) and OmniQuant on both Llama2-7B and

Table P: Ablation of comparing SliderQuant with repeated fixed-size sliding quantization on Llama2-7B under W4A4 quantization. The repeated baseline applies quantization to all layers in each window multiple times with window size $s$. Left: Llama2-7B. Right: Qwen2.5-14B.

| $s$ | $L_s\&L_d$ | # Repetitions | Wikitext2 | C4 | | $s$ | $L_s\&L_d$ | # Repetitions | Wikitext2 | C4 |
|---|---|---|---|---|---|---|---|---|---|---|
| 2 | - | 1 | 12.73 | 14.45 | | 2 | - | 1 | 17.41 | 25.20 |
| 2 | - | 2 | 12.58 | 14.23 | | 2 | - | 2 | 17.23 | 25.12 |
| 2 | - | 3 | 12.41 | 14.15 | | 2 | - | 3 | 17.01 | 24.97 |
| 2 | - | 4 | 12.11 | 14.08 | | 2 | - | 4 | 16.91 | 24.81 |
| 4 | - | 1 | 11.13 | 13.52 | | 4 | - | 1 | 14.75 | 21.41 |
| 4 | - | 2 | 10.94 | 13.39 | | 4 | - | 2 | 14.54 | 20.93 |
| 4 | - | 3 | 10.79 | 13.25 | | 4 | - | 3 | 14.35 | 20.71 |
| 4 | - | 4 | 10.67 | 13.12 | | 4 | - | 4 | 14.21 | 20.65 |
| 2 | 4 | 1 | **9.13** | **11.78** | | 2 | 4 | 1 | **12.99** | **18.40** |

Qwen2.5-14B under W4A4 quantization. In the experiments, we always use the same number of calibration samples (128), the same batch size (1) and the same number of training epochs (20) for all methods unless otherwise stated. Note GPTQ is known as an efficient PTQ method but is tailored to weight-only quantization and is less related to our method, so it is not compared here. Detailed results are summarized in the below Table Q, from which we can see that, on both Llama2-7B and Qwen2.5-14B under W4A4 quantization: (1) in terms of training time, our SliderQuant (with the default setting of $L_s = 4, s = 2, L_d = 4, \gamma = 0.5$) is $1.08\times$ to $1.55\times$ slower than OmniQuant and the fixed-sized sliding quantization (CBQ) with the window size $s = 2$, but is faster than the fixed-sized sliding quantization with the window size of $s = 4$ or $s = 3$, and our SliderQuant-Fast (the only change is the number of training epochs is 10 instead of 20) is the fastest ($1.21\times$ to $1.78\times$ faster than OmniQuant and the fixed-sized sliding quantization (CBQ) with the window size $s = 2$) and achieves slightly worse quantization results compared to SliderQuant; (2) in terms of memory overhead, our SliderQuant (with the default setting of $L_s = 4, s = 2, L_d = 4, \gamma = 0.5$), SliderQuant-Fast and the fixed-sized sliding quantization (CBQ) with the window size of $s = 4$ have the same memory usage, which is around $2\times$ compared to the memory usage by the fixed-sized sliding quantization (CBQ) with the window size of $s = 2$ and OmniQuant; (3) in terms of model accuracy, both SliderQuant (with the default setting of $L_s = 4, s = 2, L_d = 4, \gamma = 0.5$) and SliderQuant-Fast always achieve significantly better perplexity than other counterpart methods, showing perplexity reductions ranging from 2.21 to 23.70 on Wikitext-2 and 1.93 to 41.15 on C4. Furthermore, Figure B in this Appendix presents a comprehensive view of SliderQuant's performance-efficiency trade-offs across different training time configurations. These results demonstrate that SliderQuant can achieve even greater training efficiency, showcasing up to $6.42\times$ speedup compared to OmniQuant on Llama2-7B while maintaining better perplexity on both datasets. Summarily, these results show that the training cost of SliderQuant is decent, which guarantees the practicality of our method to quantize different LLMs, also thanks to its simplicity.

Table Q: Comparison of the training time and the memory overhead of our SliderQuant, fixed-size sliding quantization baseline and OmniQuant on Llama2-7B and Qwen2.5-14B under W4A4 quantization. All experiments are conducted on a single NVIDIA A6000-48G GPU, and we set the number of calibration samples to 128 (a popular choice in the quantization community), the batch size to 1, and the number of training epochs to 20 for all methods except our SliderQuant-Fast with 10 training epochs. Best results are bolded.

| Model | Method | Wikitext2 ↓ | C4 ↓ | Training Time (GPU Hours) | Memory Overhead (GB) |
|---|---|---|---|---|---|
| Llama2-7B | OmniQuant | 14.26 | 18.02 | 4.75 | **15.04** |
| | Fixed-sized sliding (s=2, CBQ) | 12.73 | 14.45 | 5.66 | 16.24 |
| | Fixed-sized sliding (s=3, CBQ) | 11.18 | 13.94 | 7.65 | 23.02 |
| | Fixed-sized sliding (s=4, CBQ) | 11.13 | 13.52 | 9.16 | 29.80 |
| | SliderQuant-Fast | 8.92 | 11.59 | **3.24** | 29.80 |
| | SliderQuant | **8.34** | **11.10** | 6.14 | 29.80 |
| Qwen2.5-14B | OmniQuant | 34.70 | 61.75 | 7.38 | 22.12 |
| | Fixed-sized sliding (s=2, CBQ) | 17.41 | 25.20 | 10.83 | **21.12** |
| | Fixed-sized sliding (s=3, CBQ) | 15.94 | 24.15 | 13.56 | 30.23 |
| | Fixed-sized sliding (s=4, CBQ) | 14.75 | 21.41 | 15.36 | 43.71 |
| | SliderQuant-Fast | 12.19 | 17.53 | **6.08** | 43.71 |
| | SliderQuant | **11.00** | **16.60** | 11.43 | 43.71 |

### G.4 DATA-EFFICIENCY AND CALIBRATION ROBUSTNESS

In the below Table R, we provide an ablation to study the performance of our SliderQuant under a smaller number of calibration samples 32, 64 and a larger number of calibration samples 256, besides the default number of calibration samples 128, and we compare our results with the reported results in the original papers of OmniQuant and DuQuant (a top rotation-based PTQ method). We can see that: (1) when using a smaller or larger number of calibration samples, our SliderQuant and

Table R: Comparison of calibration data efficiency under W4A4 quantization. Left: Llama-7B, comparing SliderQuant and OmniQuant with different numbers of calibration samples. Right: Llama2-7B, comparing SliderQuant+ (SliderQuant with rotation) and DuQuant with different numbers of calibration samples. Underlines denote the default configuration; bold indicates the best results.

| #Samples | Method | Wikitext2 | C4 | | #Samples | Method | Wikitext2 | C4 |
|---|---|---|---|---|---|---|---|---|
| 32 | OmniQuant | 11.48 | 14.80 | | 32 | DuQuant | 6.31 | 7.99 |
| 32 | SliderQuant | 8.71 | 11.19 | | 32 | SliderQuant+ | 5.83 | 7.85 |
| 64 | OmniQuant | 11.40 | 14.57 | | 64 | DuQuant | 6.29 | 7.88 |
| 64 | SliderQuant | 8.30 | 11.13 | | 64 | SliderQuant+ | 5.81 | 7.83 |
| 128 | OmniQuant | 11.23 | 14.61 | | 128 | DuQuant | 6.28 | 7.90 |
| 128 | SliderQuant | **8.01** | **10.58** | | 128 | SliderQuant+ | 5.71 | 7.68 |
| 256 | OmniQuant | 11.41 | 14.90 | | 256 | DuQuant | 6.23 | 7.88 |
| 256 | SliderQuant | 8.49 | 11.26 | | 256 | SliderQuant+ | **5.64** | **7.53** |

SliderQuant+ (using rotation transformations) achieve significantly better results than OmniQuant and also better results than DuQuant (using learnable rotation transformations) consistently; (2) intriguingly, even with 32 calibration samples, our SliderQuant and SliderQuant+ show superior performance than OmniQuant and DuQuant with 256 calibration samples, respectively.

## H VISUALIZATIONS OF THE QUANTIZATION IMPACT OF DIFFERENT LAYERS TO MODEL ACCURACY

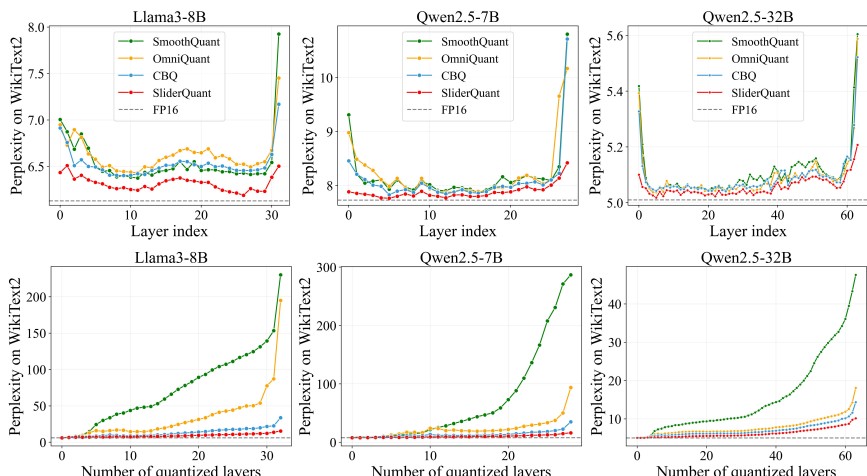

Figure C: Illustrations on the quantization impact of different layers to model accuracy: (1) quantizing a single layer (the first row) and (2) quantizing the first $l$ layers (the second row) of Llama3-8B, Qwen2.5-7B and Qwen2.5-32B. Here, we select three representative layer-wise, block-wise and multi-block-wise quantization methods, SmoothQuant, OmniQuant and CBQ, and examine them in 4-bit weight-activation (W4A4) quantization on WikiText2.

To further validate our empirical observation that *different layers in LLMs exhibit varying sensitivity to quantization*, we provide additional visualizations on Llama3-8B, Qwen2.5-7B, and Qwen2.5-32B, as shown in Figure C. Consistent with the observations discussed in the main paper (illustrated in Figure 1), these results reveal several important trends. First, for all tested LLMs, intermediate layers tend to be less sensitive to quantization, incurring smaller accuracy degradation compared to

shallow and deep layers. This confirms that shallow and deep layers are more difficult to quantize and require special attention. Second, the first and last layers exhibit the highest quantization sensitivity, leading to the most significant increases in perplexity when they are quantized. This highlights their critical role in maintaining model fidelity. Third, as more layers are quantized sequentially from shallow to deep, the cumulative quantization error increases gradually, further demonstrating the compounding effect of poor quantization in sensitive layers. Among all methods, SliderQuant consistently achieves the lowest perplexity in both the single-layer and cumulative-layer quantization settings. This is because it explicitly focuses on reducing quantization errors in the more vulnerable shallow and deep layers, effectively controlling overall errors propagation. These additional visualizations provide further empirical evidence for layer-wise differences in quantization sensitivity, and reinforce the design rationale of SliderQuant. They demonstrate the necessity of quantization-aware strategies that account for such sensitivity variation, especially when targeting low-bit quantization.

# I   VISUALIZATIONS OF WEIGHTS AND ACTIVATIONS IN SLIDERQUANT

To better understand how SliderQuant improves the quantization process, we visualize the numerical ranges of both activations and weights before and after applying it. Specifically, we compute the range as the difference between the maximum and minimum values—per channel for weights and per token for activations. Large value ranges are known to complicate quantization, especially under low-bit settings, as they increase the risk of information loss. Therefore, reducing these ranges can significantly ease quantization and improve accuracy. Taking Llama2-7B under the W4A4 quantization setting as an example, we examine the value ranges after merging the learnable parameters from OmniQuant and SliderQuant into the original weights, without actually applying quantization. This allows us to isolate the effect of these methods on the intrinsic distribution of the weights and activations, offering a clearer view of the quantization difficulty induced by each approach. As shown in Figures D to F, SliderQuant consistently reduces the channel-wise weight ranges across shallow, middle, and deep layers, clearly outperforming OmniQuant. Similarly, as shown in Figures G to R, the token-wise activation ranges after applying SliderQuant are significantly smaller than those in the original model and OmniQuant across a wide range of representative samples and layers. By simultaneously compressing the value ranges of both activations and weights across different model depths and inputs, SliderQuant reduces quantization difficulty and enables effective low-bit quantization with minimal performance degradation. This dual-range suppression is a key factor behind SliderQuant's robustness and near-lossless performance in challenging low-bit regimes.

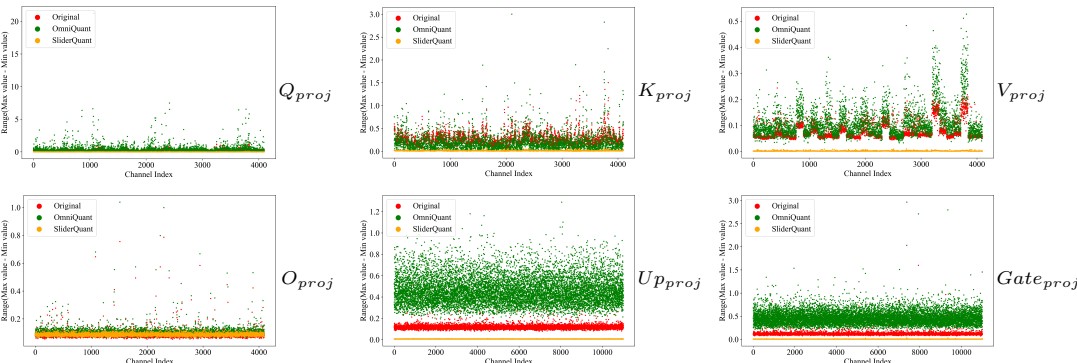

Figure D: Visualization of channel-wise weight ranges (max–min) in the 1st layer of Llama2-7B under W4A4 quantization.

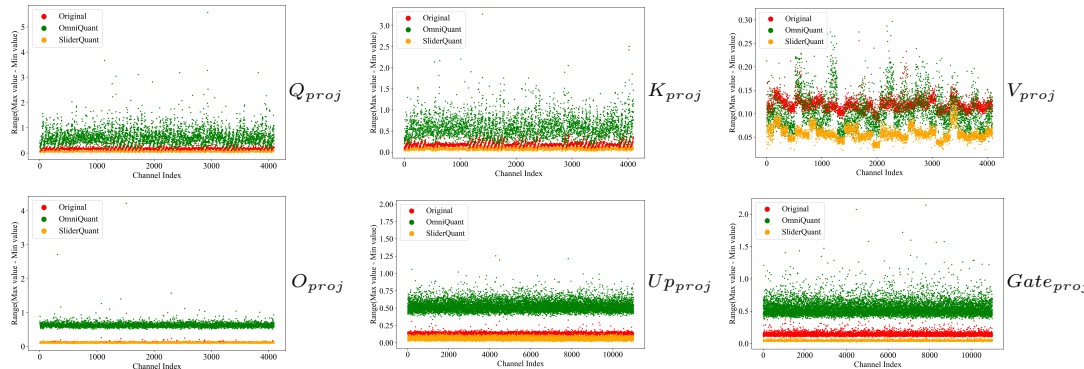

Figure E: Visualization of channel-wise weight ranges (max–min) in the 18th layer of Llama2-7B under W4A4 quantization.

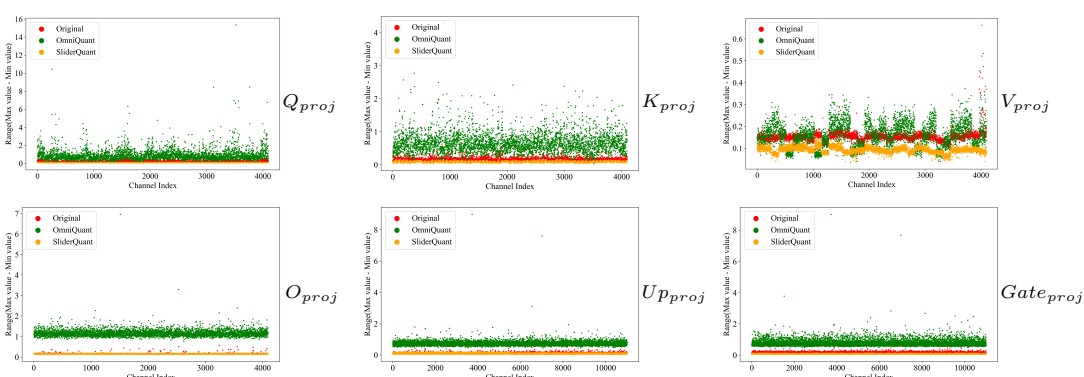

Figure F: Visualization of channel-wise weight ranges (max–min) in the 31st layer of Llama2-7B under W4A4 quantization.

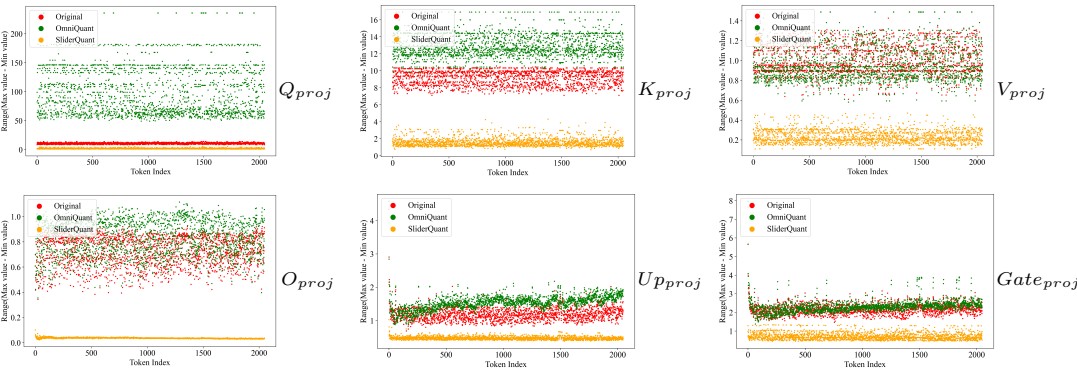

Figure G: Visualization of token-wise activation ranges (max–min) in the 1st layer of Llama2-7B for the first sample of 4 samples randomly selected from Wikitext2 under W4A4 quantization.

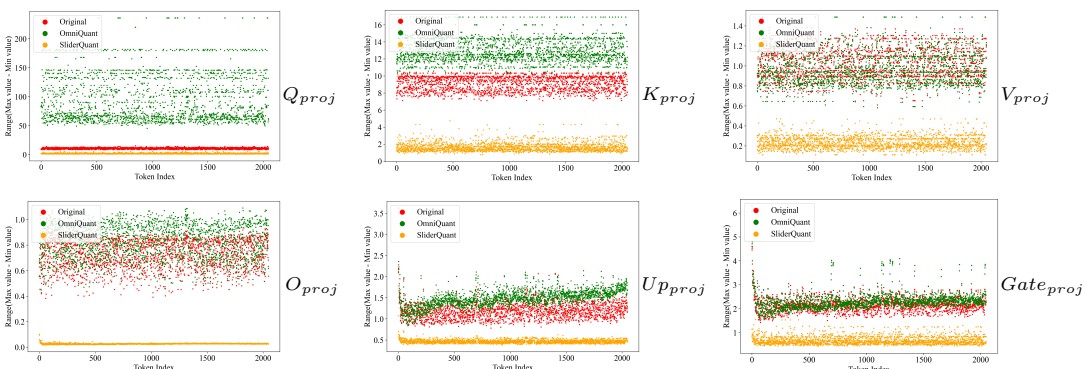

Figure H: Visualization of token-wise activation ranges (max–min) in the 1st layer of Llama2-7B for the second sample of 4 samples randomly selected from Wikitext2 under W4A4 quantization.

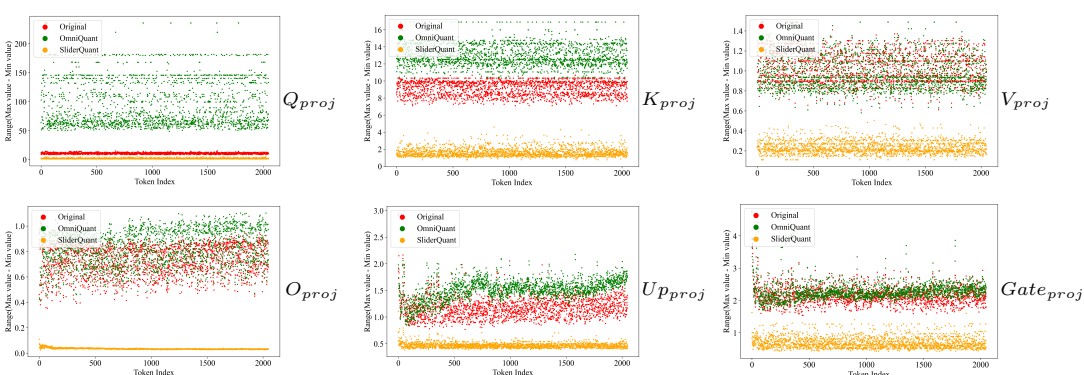

Figure I: Visualization of token-wise activation ranges (max–min) in the 1st layer of Llama2-7B for the third sample of 4 samples randomly selected from Wikitext2 under W4A4 quantization.

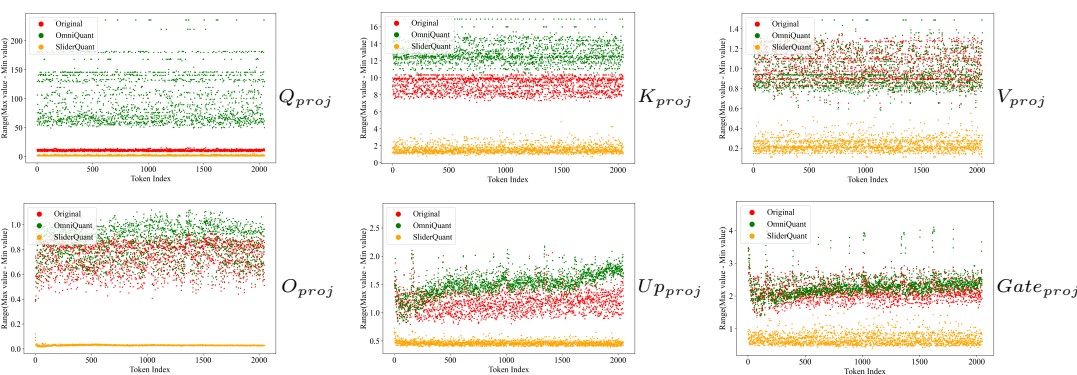

Figure J: Visualization of token-wise activation ranges (max–min) in the 1st layer of Llama2-7B for the fourth sample of 4 samples randomly selected from Wikitext2 under W4A4 quantization.

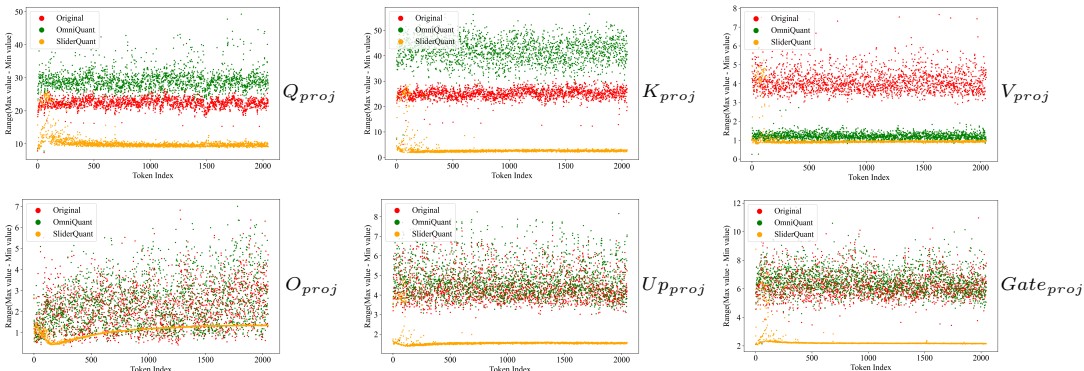

Figure K: Visualization of token-wise activation ranges (max–min) in the 18th layer of Llama2-7B for the first sample of 4 samples randomly selected from Wikitext2 under W4A4 quantization.

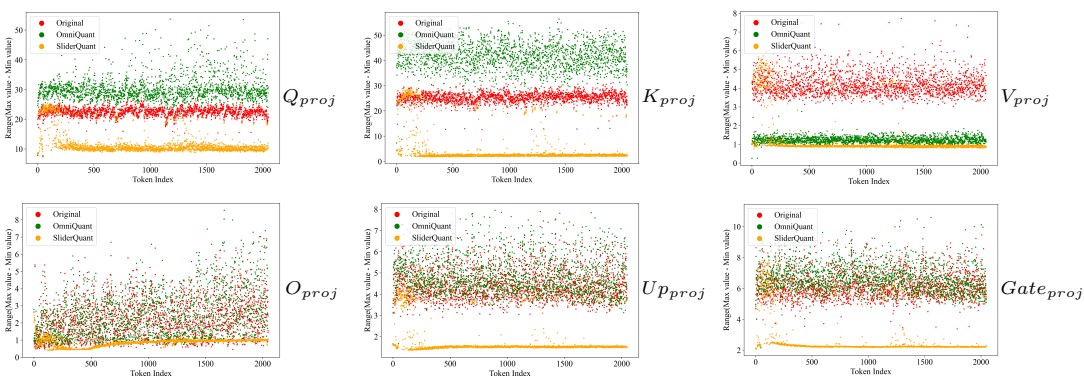

Figure L: Visualization of token-wise activation ranges (max–min) in the 18th layer of Llama2-7B for the second sample of 4 samples randomly selected from Wikitext2 under W4A4 quantization.

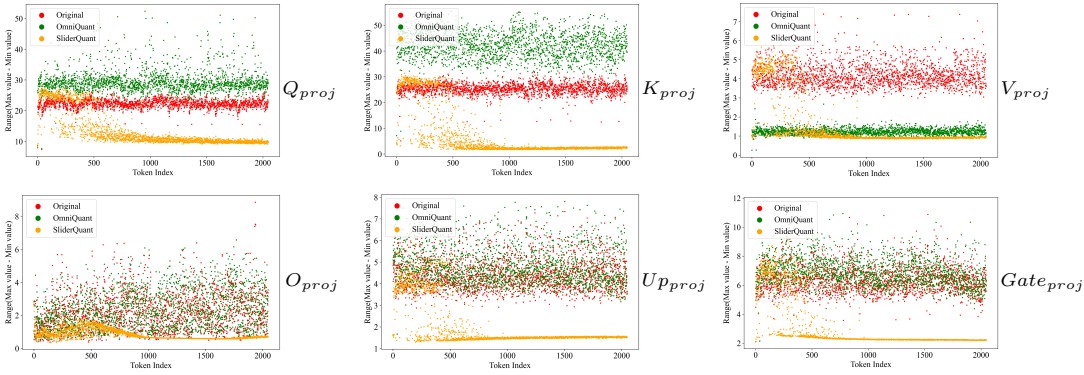

Figure M: Visualization of token-wise activation ranges (max–min) in the 18th layer of Llama2-7B for the third sample of 4 samples randomly selected from Wikitext2 under W4A4 quantization.

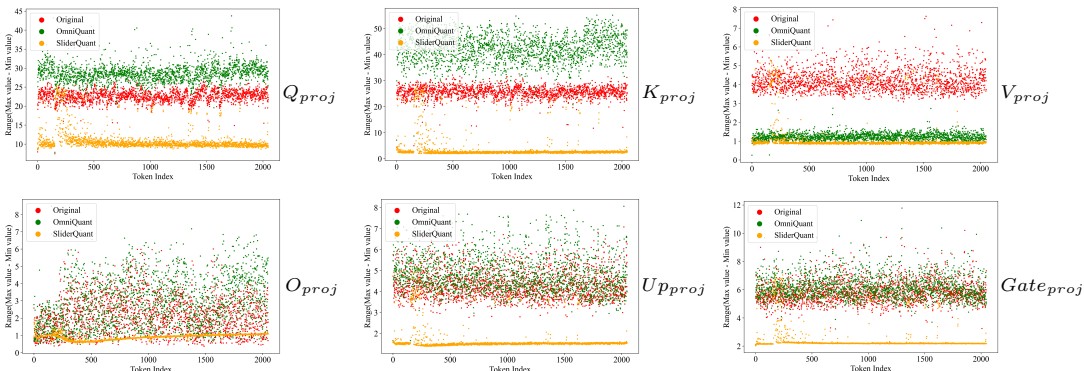

Figure N: Visualization of token-wise activation ranges (max–min) in the 18th layer of Llama2-7B for the fourth sample of 4 samples randomly selected from Wikitext2 under W4A4 quantization.

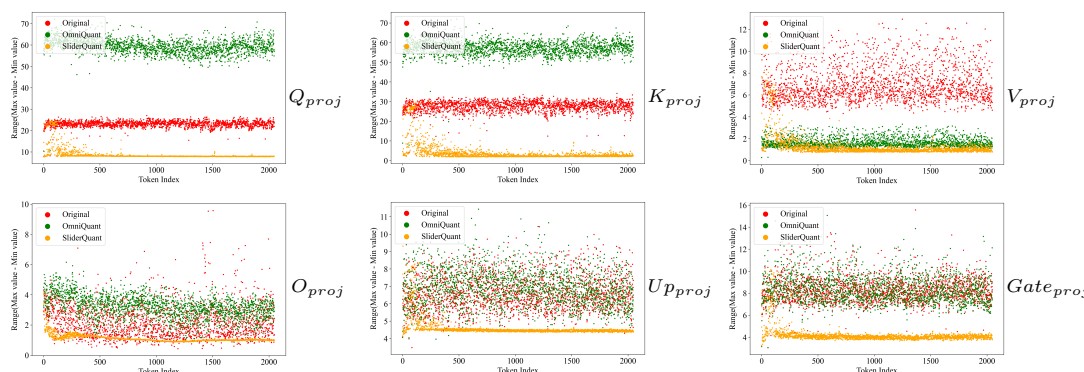

Figure O: Visualization of token-wise activation ranges (max–min) in the 31st layer of Llama2-7B for the first sample of 4 samples randomly selected from Wikitext2 under W4A4 quantization.

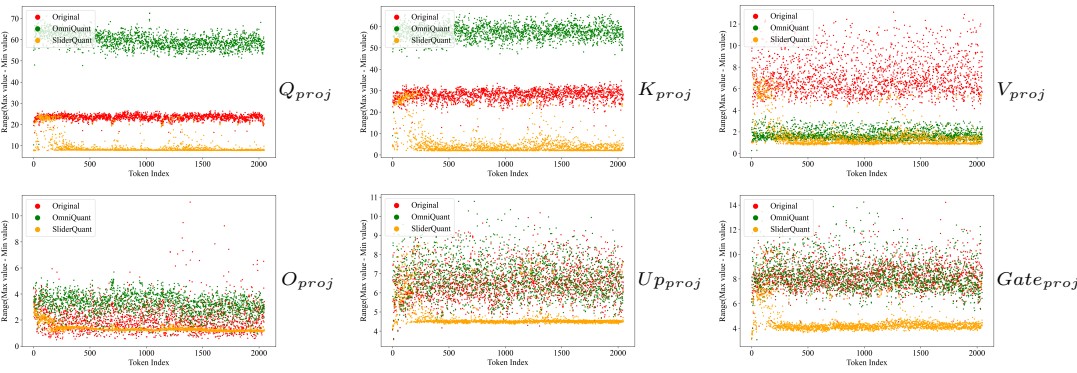

Figure P: Visualization of token-wise activation ranges (max–min) in the 31st layer of Llama2-7B for the second sample of 4 samples randomly selected from Wikitext2 under W4A4 quantization.

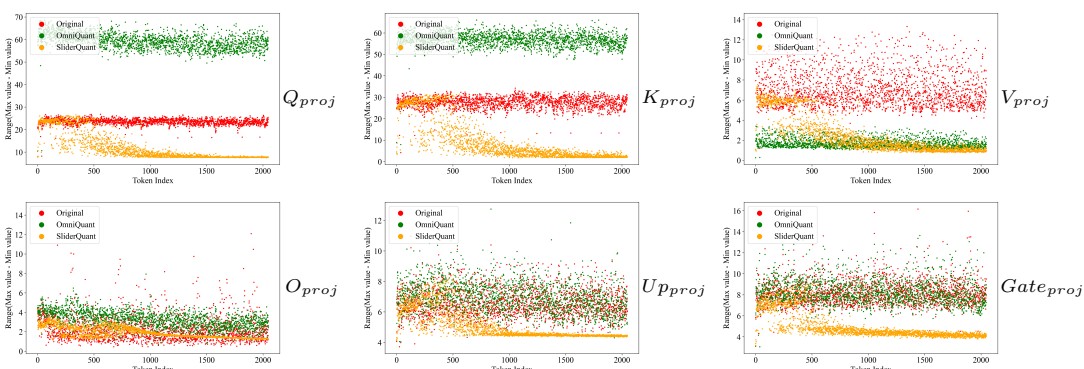

Figure Q: Visualization of token-wise activation ranges (max–min) in the 31st layer of Llama2-7B for the third sample of 4 samples randomly selected from Wikitext2 under W4A4 quantization.

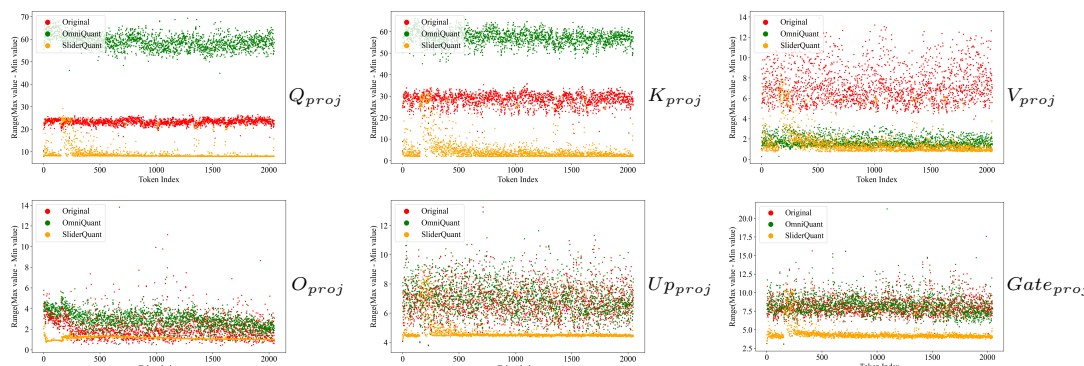

Figure R: Visualization of token-wise activation ranges (max–min) in the 31st layer of Llama2-7B for the fourth sample of 4 samples randomly selected from Wikitext2 under W4A4 quantization.

