# OpenReview forum: "SliderQuant: Accurate Post-Training Quantization for LLMs"
_ICLR.cc/2026/Conference — ICLR 2026 Poster_

### Official Review · Reviewer_BEoY · 2025-10-23

**Soundness:** 3
**Presentation:** 3
**Contribution:** 2
**Rating:** 4
**Confidence:** 3

**Summary:**

This paper introduces a new PTQ framework that accounts for varying layer sensitivities in large language models. The authors observe that the first and last layers are far more quantization-sensitive than intermediate ones and propose SliderQuant, which adaptively applies inter-layer and intra-layer sliding quantization on weights or weights+activations to mitigate these effects. This design tailors window sizes and quantization granularity across layers, effectively reducing accumulated quantization errors. Experiments on several models and benchmarks show its superior performance at low bitwidths.

**Strengths:**

- The paper is clearly written and well-organized, with intuitive figures and tables that make the methodology and results easy to follow.
- The experimental evaluation is comprehensive, covering a wide range of model families including Llama, Qwen, and MoE architectures. The authors conduct extensive tests not only on standard language generation and reasoning benchmarks but also on more challenging tasks like mathematical reasoning, providing rich empirical evidence for the method’s effectiveness. The ablation studies are also sufficient for understanding different factors that could affect the performance.

**Weaknesses:**

- The empirical observations and methodological novelty are limited. Specifically:
  - The finding that different layers of LLMs exhibit varying sensitivity to quantization—particularly that the first and last layers are most sensitive—has already been clearly identified in several prior works (e.g. [[1]](https://arxiv.org/abs/2412.03599?utm_source=chatgpt.com), [[2]](https://nicsefc.ee.tsinghua.edu.cn/%2Fnics_file%2Fpdf%2F5c805adc-b555-499f-9882-5ca35ce674b5.pdf)), so this observation might not be suitable to be regarded as a novel contribution.
  - The proposed method, while effective and technically sound, is largely incremental. Its design combines previously explored ideas, such as sliding-window quantization (e.g. [[3]](https://arxiv.org/abs/2405.06219), [[4]](https://arxiv.org/abs/2312.07950)) and layer-wise sensitivity-aware quantization (e.g. [[1]](https://arxiv.org/abs/2412.03599?utm_source=chatgpt.com), [[2]](https://nicsefc.ee.tsinghua.edu.cn/%2Fnics_file%2Fpdf%2F5c805adc-b555-499f-9882-5ca35ce674b5.pdf))), and the concept of quantization synergy across successive layers somewhat resembles extensions of the GPTQ framework's concepts.
- Some experimental results have limited practical significance. For instance, in Table 2, all methods (including SliderQuant) show more than 10% accuracy degradation under W4A4 quantization on almost all the benchmarks, and similar issues appear in Table 5 under W2A16. Although SliderQuant achieves numerically better results than baselines, such large performance drops render these configurations impractical, raising doubts about the meaningfulness of these ultra-low-bit experiments.

**Questions:**

- Could the authors provide more theoretical or intuitive explanations for the observations that the first / last layers are much more sensitive to quantization, beyond the empirical observations? I believe these could provide insights to future studies and add more novelty to this paper.
- While the bitwidth is set to be the same for all layers in SliderQuant, mixed-precision quantization is also a popular way for LLM quantization that could utilize the characteristics that different layers have different sensitivity to quantization  (e.g. [[2]](https://nicsefc.ee.tsinghua.edu.cn/%2Fnics_file%2Fpdf%2F5c805adc-b555-499f-9882-5ca35ce674b5.pdf)). Could the authors provide some comparison with such kind of methods, to show whether the sliding-layer mechanism could complement or outperform mixed-precision strategies?

---

> ### Author Response · Authors · 2025-11-24
> **Rebuttal to Official Review by Reviewer BEoY: 1/6**
>
> Thanks for your detailed review and the recognition of the technical soundness of our method, good presentation, comprehensive experimental benchmark and ablations of our work. Next, we address your concerns and questions one by one.
>
> **1. To your comments about the main weakness** “The empirical observations and methodological novelty are limited. Specifically: The finding that … contribution. The proposed method … concepts.”
>
> **Our responses:** *we politely argue that both the novelty of our empirical observations and the novelty of our methodology are **decent rather than incremental***. It seems that your judgements are due to some potential misunderstandings of [1], [2] and our work, and missed critical clarifications and discussions in our original paper submission, which are thoroughly clarified in the following two parts.
>
> **Part 1: About the novelty of our empirical observations**: the differences of our method to existing mixed-precision quantization methods in the quantization regime, focus and design.
>
> Thank you for pointing out these two mixed-precision quantization works [[1]](https://arxiv.org/pdf/2412.03599) (Nanda et al., arXiv 2024, a non-peer reviewed manuscript) and [[2]](https://nicsefc.ee.tsinghua.edu.cn/%2Fnics_file%2Fpdf%2F5c805adc-b555-499f-9882-5ca35ce674b5.pdf) (Li et al., NeurIPS ENLSP Workshop 2023, a short paper). **Generally, both [1] and [2] have not explicitly and consistently identified** that the first and last layers of different full-precision LLMs are most sensitive to quantization in terms of quantization impact on model accuracy:
>
> **(i)** Layer sensitivity to quantization is a basic concept in mixed-precision quantization research, **which is estimated before quantization**. While in our work, **layer sensitivity to quantization (precisely speaking, quantization impact) is defined as the real quantization error** and our empirical observations are from evaluating recent layer-wise, block-wise and multi-block-wise post-training quantization methods on a lot of currently prevailing LLMs under the challenging 4-bit weight-activation quantization. **Actually, in the Related Work section** (Line#136-140, Line#146-149, Line#154-157 and Line#167-168) of our original paper submission, **we already had an in-depth discussion** of seminal mixed-precision quantization research works for LLMs, such as, Q-BERT (Shen et al., AAAI 2020), GOBO (Zadeh et al., MICRO 2020), SpQR (Dettmers et al., ICLR 2024) and AWQ (Lin et al., MLSys 2024) using different mixed-precision schemes for weight-only quantization of LLMs, LLM.int8() (Dettmers et al., NeurIPS 2022) and QUIK (Ashkboos et al., EMNLP 2024) using different mixed-precision schemes for weight-activation quantization of LLMs. In the line of mixed-precision quantization research, *its basic premise is that weights/activations at the same layer or different layers are expected to exhibit different sensitivity to quantization*. Under this premise, retaining few sensitive weights/activations (i.e., outliers, typically identified as a small portion of weights/activations having the larger magnitude than the others) at the same layer or few sensitive layers in high-precision format and quantizing the others into low-precision format could alleviate the model accuracy drop compared to using the same low-precision format to all layers. So, given an FP16 LLM, the core problem for mixed-precision quantization is how to design an effective bit-width allocation scheme. In the quantization community, a notable disadvantage of mixed-precision quantization research is its hybrid and inefficient implementation on hardware systems due to using multiple bit width values instead of a shared bit width value. AWQ (Lin et al., MLSys 2024) additionally addresses this problem by searching activation-aware channel-wise factors to scale down sensitive weights, as we already discussed in our original paper submission.

---

> ### Author Response · Authors · 2025-11-24
> **Rebuttal to Official Review by Reviewer BEoY: 2/6**
>
> **(ii)** Same to Q-BERT, GOBO, SpQR and AWQ, **[1] and [2] also focus on mixed-precision weight-only quantization of LLMs**. For the bit width allocation across different layers, both [1] and [2] adopt a pioneering dynamic integer programming scheme originally proposed in ZeroQ (Cai et al., CVPR 2020), but use different metrics to estimate layer sensitivity. Specifically, given an FP16 LLM, [2] adopts the magnitude-based metric popularly used in existing works like Q-BERT and AWQ, and retains top 0.5% largest weights at each layer in FP16 while assigning {4, 3, 2} bit width values to the remaining weights at different layers via dynamic integer programming. **[2] neither provided nor identified any empirical observation at all** (please be kind to let us know any evidence place in the paper [2] if we missed it on this focus). **Compared to [2], [1] just considers a technically much easier problem**: assigning {16, 8, 4} bit width values to weights of different layers. [1] adopts three different metrics independently, namely correlation-based metric (CMPQ), pruning-based metric (PMPQ) and Taylor decomposition-based metric (TDMPQ), all of which do not consider the real quantization process in definition. More specifically, CMPQ uses popular canonical correlation analysis (CCA) to compute the layer sensitivity as the output correlation of each layer to the other layers: the lower the layer correlation, the higher the layer sensitivity, and vice versa. PMPQ first prunes a pre-defined sparsity ratio of weights in each layer separately, then computes the layer sensitivity as its accuracy drop over the original dense model: the higher the accuracy drop, the higher the layer sensitivity, and vice versa. TDMPQ uses the first-order Taylor approximation to compute the layer sensitivity as the loss function of adding small random weight perturbations to each layer separately: the lower the loss function, the lower the layer sensitivity, and vice versa. *Actually, in [1], only an old and very small model, OPT-350M with 24-layer, is used to briefly study empirical behaviors of CMPQ, PMPQ and TDMPQ which exhibit quite different empirical observations even for only OPT-350M* (please see Fig 2, Fig 3 and Fig 4 in [1], and in these three figures, 24 layer indices {layer_0, layer_1,…,layer_23} of OPT-350M are even not in the same order). Specifically, (1) for CMPQ in Fig 2, the third and sixth layers (layer_2 and layer_5) are most sensitive and the first layer (layer_0) is least sensitive; (2) for PMPQ in Fig 3, the results are very weird, all 24 layers have only two sensitivity values which are almost the same, and {layer_0, layer_1, layer_2, layer_3, layer_4, layer_6, layer_9, layer_13, layer_20, layer_21, layer_22, layer_23} have the same sensitivity value; (3) for TDMPQ in Fig 4, {layer_21, layer_23} are most sensitive. More importantly, according to our above clarifications on the definitions of CMPQ, PMPQ and TDMPQ for estimating layer sensitivity using FP16 LLMs (see Algorithm 1/2/3 in the paper of [1]), they are just proxy sensitivity metrics but have no direct relation to the real quantization process. That is, they do not measure the real quantization impact on model accuracy. **Now it is clear that [1] also has not explicitly and consistently identified** that the first and last layers of different pre-trained LLMs are most sensitive to quantization in terms of quantization impact on model accuracy (please be kind to let us know any evidence place in the paper [1] if we missed it on this focus). Additionally, to the best of our understanding, the experiments of [1] are seriously problematic, which we will clarify later in comparing the performance of our method and mixed-precision methods.

---

> ### Author Response · Authors · 2025-11-24
> **Rebuttal to Official Review by Reviewer BEoY: 3/6**
>
> **(iii)** Unlike existing mixed-precision quantization works including [1] and [2] which assume that different layers of LLMs are expected to exhibit different sensitivity to quantization (which is estimated but not defined in terms of real quantization error, as we clarifed above) and thus use multiple bit width values to reduce model accuracy drop, our work attempts to improve the predominant sequential post-training quantization framework for LLMs, in which different layers of any FP16 LLM are treated equally in the quantization process. We conjecture this may be not optimal in challenging bit-width settings. We start with a comprehensive empirical study via questioning in the real quantization process of state-of-the-art sequential quantization methods, whether for these methods, different layers of currently prevailing LLMs tend to exhibit similar quantization impacts on model accuracy. To explore this, we select three representative layer-wise, block-wise and multi-block-wise quantization methods, SmoothQuant (Xiao et al., ICML 2023), OmniQuant (Shao et al., ICLR 2024) and CBQ (Ding et al., ICLR 2025), and examine them on a lot of currently prevailing LLMs in the challenging 4-bit weight-activation quantization regime. Our empirical observations sufficiently illustrated by Fig 1 and Fig C in the main body and Appendix of our original paper submission, including (1) shallow/deep layers are usually more sensitive to quantization than intermediate layers, and (2) among shallow/deep layers, the most sensitive one is the first/last layer, which exhibits significantly larger quantization error than others, are novel and consistent over Llama-2, Llama-3 and Qwen-2.5 model families. They motivate us to design inter-layer sliding quantization and intra-layer sliding quantization, two core components of our SliderQuant method. These points were also partially discussed in the Introduction section (Line#44-104) of our original paper submission.
>
> **Part 2: About the novelty of our methodology**: the connections and differences of our method to your mentioned previously explored ideas, such as sliding-window quantization methods, mixed-precision quantization methods and extensions of the GPTQ’s framework concepts.
>
> **(i)** After reading our above responses, we sincerely hope you will no longer think our method combines the previously explored idea in mixed-precision quantization methods (e.g., [1] and [2]).
>
> **(ii)  Fixed-size sliding window is our baseline reference as we clearly clarified at multiple places in our original paper submission. Specifically**, in the Introduction section (Line#104-111) and the Method section (Line#182-194) of our original paper submission, we already discussed the connections and differences of our method to existing sliding-window quantization methods [3] (i.e., SKVQ, Duanmu et al., COLM 2024) and [4] (i.e., CBQ, Ding et al., ICLR 2025). Unlike [4] that focuses on both weight-only and weight-activation quantization of LLMs, [3] is tailored to KV cache quantization (a special quantization problem). Both of them use a fixed-size sliding window. With the fixed-size sliding quantization, shallow, intermediate and deep layers of a given FP16 LLM will be quantized with the same window size and moving interval per step. Thus, the fixed-size sliding quantization still has a large gap to attain our desired goals in light of our empirical observations -- two ingredients that are essential to formulate an improved sequential quantization framework: (1) the concentration of quantization process is required on shallow and deep layers, particularly the first and last layers; (2) the quantization synergy of successive layers is required to reduce quantization errors across layers. We fill this gap by presenting two novel sliding quantization components illustrated in Fig 2 of our original paper submission. Our base component, inter-layer sliding quantization, incorporates three types of sliding window designs tailored for adaptively quantizing shallow, intermediate and deep layers with a smart optimization relay across them. Specifically, it first allocates a progressively expanded sliding window along shallow layers, a fixed-size sliding window along intermediate layers and a progressively contracted sliding window along deep layers, and then performs the sliding quantization progressively. With three types of sliding window designs, our inter-layer sliding quantization component can leverage our identified empirical observations about the varying layer sensitivity to quantization. To exploit these empirical observations further, we present another complementary component called intra-layer sliding quantization. It extends the progressively expanded sliding design within each window of inter-layer sliding quantization component, by which all layers in each window are jointly quantized in an incremental manner. Coupling these two components in this way forms a neat implementation of SliderQuant.

---

> ### Author Response · Authors · 2025-11-24
> **Rebuttal to Official Review by Reviewer BEoY: 4/6**
>
> **(iii)** As we discussed in the Introduction section (Line# 45-49) and the Related Work section (Line# 140-145) of our original paper submission, GPTQ uses a layer-wise weight-only quantization framework based on approximate second-order Hessian matrices to reduce model accuracy drop. In GPTQ and its extensions such as QuIP (Chee et al., NeurIPS 2024), there is no concept of quantization synergy across successive layers due to the layer-wise quantization framework while our method has this concept by its inter-layer sliding quantization component and intra-layer sliding quantization component.
>
> **(iv)** Our method outperforms the methods of all these three types discussed above with clear margins, as can be clearly seen from experiments (Table 1 to Table 3) in our original paper submission and extra experiments provided in this rebuttal (please refer to our responses to your question about comparing the performance of our method and mixed-precision methods).
>
> **(v)** With the above detailed comparisons, connection and difference clarifications in motivation, design and performance of our method and other three types of methods mentioned by you, it should be clear the novelty of our method is solid rather than incremental.
>
> **2. To your comments about the second weakness** “Some experimental results have limited practical significance. For instance, in Table 2…under W4A4 quantization, and…in Table 5 under W2A16…raising doubts about the meaningfulness of these ultra-low-bit experiments.”
>
> **Our responses**: **(1)** Table 2 provides the results comparison on the zero-shot commonsense reasoning tasks, in which we compare our method with state-of-the-art post-training quantization methods **which do not modify LLM architectures and thus do not introduce extra inference-time costs**. Under this setting, your mentioned “more than 10% accuracy degradation under W4A4 quantization on almost all the benchmarks” is true but is the new state-of-the-art. If you look at the results comparison in Table 3, you can find such a more than 10% accuracy degradation under W4A4 quantization on all zero-shot commonsense reasoning benchmarks will be significantly reduced to less than 1.5% on different LLMs with our method using rotation transformations, e.g., the averaged accuracy degradation is only 0.47% on Llama2-13B. In Table 3, we compare our method with state-of-the-art post-training quantization methods including rotation transformation based methods and others **which modify LLM architectures and thus introduce extra inference-time costs**, again our method achieves the best results. We clarified the organization of these experiments and counterpart methods in the Experiments section (Line#293-323) of our original paper submission, which intends to have more fair comparisons; **(2)** Table 5 provides the experimental results to explore the performance of our method on five challenging math and code tasks with the state-of-the-art DeepSeek-R1 distilled models. It is true that our method leads to still large accuracy degradation under the challenging W2A16 quantization, but its performance is significantly better than existing methods like OmniQuant across all math and code tasks and model scales, e.g., showing 23.09% and 26.54% accuracy margins on DeepSeek-R1-Distill-Qwen-14B and DeepSeek-R1-Distill-Qwen-32B, respectively. As math and code tasks are rarely explored in the post-training quantization community, our experiments also reveal that more research attention is needed toward challenging real-world applications.

---

> ### Author Response · Authors · 2025-11-24
> **Rebuttal to Official Review by Reviewer BEoY: 5/6**
>
> **3. To your first question** “Could the authors provide more theoretical or intuitive explanations for the observations that the first/last layers are much more sensitive to quantization, beyond the empirical observations? I believe these could provide insights to future studies and add more novelty to this paper.”
>
> **Our responses**: regarding our most critical empirical observation that the first and last layers are much more sensitive to quantization than the other layers of LLMs, here we provide theoretically intuitive explanations from a perspective of their local and global roles. **The first and last layers are the most important in terms of their local roles in feature extraction**. Specifically, the first layer is responsible for the very basic feature extraction, which has a strong correlation to all succeeding layers because of the forward propagation and the backward propagation at the model learning phase. The last layer is responsible for the final feature abstraction, which generates the most discriminative feature input for final predication supervised with task labels. We think this is the main reason why the first and last layers are the most sensitive to quantization, as we observed consistent over Llama-2, Llama-3 and Qwen-2.5 model families when using layer-wise, block-wise and multi-block-wise quantization methods in the challenging 4-bit weight-activation quantization regime. **The first and last layers are the most important in terms of their global roles in quantization**. Specifically, the quantization error from the first layer will be propagated to all succeeding layers due to the dominant sequential quantization framework, while has a risk to be magnified gradually when quantizing succeeding layers sequentially. Also because of the dominant sequential quantization framework, the last layer inherits the accumulated error from all preceding layers, which makes the quantization of the last layer the most difficult compared to the other layers. We think this is another underlying reason why the first and last layers are the most sensitive to quantization.
>
> **4. To your second question** “While the bitwidth is set to be the same for all layers in SliderQuant, mixed-precision quantization is also a popular way for LLM quantization that could utilize the characteristics that different layers have different sensitivity to quantization (e.g. [2]). Could the authors provide some comparison with such kind of methods, to show whether the sliding-layer mechanism could complement or outperform mixed-precision strategies?”

---

> ### Author Response · Authors · 2025-11-24
> **Rebuttal to Official Review by Reviewer BEoY: 6/6**
>
> **Our responses**: following you constructive suggestion, in the below Table A and Table B, we provide thorough performance comparisons of our SliderQuant with state-of-the-art mixed-precision quantization methods including LLM-MQ [2] mentioned by you and SpQR (Dettmers et al., ICLR 2024) for weight-only quantization, and QUIK (Ashkboos et al., EMNLP 2024) for weight-activation quantization. All results for LLM-MQ, SpQR and QUIK are collected from their original papers. We can see that: **(1)** our SliderQuant under lower bit width quantization W4A16/W3A16/W2A16 even outperforms LLM-MQ [2] under higher bit width quantization W4A16 (0.5% in FP16)/W3.4A16 (0.5% in FP16)/W2.2A16 (0.5% in FP16) by significant margins, and our SliderQuant with W4A4 also outperforms QUIK with W4.9A4 (5% in FP16), both on language generation and commonsense reasoning benchmarks; **(2)** similarly, our SliderQuant also gets significantly performance gains compared to SpQR; **(3)** the performance gain of our SliderQuant against LLM-MQ [2] or QUIK is further pronounced when retaining a smaller ratio of weights in FP16 or using rotation transformations. These experimental results sufficiently show that our sliding-layer mechanism could outperform and complement mixed-precision quantization methods.
>
> **Table A**: Results comparison of SliderQuant with mixed-precision quantization methods on Llama2-13B. In the table, (1) for LLM-MQ/QUIK, “0.5% in FP16”/“5% in FP16” indicates the corresponding ratio of weight/weight-activation outliers at each layer stored in FP16; (2) for 2.1-bit weight quantization by our SliderQuant, we simply use 4-bit quantization to the first and last layers while quantizing all other layers to 2-bit, and SliderQuant+ denotes our best version of SliderQuant using rotation transformations. Best results are bolded.
>
> |#Bits|Method|Wikitext2↓|PIQA↑|ARC-e↑|HellaSwag↑|Winogrande↑|Avg↑|
> |-|:-:|:-:|:-:|:-:|:-:|:-:|:-:|
> |W16A16|-|4.88|80.41|77.40|79.37|72.14|77.33|
> |W4A16 (0.5% in FP16)|LLM-MQ[2]|8.03|79.49|58.50|76.31|69.30|70.90|
> |W4A16|SliderQuant|**5.00**|**80.41**|**76.73**|**78.30**|**72.14**|**76.90**|
> |W3.4A16 (0.5% in FP16)|LLM-MQ[2]|8.61|79.49|58.12|74.77|69.61|70.50|
> |W3A16|SliderQuant|**5.27**|**79.11**|**74.16**|**76.78**|**69.76**|**74.95**|
> |W2.2A16 (0.5% in FP16)|LLM-MQ[2]|10.80|76.77|55.26|70.83|67.09|67.49|
> |W2.1A16|SliderQuant|**7.64**|**77.21**|**62.58**|**71.05**|**67.51**|**69.59**|
> |W2A16 (0.5% in FP16)|LLM-MQ[2]|12.17|75.84|54.29|68.32|65.51|65.99|
> |W2A16|SliderQuant|7.71|73.56|67.47|64.75|63.22|67.25|
> |W2A16 (0.5% in FP16)|SliderQuant|**7.64**|**76.13**|**67.51**|**69.25**|**66.12**| **69.75**|
> |W4.9A4 (5% in FP16)|QUIK|5.28|79.22|74.92|**78.36**|71.90|76.10|
> |W4A4|SliderQuant+|**5.07**|**79.96**|**77.27**|77.96|**71.98**|**76.79**|
>
> **Table B**: Results comparison of SliderQuant with mixed-precision quantization methods on Llama-7B and Llama-13B in the Llama family. Best results are bolded.
>
> |Model|#Bits|Method|Wikitext2↓|C4↓|PIQA↑|ARC-e↑|Arc-c↑|HellaSwag↑|Winogrande↑|Avg↑|
> |-|:-:|:-:|:-:|:-:|:-:|:-:|:-:|:-:|:-:|:-:|
> |Llama-7B|W16A16|-|5.68|7.08|79.43|73.15|45.05|76.16|70.24|68.81|
> |Llama-7B|W3.45A16|SpQR|5.87|7.28|78.13|65.87|38.05|55.27|67.48|60.96|
> |Llama-7B|W3A16|SliderQuant|**5.82**|**7.13**|**77.42**|**69.70**|**40.36**| **71.74**|**67.96**|**65.44**|
> |Llama-13B|W16A16|-|5.09|5.62|80.41|74.71|47.95|79.08|73.09|71.05|
> |Llama-13B|W3.45A16|SpQR|5.22|6.72|78.73|73.27|42.75|58.22|68.90|64.37|
> |Llama-13B|W3A16|SliderQuant|**5.21**|**6.78**|**79.33**|**73.15**|**45.73**|**76.19**|**70.72**|**69.02**|
>
> Additionally, we politely point out that the experiments of [1] are seriously problematic which makes the comparison of our method with it impossible, to the best of our understanding. In the experiments of [1], the authors separately use the aforementioned three layer sensitivity metrics to old-fashioned small Bert and OPT models with the largest model size 2.7B, and adopt the FP16 model as the baseline, but mistakenly report all FP16 baseline models lead to serious accuracy drop (10 or 12 in perplexity) on Wikitext, without providing baseline perplexity results, and also without considering any existing mixed-precision quantization methods, popular large LLMs, and zero-shot commonsense reasoning tasks, totally ignoring de-facto benchmark protocols in the LLM quantization community.
>
> **Finally**, based on the constructive comments by you and the other three reviewers and our responses, **we carefully revised and updated the manuscript of our work**. Regarding more experiments and discussions that we made during the rebuttal phase, you are referred to our responses to the other three reviewers, and the revised manuscript.

---

> > ### Comment · Reviewer_BEoY · 2025-11-27
> >
> > Thanks for your response. The additional experimental results and explanations could address my concerns a lot, and I decide to increase my rating accordingly.

---

> ### Author Response · Authors · 2025-11-28
> **Thank You for the Recognition of Our Rebuttal and Work**
>
> Dear Reviewer BEoY,
>
> We are glad to see that you are satisfied with our rebuttal and have increased your score to 6. We will continue to improve the quality of our work.
>
> Thanks again for your constructive comments, time and patience.
>
> The authors.

---

### Official Review · Reviewer_DdkN · 2025-10-29

**Soundness:** 3
**Presentation:** 4
**Contribution:** 3
**Rating:** 4
**Confidence:** 4

**Summary:**

This paper discovers the distinct quantization sensitivity among model layers, and proposes SliderQuant, a novel learnable quantization framework which contains two level sliding quantization concept. Experiments indicate that this method becomes SOTA sliding-based PTQ method.

**Strengths:**

1. The authors identify varying sensitivities of different layers to quantization and improve the quantization performance of layers with different sensitivities through a sliding-window design, rather than directly adopting a mixed-precision approach. This provides a novel and interesting perspective.
2. The writing is clear and well-structured, the experiments are thorough, and the figures and tables are elegantly designed.

**Weaknesses:**

1. The description of intra-layer sliding quantization is the main weakness of the paper. As one of the core innovations, its explanation is too brief, which makes it confusing. Does it mean that the weights/activation matrices are also partitioned and quantized sequentially within each layer?
2. I'm afraid that whether the effectiveness of both learnable low-rank matrices A and B will be influenced after quantization because they have been integrated into weights before quantization during inference.
3. The authors ignores to describe the training details, such as loss function, supervision information. Is it followed by OmniQuant?
4. The authors ignores to provide the memory usage and runtime of SliderQuant. As described, SliderQuant requires loading multiple blocks into the GPU simultaneously, which can lead to substantial memory overhead. Based on prior experience, training a 7B LLM with OmniQuant takes approximately 0.8 hours and 15 GB of memory on an A100 GPU, while SliderQuant is likely to demand even more.

**Questions:**

1. More detailed description about intra-layer sliding quantization is needed (W1).
2. More training details should be provided (W3).
3. Although the appendix mentions that the training can be completed on a single A6000 GPU, providing explicit comparative data (e.g., against OmniQuant, CBQ, and the more efficient GPTQ) would make the analysis more intuitive. If the training cost is excessively high, the practicality of SliderQuant may be questioned.(W4).

I promise to increase my score if all my concerns are addressed.

---

> ### Author Response · Authors · 2025-11-24
> **Rebuttal to Official Review by Reviewer DdkN: 1/3**
>
> Thanks for your constructive review and the recognition of the empirical observations, the motivation, the novelty, the experiments and the writing of our work. Next, we address your concerns and questions one by one.
>
> **1. To your comments about the main weakness and the first question** “The description of intra-layer sliding quantization is the main weakness of the paper. As one of the core innovations, its explanation is too brief, which makes it confusing. Does it mean that the weights/activation matrices are also partitioned and quantized sequentially within each layer?” and “More detailed description about intra-layer sliding quantization is needed (W1).”
>
> **Our responses**: Thanks for your careful comments and questions. In our SliderQuant, intra-layer sliding quantization is applied into the current sliding window of inter-layer sliding quantization. Just as you said, for intra-layer sliding quantization, the weight and activation matrices at each layer in the current sliding window of inter-layer sliding quantization are also partitioned with a ratio $\gamma$ and quantized incrementally, resembling our progressively expanded sliding window design for shallow layers. With the default setting of $L_s=4, s=2, L_d=4, \gamma=0.5$, the current sliding window of inter-layer sliding quantization has at most 4 layers and at least 1 layer. Let the current sliding window of inter-layer sliding quantization has $s_c$ layers, $1\leq s_c \leq 4$, then with intra-layer sliding quantization, the joint quantization of these $s_c$ layers will be completed incrementally in $N=2$ stages as $N=1/\gamma$. In the first stage, the first half of weight/activation matrices in these $s_c$ layers is quantized jointly. In the second stage, the whole of weight/activation matrices (including the first half previously quantized in the first stage) in these $s_c$ layers is quantized jointly. Therefore, we can see that intra-layer sliding quantization uses **a fine-grained variant of our progressively expanded sliding window** performing incremental quantization in $N=1/\gamma$ stages with a stage-wise expanded ratio of $\gamma$, building a local to global parameter synergy across $s_c$ layers within the current sliding window of inter-layer sliding quantization to suppress quantization error. We briefly clarified and illustrated this working mechanism of intra-layer sliding quantization in Line#249 to 250 and the right part of Figure 2 in our original paper submission. We are sorry for bringing any unclear description and confusion to you. In the revised paper, we add extra clarifications to have a clearer and detailed description of intra-layer sliding quantization.
>
> **2. To your comments about the second weakness** “I'm afraid that whether the effectiveness of both learnable low-rank matrices A and B will be influenced after quantization because they have been integrated into weights before quantization during inference.”
>
> **Our responses**: the effectiveness of both learnable low-rank matrices $A$ and $B$ **will not be influenced after quantization** for inference. Taking weight-only quantization as an example, during training, for the current sliding window of our SliderQuant, the iteratively learned low-rank matrices $A$ and $B$ via minimizing a window-wise reconstruction loss function (which will be clarified in our responses to your next question) on a small number of calibration samples (e.g., 128) are merged into the original weight matrices $W$ by $\tilde{W} = W + AB$ layer by layer first, and then quantization is performed on the merged weight matrices $\tilde{W}$, exactly following the real deployment workflow. Once all layers of an FP16 LLM are quantized with our SliderQuant, the resulting quantized model is used for direct evaluation by real-time inference. That is, all results reported in our paper are obtained in this manner. **This *merge-then-quantize* pipeline during training ensures that quantization does not affect the effectiveness of learned low-rank matrices $A$ and $B$**, which is also the advantage of popular low-rank adaptation as we clarified in our original paper submission.

---

> ### Author Response · Authors · 2025-11-24
> **Rebuttal to Official Review by Reviewer DdkN: 2/3**
>
> **3. To your comments about the third weakness and the second question** “The authors ignores to describe the training details, such as loss function, supervision information. Is it followed by OmniQuant?” and “More training details should be provided (W3).”
>
> **Our responses**: in our SliderQuant, we also adopt **the reconstruction-based optimization strategy** predominantly used in most existing PTQ methods for LLMs such as GPTQ, SmoothQuant, OmniQuant, AWQ, QLLM, QuaRot, SpinQuant, FlatQuant, etc., which are thoroughly compared in our experiments. Unlike them, since our SliderQuant performs sequential sliding-window-based quantization with two levels of sliding quantization concepts (i.e., inter-layer sliding quantization and intra-layer sliding quantization),  **our loss function is defined window by window**. Specifically, taking weight-only quantization as an example, for quantizing the current sliding window containing $s$ layers, its loss function is a mean square error defined as:
>
> $\underset{\hat{W}} {argmin} \||F(W, X) - F(\hat{W}, X)\||_2^2$,  where $\hat{W}=quantizer(W+AB)$,        (1)
>
> which forces the output $F(\hat{W}, X)$ from the current sliding window of the quantized model to approximate the output $F(W, X)$ from the current sliding window of the FP16 model conditioned on the same FP16 input $X$. Here, $W$/$\hat{W}$ denotes the FP16/quantized weight matrices for $s$ layers in the current sliding window, and $\hat{W}$ is obtained by applying a pre-defined uniform quantizer to the refined FP16 weight matrices $W+AB$, where $A$ and $B$ are learnable low-rank matrices for $s$ layers. **For brevity, here we omit subscripts for different weight matrices at $s$ layers in the current sliding window**. That is, when minimizing this sliding-window-based loss function in SliderQuant, the output from the current sliding window (or say, the last layer output of the current sliding window) of the FP16 model is always used as only supervision information. For weight-activation quantization, its loss function to be minimized is defined by simply replacing $F(\hat{W}, X)$ in the above loss function by $F(\hat{W}, \hat{X})$:
>
> $\underset{\hat{W},\hat{X}} {argmin} \||F(W, X) - F(\hat{W}, \hat{X})\||_2^2 $, where $\hat{W}=quantizer(W \odot \alpha+AB)$ and $\hat{X}=quantizer(X \oslash \alpha)$,        (2)
>
> where $\alpha$ denotes learnable channel-wise scaling vectors to scale the original FP16 input $X$ and reversely scale the original FP16 weight matrices $W$ for $s$ layers in the current sliding window, $\hat{W}$ and $\hat{X}$ are obtained by applying a pre-defined uniform quantizer to the refined FP16 weight matrices $W \odot \alpha+AB$ and the scaled FP16 input $X \oslash \alpha$, respectively.
>
> In our original paper submission, we separately defined the above loss function both for weight-only and weight-activation quantization tasks in the Sub-section 3.1 (see Line#182 to 194 and Eq.1), and defined the above quantization process with learnable $A$, $B$ and $\alpha$ in the Sub-section 3.1 (see Line#262 to 273 and Eq.2), and put the definition of the uniform quantizer and the training details in the Appendix (see the Section B and the Section C). We are sorry for bringing any unclear description and confusion to you. In the revised paper, we add extra clarifications to have a clearer and detailed definition of loss function and supervision signal in our SliderQuant.
>
> **4. To your comments about the fourth weakness and the third question** “The authors ignores to provide the memory usage and runtime of SliderQuant…is likely to demand even more.” and “Although the appendix mentions…providing explicit comparative data (e.g., against OmniQuant, CBQ, and the more efficient GPTQ) would make the analysis more intuitive…the practicality of SliderQuant may be questioned.(W4)”.

---

> ### Author Response · Authors · 2025-11-24
> **Rebuttal to Official Review by Reviewer DdkN: 3/3**
>
> **Our responses**: Following your constructive suggestion, we conducted an ablation to compare the training time and the memory overhead of our SliderQuant (including our default version and a fast version), the fixed-size sliding quantization baseline (CBQ, which is closely related to our SliderQuant) and OmniQuant on both Llama2-7B and Qwen2.5-14B under W4A4 quantization. In the experiments, we always use the same number of calibration samples (128), the same batch size (1) and the same number of training epochs (20) for all methods unless otherwise stated. Note GPTQ is known as an efficient PTQ method but is tailored to weight-only quantization and is less related to our method, so it is not compared here. Detailed results are summarized in the below Table A, from which we can see that, on both Llama2-7B and Qwen2.5-14B under W4A4 quantization: **(1)** in terms of training time, our SliderQuant (with the default setting of $L_s=4, s=2, L_d=4, \gamma=0.5$) is $1.08\times$ to $1.55\times$ slower than OmniQuant and the fixed-sized sliding quantization (CBQ) with the window size of $s=2$, but is faster than the fixed-sized sliding quantization with the window size of $s=4$ or $s=3$, and our SliderQuant-Fast (the only change is the number of training epochs is 10 instead of 20) is the fastest ($1.21\times$ to $1.78\times$ faster than OmniQuant and the fixed-sized sliding quantization (CBQ) with the window size of $s=2$) and achieves slightly worse quantization results compared to SliderQuant; **(2)** In terms of memory overhead, our SliderQuant, SliderQuant-Fast and the fixed-sized sliding quantization (CBQ) with the window size of $s=4$ have the same memory usage, which is around $2\times$ compared to the memory usage by the fixed-sized sliding quantization (CBQ) with the window size of $s=2$ and OmniQuant; **(3)** in terms of model accuracy, both SliderQuant and SliderQuant-Fast always achieve significantly better perplexity than other counterpart methods, showing perplexity reductions ranging from 2.21 to 23.70 on Wikitext2 and 1.93 to 41.15 on C4. **Additionally**, Figure B in the Appendix of our original paper submission presents a comprehensive view of SliderQuant's performance-efficiency trade-offs across different training time configurations. These results demonstrate that SliderQuant can achieve even greater training efficiency, showcasing up to $6.42\times$ speedup compared to OmniQuant on Llama2-7B while maintaining better perplexity on both Wikitext2 and C4. Summarily, these results show that the training cost of SliderQuant is decent, which guarantees the practicality of our method to quantize different LLMs, also thanks to the simplicity of our method.
>
> **Table A**: A comparison of the training time and the memory overhead of our SliderQuant, fixed-size sliding quantization baseline (CBQ) and OmniQuant on Llama2-7B and Qwen2.5-14B under W4A4 quantization. All experiments are conducted on a single NVIDIA A6000-48GB GPU, and we set the number of calibration samples to 128 (a popular choice in the quantization community), the batch size to 1, and the number of training epochs to 20 for all methods except our SliderQuant-Fast with 10 training epochs. Best results are bolded. **Note**: in the paper of [OmniQuant](https://openreview.net/pdf?id=8Wuvhh0LYW) (its Table A12), quantizing Llama-7B on a single NVIDIA A100-80G GPU with the same setting as ours takes 1.6 hours.
>
> |Model|Method|Wikitext2↓|C4↓|Training Time (GPU Hours)|Memory Overhead (GB)|
> |-|-|:-:|:-:|:-:|:-:|
> |Llama2-7B|OmniQuant |14.26|18.02|4.75|**15.04**|
> |Llama2-7B|Fixed-size sliding (s=2, CBQ) |12.73|14.45|5.66|16.24|
> |Llama2-7B|Fixed-size sliding (s=3, CBQ) |11.18|13.94|7.65|23.02|
> |Llama2-7B|Fixed-size sliding (s=4, CBQ) |11.13|13.52|9.16|29.80|
> |Llama2-7B|SliderQuant-Fast|8.92|11.59|**3.24**|29.80|
> |Llama2-7B|SliderQuant|**8.34**|**11.10**|6.14|29.80|
> |Qwen2.5-14B|OmniQuant|34.70|61.75|**7.38**|22.12|
> |Qwen2.5-14B|Fixed-size sliding (s=2, CBQ) |17.41|25.20|10.83|**21.12**|
> |Qwen2.5-14B|Fixed-size sliding (s=3, CBQ) |15.94|24.15|13.56|30.23|
> |Qwen2.5-14B|Fixed-size sliding (s=4, CBQ) |14.75|21.41|15.36|43.71|
> |Qwen2.5-14B|SliderQuant-Fast|12.19|17.53|**6.08**|43.71|
> |Qwen2.5-14B|SliderQuant|**11.00**|**16.60**|11.43|43.71|
>
> **Finally**, based on the constructive comments by you and the other three reviewers and our responses, **we carefully revised and updated the manuscript of our work**. Regarding more experiments and discussions that we made during the rebuttal phase, you are referred to our responses to the other three reviewers, and the revised manuscript.

---

> > ### Comment · Reviewer_DdkN · 2025-11-27
> > **Authors provide a detailed rebuttal which completely address my concerns**
> >
> > Dear Authors,
> >
> > Thanks for your detailed rebuttal. Your answers well address my concerns. I have read the new version PDF which has included all the required revisions. However, from the results we get quantizing an LLM with 7B parameters requires nearly 30GB memory, which is really costly. For comparison, GPTQ only takes up 5GB. Please point out this limitation in your PDF.
> >
> > I have modified my score accordingly.
> >
> > Best,
> >
> > Reviewer DdkN

---

> ### Author Response · Authors · 2025-11-28
> **Thank You for the Recognition of Our Rebuttal and Work**
>
> Dear Reviewer DdkN,
>
> We are glad to see that you are satisfied with our rebuttal and have increased your score to 6. Indeed, during the calibration training, just like other sliding-window-based PTQ methods, our SliderQuant also requires much more memory cost compared to the currently most efficient layer-wise PTQ method GPTQ tailored for weight-only quantization. **Following your suggestion, we have pointed out this limitation in the Method section (see Line#201-203) of our newly updated manuscript PDF**. According to the thorough experimental results shown in Table 1-5, our method achieves state-of-the-art results for both weight-only quantization and weight-activation quantization on various tasks with Llama-2/Llama-3/Qwen-2.5 model families, DeepSeek-R1 distilled models and large MoE models under different bit width settings, and demonstrates significantly better performance than GPTQ and its variants like QuIP which mostly fail in 2-bit weight quantization and cannot handle weight-activation quantization. These performance advantages and the direct deployment of our quantized models for real-time inference make much heavier memory cost of our method compared to GPTQ worthwhile. We will continue to improve the quality of our work.
>
> Thanks again for your constructive comments, time and patience.
>
> The authors.

---

### Official Review · Reviewer_R2bM · 2025-10-29

**Soundness:** 3
**Presentation:** 3
**Contribution:** 3
**Rating:** 6
**Confidence:** 3

**Summary:**

The paper proposes SliderQuant, a quantization framework. It introduces an adaptive sliding-layer strategy that assigns progressively expanded, fixed, and contracted windows to shallow, middle, and deep layers, respectively, addressing the unequal quantization sensitivity across layers—higher at the top and bottom. The method also includes intra-layer sliding quantization to further improve performance. Experiments show that SliderQuant achieves lower perplexity and higher reasoning accuracy than existing PTQ baselines, especially under challenging low-bit configurations.

**Strengths:**

1. Clear and strong motivation: The paper is motivated by an empirically grounded observation on layer-wise sensitivity to quantization in LLMs. The motivation is clearly presented and addresses an overlooked aspect in post-training quantization.

2. Comprehensive experiments: The evaluation covers multiple model families and various bit-width settings, demonstrating the generality of the proposed framework.

3. Intuitive and well-written method: The proposed sliding-layer quantization framework is easy to follow and clearly described, with both figures and ablation studies supporting the core design choices. The overall paper is well written and organized.

**Weaknesses:**

1. Uneven optimization frequency of middle layers: According to Figure 1 and the default hyperparameter setting, the 4th and 5th layers appear to be quantized only once. This means that some middle layers receive fewer optimization passes than their neighbors. Could this uneven optimization frequency introduce instability or suboptimal performance? In particular, when the middle-layer window size is larger than two, how do you ensure that all middle layers are optimized an equal number of times?

2. Training cost comparison with baselines is unclear: Although Table 7 usefully extends the baseline window size from (s=2) to (s=4), the training (quantization) cost of SliderQuant versus the baselines remains unclear. Please report quantitative efficiency metrics such as GPU hours. Ideally, a main table could compare performance under equal training budgets to ensure fairness.

3. Hyperparameter generalization is under-discussed: The paper introduces several hyperparameters, but the discussion of their sensitivity is limited to a single model (LLaMA-2-7B). It remains unclear how these settings generalize across models of different sizes and depths. Please analyze or at least discuss how to scale or tune these parameters for larger models.

**Questions:**

1. Missing baseline results in Table 1: Some baselines in Table 1 have missing or abnormal values. Could the authors clarify the cause?

2. Source of improvement: sliding vs. repeated optimization: The proposed method combines a sliding-window design and more times of optimization of the first and last layers. Which factor contributes more to the final improvement? An additional ablation that isolates “adaptive sliding” from “frequency of re-quantization” would help clarify the main source of gain.

---

> ### Author Response · Authors · 2025-11-24
> **Rebuttal to Official Review by Reviewer R2bM: 1/3**
>
> Thanks for your constructive review and the recognition of the motivation, the core designs, the compressive experiments and the writing of our work. Next, we address your concerns and questions one by one.
>
> **1. To your comments about the first weakness** “Uneven optimization frequency of middle layers: According to Figure 1…how do you ensure that all middle layers are optimized an equal number of times?”
>
> **Our responses**: we think your mentioned “Figure 1” should be “Figure 2” which illustrates the overall design of our SliderQuant. Actually, under the default hyperparameter setting, fixed-size sliding quantization at middle layers has the widow size of {$s=2$}, and we use an even optimization frequency instead of an uneven optimization frequency. Specifically, we set one overlapped layer between shallow layers and middle layers, and also set one overlapped layer between middle layers and deep layers. That is, in Figure 2, your mentioned 4th layer is the last layer of shallow layers and is also the first layer of middle layers, and the first layer of deep layers is also the last layer of middle layers. **Actually, we clarified this in the paragraph titled** “Inter-Layer Sliding Quantization” (see Line#242 to 244) of our original paper submission. When the window size of fixed-size sliding quantization at middle layers is larger than 2 (i.e., $s>2$), we can easily ensure an even optimization frequency by changing the number of overlapped layers between middle layers and shallow/deep layers, e.g., 2 overlapped layers when $s=3$. We are sorry for this confusion. In the updated manuscript, we revised Figure 2 and its caption to remove this confusion.
>
> **2. To your comments about the second weakness** “Training cost comparison…Although Table 7 usefully extends the baseline window size from ($s=2$) to ($s=4$), the training (quantization) cost of SliderQuant versus the baselines remains unclear. Please report…GPU hours. Ideally…under equal training budgets to ensure fairness.”
>
> **Our responses**: we think your mentioned “Table 7” should be “Table 9” which studies the baseline fixed-size sliding quantization by changing its window size from ($s=2$) to ($s=4$). Following your constructive suggestion, we conducted an ablation to compare the training time and the memory overhead of our SliderQuant and the fixed-size sliding quantization baseline, using the same number of calibration samples (128), the same batch size (1) the same number of training epochs (20) as in Table 9. Detailed results are summarized in the below Table A, from which we can see that, on both Llama2-7B and Qwen2.5-14B under W4A4 quantization: **(1)** compared to the fixed-sized sliding quantization with the window size of $s=4$ applied to all layers of each FP16 LLM, our SliderQuant with the default setting of $L_s=4, s=2, L_d=4, \gamma=0.5$ needs less training time and the same memory overhead, and achieves significantly better quantization results; **(2)** the fixed-sized sliding quantization with the window size of $s=2$ is more efficient in terms of the training time and the memory overhead compared to the fixed-sized sliding quantization with the window size of $s=4$/$s=3$ and our SliderQuant, but shows much worse quantization results; **(3)** SliderQuant-Fast (the only change over SliderQuant is the number of calibration epochs is 10 instead of 20) needs the least training time and achieves slightly worse quantization results compared to SliderQuant, but its memory overhead keeps the same to SliderQuant. Summarily, these results validate the training efficiency of SliderQuant, which guarantees the practicality of our method to quantize different LLMs, also thanks to the simplicity of our method.
>
> **Table A**: A comparison of the training time and the memory overhead of our SliderQuant and the fixed-size sliding quantization baseline on Llama2-7B and Qwen2.5-14B under W4A4 quantization. All experiments are conducted on a single NVIDIA A6000-48GB GPU, and we set the number of calibration samples to 128 (a popular choice in the quantization community), the batch size to 1, and the number of training epochs to 20 for all methods except our SliderQuant-Fast with 10 training epochs. Best results are bolded.
>
> |Model|Method|Wikitext2↓|C4↓|Training Time (GPU Hours)|Memory Overhead (GB)|
> |-|-|:-:|:-:|:-:|:-:|
> |Llama2-7B|Fixed-size sliding (s=2)|12.73|14.45|5.66|**16.24**|
> |Llama2-7B|Fixed-size sliding (s=3)|11.18|13.94|7.65|23.02|
> |Llama2-7B|Fixed-size sliding (s=4)|11.13|13.52|9.16|29.80|
> |Llama2-7B|SliderQuant-Fast|8.92|11.59|**3.24**|29.80|
> |Llama2-7B|SliderQuant|**8.34**|**11.10**|6.14|29.80|
> |Qwen2.5-14B|Fixed-size sliding (s=2)|17.41|25.20|10.83|**21.12**|
> |Qwen2.5-14B|Fixed-size sliding (s=3)|15.94|24.15|13.56 |30.23|
> |Qwen2.5-14B|Fixed-size sliding (s=4)|14.75|21.41|15.36|43.71|
> |Qwen2.5-14B|SliderQuant-Fast|12.19|17.53|**6.08**|43.71|
> |Qwen2.5-14B|SliderQuant|**11.00**|**16.60**|11.43|43.71|

---

> ### Author Response · Authors · 2025-11-24
> **Rebuttal to Official Review by Reviewer R2bM: 2/3**
>
> **3. To your comments about the third weakness** “Hyperparameter generalization is under-discussed: The paper introduces several hyperparameters, but the discussion…is limited to a single model (Llama2-7B)…Please…at least discuss how to scale or tune these parameters for larger models.”
>
> **Our responses**: as you said, our SliderQuant has a sliding quantization schedule consisting of four hyperparameters used in a progressively expanded sliding window (PESW) for $L_s$ shallow layers, a fixed-size sliding window with the size of $s$ for $L_i$ intermediate layers and a progressively contracted sliding window (PCSW) for $L_d$ deep layers used in inter-layer sliding quantization (Inter-S), and an incremental quantization ratio $\gamma$ for intra-layer sliding quantization (Intra-S) within each window of Inter-S. Indeed, in our original paper submission, we only used Llama2-7B to study their choices from the perspectives of their individual roles and combined roles, and chose $L_s=4, s=2, L_d=4, \gamma=0.5$ as the default hyperparameter setting of our SliderQuant applied to all 9 different LLMs tested in our paper, for a simple implementation. Also, this default setting is not optimal even for Llama2-7B, let alone other LLMs. To better compare their choices, we additionally conducted multiple sets of ablative experiments with larger and recently released Qwen2.5-14B under W4A4 quantization. Detailed results are summarized in the below Table B, Table C, Table D and Table E, from which we can see that, in our SliderQuant: **(1)** with our default hyperparameter setting, the proposed sliding quantization designs are complementary to each other (see Table B); **(2)** for each individual sliding quantization design, its default hyperparameter setting is usually not the best for both Llama2-7B and Qwen2.5-14B (see Table C and Table E); **(3)** combining it with PESW and PCSW is significantly better than merely using fixed-size sliding quantization to all layers of each FP16 LLM, both on Llama2-7B and Qwen2.5-14B (see Table D). Although our SliderQuant with the default hyperparameter setting already achieves superior results to existing PTQ methods, these ablations indicate that there is still room to get improved quantization performance of our SliderQuant on different LLMs by choosing an LLM-specific hyperparameter setting.
>
>  **Table B**: Ablation of combining different designs in inter-layer sliding quantization (Inter-S) and intra-layer sliding quantization (Intra-S). In the experiments, we use the default hyperparameter setting of $L_s=4, s=2, L_d=4, \gamma=0.5$. Best results are bolded.
>
> |Model|PESW|PCSW|Intra-S|Wikitext2↓|C4↓|
> |-|:-:|:-:|:-:|:-:|:-:|
> |Llama2-7B||| |12.73|14.45|
> |Llama2-7B|||√|10.34|13.46|
> |Llama2-7B|√|||10.30|13.78|
> |Llama2-7B||√||9.84|13.31|
> |Llama2-7B|√|√||9.13|11.78|
> |Llama2-7B|√|√|√|**8.34**|**11.10**|
> |Qwen2.5-14B||| |17.41|25.20|
> |Qwen2.5-14B|||√|16.01|23.71|
> |Qwen2.5-14B|√|||15.42|20.71|
> |Qwen2.5-14B||√||14.08|21.48|
> |Qwen2.5-14B|√|√||12.99|18.40|
> |Qwen2.5-14B|√|√|√|**11.00**|**16.60**|
>
> **Table C**: Ablation of inter-layer sliding quantization with different choices of $L_s$ and $L_d$. Best results are bolded.
>
> |Model|$L_s$|$L_d$|Wikitext2↓|C4↓|
> |-|:-:|:-:|:-:|:-:|
> |Llama2-7B|2|2|10.23|13.24|
> |Llama2-7B|3|3|9.66|12.87|
> |Llama2-7B (default setting)|4|4|9.13|11.78|
> |Llama2-7B|5|5|8.98|11.72|
> |Llama2-7B|6|6|**8.94**|**11.67**|
> |Qwen2.5-14B|2|2|14.95|20.30|
> |Qwen2.5-14B|3|3|13.83|19.07|
> |Qwen2.5-14B (default setting)|4|4|12.99|18.40|
> |Qwen2.5-14B|5|5|12.45|18.12|
> |Qwen2.5-14B|6|6|**12.22**|**18.01**|
>
> **Table D**: Ablation of fixed-size sliding quantization with different choices of window size $s$. Best results are bolded.
>
> |Model|$s$|$L_s$&$L_d$|Wikitext2↓|C4↓|
> |-|:-:|:-:|:-:|:-:|
> |Llama2-7B|2|-|12.73|14.45|
> |Llama2-7B|3|-|11.18|13.94|
> |Llama2-7B|4|-|11.13|13.52|
> |Llama2-7B|2|4|**9.13**|**11.78**|
> |Qwen2.5-14B|2|-|17.41|25.20|
> |Qwen2.5-14B|3|-|15.94|24.15|
> |Qwen2.5-14B|4|-|14.75|21.41|
> |Qwen2.5-14B|2|4|**12.99**|**18.40**|
>
> **Table E**: Ablation of intra-layer sliding quantization with different choices of  $\gamma$. Note, the number of incremental quantization stages is determined by ($N=1/\gamma$). Best results are bolded.
>
> |Model|Ratio $\gamma$|#Stage $N$|Wikitext2↓|C4↓|
> |-|:-:|:-:|:-:|:-:|
> |Llama2-7B|1.0|1|12.73|14.45|
> |Llama2-7B (default setting)|0.5|2|**10.34**|13.46|
> |Llama2-7B|0.33|3|10.56|**13.30**|
> |Llama2-7B|0.25|4|11.32|14.10|
> |Qwen2.5-14B|1.0|1|17.41|25.20|
> |Qwen2.5-14B (default setting)|0.5|2|16.01|23.71|
> |Qwen2.5-14B|0.33|3|**15.67**|23.03|
> |Qwen2.5-14B|0.25|4|16.07|**22.71**|

---

> ### Author Response · Authors · 2025-11-24
> **Rebuttal to Official Review by Reviewer R2bM: 3/3**
>
> **4. To your first question** “Missing baseline results in Table 1: Some baselines in Table 1 have missing or abnormal values. Could the authors clarify the cause?”
>
> **Our responses**: for baseline results in Table 1, the reasons for few missing values (denoted as “-”) or an abnormal value (denoted as “NAN”) are: **(1)** for Llama2-7B, Llama2-13B, Llama2-70B, all the results of baseline methods are collected from the original papers of recent works OmniQuant, QuIP, FlatQuant and etc., so few missing values (denoted as “-”) for QuIP and AffineQuant mean that they did not report their results on the corresponding LLMs and datasets, and an abnormal value (denoted as “NAN”) is reported in the original paper of [OmniQuant](https://openreview.net/pdf?id=8Wuvhh0LYW) (see its Table A19). For a fair comparison, we always use their originally reported values; **(2)** for Llama3-8B, Qwen2.5-7B and Qwen2.5-14B which were released after 2024 and thus are not used for experiments by these existing PTQ methods. To have a fair comparison, we report the best results obtained from the experiments by us for some baseline methods whose codes are publicly available and high-quality, and few missing values (denoted as “-”) for QuIP and AffineQuant mean that we could not get meaningful results for them as their codes do not well support the quantization of these new models.
>
> **5. To your second question** “Source of improvement: sliding vs. repeated optimization… Which factor contributes more to the final improvement? An additional ablation ... would help clarify the main source of gain.”
>
> **Our responses**: thanks for your insightful question. We conducted an ablation to compare the roles of the sliding-window quantization design and the repeated optimization (quantization of $s$ layers in each window by multiple times) on Llama2-7B and Qwen2.5-14B under W4A4 quantization. To better illustrate the effect of repeated quantization, we remove intra-layer sliding quantization in our SliderQuant. Detailed results are summarized in the below Table F and Table G, from which we can see that, on both Llama2-7B and Qwen2.5-14B: **(1)** for fixed-size sliding quantization with the window size of $s=2$ or $s=4$ applied to all model layers, performing multiple times of quantization of $s$ layers in each window can improve the quantization performance, but the gain against its corresponding single-time baseline is marginal (<1 perplexity); **(2)** comparatively, retaining a fixed-size sliding window $s=2$ for intermediate layers, by introducing a progressively expanded sliding window (PESW) for $L_s=4$ shallow layers and a progressively contracted sliding window (PCSW) for $L_d=4$ deep layers, our SliderQuant with this default setting achieves significant perplexity reductions (ranging from 1.54 to 3.60 for Llama2-7B and 1.23 to 4.42 for Qwen2.5-14B on Wikitext2, and with similar reductions observed on C4) over all fixed-sized sliding quantization baselines and their repeated variants. These experimental results indicate that our adaptive sliding quantization (coupling three different sliding window designs tailored for shallow, intermediate and deep layers, respectively) contributes significantly more to the final improvement of quantization performance compared to merely applying fixed-size sliding window to all layers of FP16 LLMs (even with repeated optimization), which well echoes the importance of our empirical observations and method’s motivation.
>
> **Table F**: Ablation of comparing SliderQuant with repeated fixed-size sliding quantization on Llama2-7B under W4A4 quantization. The repeated baseline applies quantization to $s$ layers in each window multiple times.
>
> |Model|$s$|$L_s$&$L_d$|Number of repeated quantization times|Wikitext2↓|C4↓|
> |-|:-:|:-:|:-:|:-:|:-:|
> |Llama2-7B|2|-|1|12.73|14.45|
> |Llama2-7B|2|-|2|12.58|14.23|
> |Llama2-7B|2|-|3|12.41|14.15|
> |Llama2-7B|2|-|4|12.11|14.08|
> |Llama2-7B|4|-|1|11.13|13.52|
> |Llama2-7B|4|-|2|10.94|13.39|
> |Llama2-7B|4|-|3|10.79|13.25|
> |Llama2-7B|4|-|4|10.67|13.12|
> |Llama2-7B|2|4|1|**9.13**|**11.78**|
>
> **Table G**: Ablation of comparing SliderQuant with repeated fixed-size sliding quantization on Qwen2.5-14B under W4A4 quantization. The repeated baseline applies quantization to $s$ layers in each window multiple times.
>
> |Model|$s$|$L_s$&$L_d$|Number of repeated quantization times|Wikitext2↓|C4↓|
> |-|:-:|:-:|:-:|:-:|:-:|
> |Qwen2.5-14B|2|-|1|17.41|25.20|
> |Qwen2.5-14B|2|-|2|17.23|25.12|
> |Qwen2.5-14B|2|-|3|17.01|24.97|
> |Qwen2.5-14B|2|-|4|16.91|24.81|
> |Qwen2.5-14B|4|-|1|14.75|21.41|
> |Qwen2.5-14B|4|-|2|14.54|20.93|
> |Qwen2.5-14B|4|-|3|14.35|20.71|
> |Qwen2.5-14B|4|-|4|14.21|20.65|
> |Qwen2.5-14B|2|4|1|**12.99**|**18.40**|
>
> **Finally**, based on the constructive comments by you and the other three reviewers, **we carefully revised and updated the manuscript of our work**. Regarding more experiments and discussions, you are referred to our responses to the other three reviewers, and the revised manuscript.

---

### Official Review · Reviewer_ZBQd · 2025-11-01

**Soundness:** 3
**Presentation:** 3
**Contribution:** 2
**Rating:** 6
**Confidence:** 4

**Summary:**

The paper improves post-training quantization by making the sliding-window reconstruction approach depth-aware. Instead of a fixed window, they expand the window in shallow layers, keep it fixed in the middle, and contract it near the end so that the sensitive early/late layers are quantized with enough context. This simple schedule combined with incremental quantization within each window, outperforms common PTQ baselines (SmoothQuant/OmniQuant/CBQ) on several models(e.g. LLaMA/Qwen/MoE) especially at 4-bit and 2-bit. The main value is practical and better results without changing inference while at the same time the limitations are extra calibration cost and only incremental novelty over existing depth/adaptive PTQ ideas.

**Strengths:**

- Depth-aware sliding window actually makes early and late layers easier to quantize, instead of treating all layer depths the same.
- Inter-layer and intra-layer sliding reinforce each other, so you get denser cross-layer synergy as compared to a fixed window.
- On MoE (Table 4) it improves over OmniQuant at every bit setting, which helps generalizable, not tuned for one model claim.
- Generation Table 5 is especially strong, 2-bit OmniQuant nearly collapses on DeepSeek-R1 distilled models, while 2-bit SliderQuant keeps usable pass@1 on math and code. That’s very useful in order to have a useful model in the end.
- The method stays compatible with standard PTQ methods(scaling, re-quantizing overlaps), so it is easy to slot into existing pipelines.

**Weaknesses:**

- The method adds several schedule knobs (expand depth, contract depth, window size, γ), and robustness to non-ideal choices is not fully presented in the paper.
- Comparisons are mostly against fixed-window, non-rotated post training quantization techniques. It’s unclear how much of the gains remain vs the strongest rotation/equivalent methods.

**Questions:**

- Results against at least one rotation-based PTQ(e.g. SpinQuant[1]) to contextualize the gains would be very useful.
- Is there a way to automatically estimate the window schedule from measures e.g. per-layer sensitivity.
- What is the calibration-time required and memory overhead compared to the simplest fixed-window baseline?
- Do the improvements still hold with a smaller calibration set or if the calibration set is slightly domain-mismatched?

I'd be happy to increase my score if these questions are answered!

[1] Liu, Z., Zhao, C., Fedorov, I., Soran, B., Choudhary, D., Krishnamoorthi, R., Chandra, V., Yuandong, T., & Blankevoort, T. (2025). SpinQuant: LLM quantization with learned rotations. arXiv:2405.16406.

---

> ### Author Response · Authors · 2025-11-24
> **Rebuttal to Official Review by Reviewer ZBQd: 1/4**
>
> Thanks for your constructive review and the recognition of the motivation, the core novelties, the performance and the advantages of our method. Next, we address your concerns and questions one by one.
>
> **1. To your comments about the first weakness** “The method adds several schedule knobs (expand depth, contract depth, window size, $\gamma$), and robustness to non-ideal choices is not fully presented in the paper.”
>
> **Our responses**: as you said, given an FP16 LLM having $L$ layers, our SliderQuant has four schedule knobs including a progressively expanded sliding window (PESW) for $L_s$ shallow layers, a fixed-size sliding window with the size of $s$ for $L_i$ intermediate layers and a progressively contracted sliding window (PCSW) for $L_d$ deep layers used in inter-layer sliding quantization (Inter-S), and an incremental quantization ratio $\gamma$ for intra-layer sliding quantization (Intra-S) within each window of Inter-S. In the Subsection 4.2 (Table 6, Table 7, Table 8 and Table 9) of our original paper submission, we used Llama2-7B to study their choices from the perspectives of their individual roles and combined roles, and chose $L_s=4, s=2, L_d=4, \gamma=0.5$ as the default setting of our SliderQuant applied to all 9 different LLMs tested in our paper, for a simple implementation. Indeed, this default setting is not optimal even for Llama2-7B, let alone other LLMs. To better compare their choices, we additionally conducted multiple sets of ablative experiments with larger and recently released Qwen2.5-14B under W4A4 quantization. Detailed results are summarized in the below Table A, Table B, Table C and Table D, from which we can see that, in our SliderQuant: **(1)** with our default setting, four schedule knobs are complementary to each other (see Table A); **(2)** for each individual knob, its default setting is usually not the best for both Llama2-7B and Qwen2.5-14B (see Table B and Table D); **(3)** combining it with PESW and PCSW is significantly better than merely using fixed-size sliding quantization to all model layers, both on Llama2-7B and Qwen2.5-14B (see Table C). Although our SliderQuant with the default setting already achieves superior results to existing PTQ methods, these ablations indicate that there is still room to get improved quantization performance by choosing better settings of these four schedule knobs for different LLMs.
>
>  **Table A**: Ablation of combining different schedule knobs in inter-layer sliding quantization (Inter-S) and intra-layer sliding quantization (Intra-S). In the experiments, we use the default setting of $L_s=4, s=2, L_d=4, \gamma=0.5$. Best results are bolded.
>
> |Model|PESW|PCSW|Intra-S|Wikitext2↓|C4↓|
> |-|:-:|:-:|:-:|:-:|:-:|
> |Llama2-7B||| |12.73|14.45|
> |Llama2-7B|||√|10.34|13.46|
> |Llama2-7B|√|||10.30|13.78|
> |Llama2-7B||√||9.84|13.31|
> |Llama2-7B|√|√||9.13|11.78|
> |Llama2-7B|√|√|√|**8.34**|**11.10**|
> |Qwen2.5-14B||| |17.41|25.20|
> |Qwen2.5-14B|||√|16.01|23.71|
> |Qwen2.5-14B|√|||15.42|20.71|
> |Qwen2.5-14B||√||14.08|21.48|
> |Qwen2.5-14B|√|√||12.99|18.40|
> |Qwen2.5-14B|√|√|√|**11.00**|**16.60**|
>
> **Table B**: Ablation of inter-layer sliding quantization with different choices of $L_s$ and $L_d$. Best results are bolded.
>
> |Model|$L_s$|$L_d$|Wikitext2↓|C4↓|
> |-|:-:|:-:|:-:|:-:|
> |Llama2-7B|2|2|10.23|13.24|
> |Llama2-7B|3|3|9.66|12.87|
> |Llama2-7B (default setting)|4|4|9.13|11.78|
> |Llama2-7B|5|5|8.98|11.72|
> |Llama2-7B|6|6|**8.94**|**11.67**|
> |Qwen2.5-14B|2|2|14.95|20.30|
> |Qwen2.5-14B|3|3|13.83|19.07|
> |Qwen2.5-14B (default setting)|4|4|12.99|18.40|
> |Qwen2.5-14B|5|5|12.45|18.12|
> |Qwen2.5-14B|6|6|**12.22**|**18.01**|
>
> **Table C**: Ablation of merely using fixed-size sliding quantization with different choices of window size $s$. Best results are bolded.
>
> |Model|$s$|$L_s$&$L_d$|Wikitext2↓|C4↓|
> |-|:-:|:-:|:-:|:-:|
> |Llama2-7B|2|-|12.73|14.45|
> |Llama2-7B|3|-|11.18|13.94|
> |Llama2-7B|4|-|11.13|13.52|
> |Llama2-7B|2|4|**9.13**|**11.78**|
> |Qwen2.5-14B|2|-|17.41|25.20|
> |Qwen2.5-14B|3|-|15.94|24.15|
> |Qwen2.5-14B|4|-|14.75|21.41|
> |Qwen2.5-14B|2|4|**12.99**|**18.40**|
>
> **Table D**: Ablation of intra-layer sliding quantization with different choices of $\gamma$. Note, the number of incremental quantization stages is determined by $N=1/\gamma$. Best results are bolded.
>
> |Model|Ratio $\gamma$|#Stage $N$|Wikitext2↓|C4↓|
> |-|:-:|:-:|:-:|:-:|
> |Llama2-7B|1.0|1|12.73|14.45|
> |Llama2-7B (default setting)|0.5|2|**10.34**|13.46|
> |Llama2-7B|0.33|3|10.56|**13.30**|
> |Llama2-7B|0.25|4|11.32|14.10|
> |Qwen2.5-14B|1.0|1|17.41|25.20|
> |Qwen2.5-14B (default setting)|0.5|2|16.01|23.71|
> |Qwen2.5-14B|0.33|3|**15.67**|23.03|
> |Qwen2.5-14B|0.25|4|16.07|**22.71**|

---

> ### Author Response · Authors · 2025-11-24
> **Rebuttal to Official Review by Reviewer ZBQd: 2/4**
>
> **2. To your comments about the second weakness and the first question** “Comparisons are mostly against fixed-window, non-rotated post training quantization techniques. It’s unclear how much of the gains remain vs the strongest rotation/equivalent methods.” and “Results against at least one rotation-based PTQ(e.g. SpinQuant[1]) to contextualize the gains would be very useful.”
>
> **Our responses**: **(1)** actually, in the Subsection 4.1 of our original paper submission, Table 3 provides a comparison of our SliderQuant against four rotation-based methods DuQuant (Lin et al., NeurIPS 2024), QuaRot (Ashkboos et al., NeurIPS 2024), your mentioned SpinQuant (Liu et al., ICLR 2025) and the latest FlatQuant (Sun et al., ICML 2025), and other state-of-the-art PTQ methods for weight-activation quantization, which all modify network architectures of LLMs (i.e., we call them the PTQ method group “with extra inference-time costs”) during the quantization process. Table 1 and Table 2 provide comparisons of our SliderQuant against state-of-the-art PTQ methods without modifying network architectures of LLMs (i.e., we call them the PTQ method group “without extra inference-time costs”). For fair comparisons, we split these main experiments into two parts; **(2)** for a more comprehensive comparison of our SliderQuant and these rotation-based methods, we further extended our experiments to cover a wider range of model size and structure, and compare our results with the reported results in the original papers of DuQuant, QuaRot, SpinQuant and FlatQuant. Detailed results are summarized in the below Table E, from which we can see that our SliderQuant outperforms these four state-of-the-art rotation-based methods on different LLMs including Llama2-7B, Llama2-13B, Llama2-70B, Llama3-8B and Qwen2.5-7B-Instruct.
>
>  **Table E**: Results comparison of our method and different rotation-based methods with extra inference-time costs (it is known that rotation matrices added to some layers during the quantization process cannot be absorbed into the LLMs, due to non-linear operations). SliderQuant+ denotes our method using rotation transformations. Best results are bolded.
>
> |Model|#Bits|Method|Wikitext2↓|C4↓|PIQA↑|ARC-e↑|Arc-c↑|HellaSwag↑|Winogrande↑|Avg↑|
> |-|:-:|:-:|:-:|:-:|:-:|:-:|:-:|:-:|:-:|:-:|
> |Llama2-7B|W16A16|-|5.47|6.97|78.84|74.62|46.42|75.9|69.46|69.05|
> |Llama2-7B|W4A4|DuQuant|6.08|7.79|75.68|50.00|37.46|69.74|63.93|59.36|
> |Llama2-7B|W4A4|QuaRot|6.10|8.69|76.77|69.87|40.87|72.16|63.77|64.69|
> |Llama2-7B|W4A4|SpinQuant|5.96|8.28|76.17|69.28|41.72|72.90|66.06|65.23|
> |Llama2-7B|W4A4|FlatQuant|5.79|7.79|77.26|72.05|43.26|73.64|69.53|67.15|
> |Llama2-7B|W4A4|SliderQuant+|**5.71**|**7.68**|**77.97**|**73.15**|**43.35**|**73.71**|**69.74**|**67.58**|
> |Llama2-13B|W16A16|-|4.88|5.46|80.41|77.40|49.15|79.37|72.14|71.69|
> |Llama2-13B|W4A4|DuQuant|5.33|7.02|77.26|56.23|42.15|73.68|65.43|62.95|
> |Llama2-13B|W4A4|QuaRot|6.10|8.67|77.69|69.95|42.83|73.54|67.88|66.38|
> |Llama2-13B|W4A4|Spinquant|5.44|8.11|78.40|72.43|43.69|75.52|68.90|67.79|
> |Llama2-13B|W4A4|Flatquant|5.11|7.11|79.65|76.94|48.38|77.88|70.56|70.68|
> |Llama2-13b|W4A4|SliderQuant+|**5.07**|**7.04**|**79.96**|**77.27**|**48.95**|**77.96**|**71.98**|**71.22**|
> |Llama2-70B|W16A16|-|3.32|5.71|82.70|81.02|57.17|83.81|77.98|76.54|
> |Llama2-70B|W4A4|QuaRot|3.79|6.12|81.83|79.76|55.46|81.58|76.09|74.94|
> |Llama2-70B|W4A4|Spinquant|3.70|6.07|82.37|79.04|55.38|82.57|78.22|75.52|
> |Llama2-70B|W4A4|Flatquant|3.55|5.91|82.75|80.30|56.14|83.01|77.90|76.02|
> |Llama2-70B|W4A4|SliderQuant+|**3.50**|**5.87**|**82.75**|**81.23**|**56.57**|**83.12**|**77.93**|**76.32**|
> |Llama3-8B|W16A16|-|6.13|8.93|80.79|77.69|53.41|79.13|72.77|72.76|
> |Llama3-8B|W4A4|DuQuant|8.06|11.29|76.22|70.41|43.69|73.87|67.80|66.40|
> |Llama3-8B|W4A4|QuaRot|8.16|13.38|75.14|68.01|43.34|72.94|65.82|65.05|
> |Llama3-8B|W4A4|SpinQuant|7.39|12.19|77.37|74.20|47.27|74.55|68.51|68.38|
> |Llama3-8B|W4A4|FlatQuant|6.98|11.13|79.16|75.80|50.00|76.80|72.69|70.89|
> |Llama-3-8B|W4A4|SliderQuant+|**6.87**|**11.04**|**79.22**|**77.53**|**50.60**|**77.31**|**72.82**|**71.50**|
> |Qwen2.5-7B-Instruct|W16A16|-|8.36|14.37|80.20|75.80|51.37|79.57|69.93|71.37|
> |Qwen2.5-7B-Instruct|W4A4|FlatQuant|8.46|13.94|76.93|77.69|51.71|78.42|69.53|70.86|
> |Qwen2.5-7B-Instruct|W4A4|SliderQuant+|**8.00**|**13.38**|**79.56**|**79.05**|**52.27**|**78.66**|**69.88**|**71.88**|

---

> ### Author Response · Authors · 2025-11-24
> **Rebuttal to Official Review by Reviewer ZBQd: 3/4**
>
> **3. To your second question** “Is there a way to automatically estimate the window schedule from measures e.g. per-layer sensitivity.”
>
> **Our responses**: we sincerely thank you for this truly insightful question which points out a future direction for extending the sliding quantization mechanism of our SliderQuant. Currently, we are thinking on two potential ways to automatically estimate the window schedule from measuring per-layer sensitivity in terms of quantization error. One way is to formulate a dynamic programming strategy, and the other way is to formulate a reinforcement learning strategy. To explore these two potential ways, given an FP16 LLM, how to define a proper set of candidate windows (spanning varying expanded depth $L_s$, fixed-size sliding window size $s$, contracted depth $L_d$, and intra-layer sliding ratio $\gamma$), how to define a robust layer-sensitivity-aware quantization loss, and how to initialize, sample and update the window schedule in a learnable manner based on the methodology of either dynamic programming or reinforcement learning are three key research problems, which cannot be easily addressed in a short term. We would like to explore them in our future research.
>
> **4. To your third question** “What is the calibration-time required and memory overhead compared to the simplest fixed-window baseline?”
>
> **Our responses**: in the below Table F, we provide an ablation to compare the calibration-time and the memory overhead of our SliderQuant and the simplest fixed-size sliding quantization baseline (applied to all model layers), under the same number of calibration samples (128), the same batch size (1) and the same number of training epochs (20). We can see that, on both Llama2-7B and Qwen2.5-14B under W4A4 quantization: **(1)** compared to the fixed-sized sliding quantization with the window size of $s=4$, our SliderQuant with the default setting of $L_s=4, s=2, L_d=4, \gamma=0.5$ needs less calibration time and the same memory overhead, and achieves significantly better quantization results; **(2)** the fixed-sized sliding quantization with the window size of $s=2$ is more efficient in terms of the calibration time and the memory overhead compared to the fixed-sized sliding quantization with the window size of $s=4$ and our SliderQuant with the default setting, but shows much worse quantization results; **(3)** SliderQuant-Fast (the only change over SliderQuant is the number of calibration epochs is 10 instead of 20) needs the least calibration time and achieves slightly worse quantization results compared to SliderQuant, but its memory overhead keeps the same to SliderQuant. Summarily, these results validate the training efficiency of SliderQuant, which guarantees the practicality of our method to quantize different LLMs, also thanks to the simplicity of our method.
>
> **Table F**: A comparison of the calibration-time and the memory overhead of our SliderQuant and the simplest fixed-size sliding quantization baseline on Llama2-7B and Qwen2.5-14B under W4A4 quantization. All experiments are conducted on a single NVIDIA A6000-48GB GPU, and we set the number of calibration samples to 128 (a popular choice in the quantization community), the batch size to 1, and the number of training epochs to 20 for all methods except our SliderQuant-Fast with 10 training epochs. Best results are bolded.
>
> |Model|Method|Wikitext2↓|C4↓|Training Time (GPU Hours)|Memory Overhead (GB)|
> |-|-|:-:|:-:|:-:|:-:|
> |Llama2-7B|Fixed-size sliding (s=2) |12.73|14.45|5.66|**16.24**|
> |Llama2-7B|Fixed-size sliding (s=3) |11.18|13.94|7.65|23.02 |
> |Llama2-7B|Fixed-size sliding (s=4) |11.13|13.52|9.16|29.80|
> |Llama2-7B|SliderQuant-Fast|8.92|11.59|**3.24**|29.80|
> |Llama2-7B|SliderQuant|**8.34**|**11.10**|6.14|29.80|
> |Qwen2.5-14B|Fixed-size sliding (s=2) |17.41|25.20|10.83|**21.12**|
> |Qwen2.5-14B|Fixed-size sliding (s=3) |15.94|24.15|13.56 | 30.23 |
> |Qwen2.5-14B|Fixed-size sliding (s=4) |14.75|21.41|15.36|43.71|
> |Qwen2.5-14B|SliderQuant-Fast|12.19|17.53|**6.08**|43.71|
> |Qwen2.5-14B|SliderQuant|**11.00**|**16.60**|11.43|43.71|

---

> ### Author Response · Authors · 2025-11-24
> **Rebuttal to Official Review by Reviewer ZBQd: 4/4**
>
> **5. To your fourth question** “Do the improvements still hold with a smaller calibration set or if the calibration set is slightly domain-mismatched?”
>
> **Our responses**: the answer is Yes. In the below Table G and Table H, we provide an ablation to study the performance of our SliderQuant under a smaller number of calibration samples {32, 64} and a larger number of calibration samples 256, besides the default number of calibration samples 128, and we compare our results with the reported results in the original papers of OmniQuant and DuQuant (a top rotation-based PTQ method). We can see that: **(1)** when using a smaller or larger number of calibration samples, our SliderQuant and SliderQuant+ (using rotation transformations) achieve significantly better results than OmniQuant and also better results than DuQuant (using learnable rotation transformations) consistently; **(2)** intriguingly, even with 32 calibration samples, our SliderQuant and SliderQuant+ show superior performance than OmniQuant and DuQuant with 256 calibration samples, respectively.
>
> **Table G**: A comparison of SliderQuant and OmniQuant with different numbers of calibration samples under W4A4 quantization. Best results are bolded.
>
> |Model|Nsample|Method|Wikitext2↓|C4↓|
> |-|:-:|:-:|:-:|:-:|
> |Llama-7B|32|OmniQuant|11.48|14.8|
> |Llama-7B|32|SliderQuant|8.71|11.19|
> |Llama-7B|64|OmniQuant|11.40|14.57|
> |Llama-7B|64|SliderQuant|8.30|11.13|
> |Llama-7B|128|OmniQuant|11.23|14.61|
> |Llama-7B|128 (default setting)|SliderQuant|**8.01**|**10.58**|
> |Llama-7B|256|OmniQuant|11.41|14.90|
> |Llama-7B|256|SliderQuant|8.49|11.26|
>
> **Table H**: A comparison of SliderQuant and DuQuant with different numbers of calibration samples under W4A4 Quantization. Best results are bolded.
>
> |Model|Nsample|Method|Wikitext2↓|C4↓|
> |-|:-:|:-:|:-:|:-:|
> |Llama2-7B|32|DuQuant|6.31|7.99|
> |Llama2-7B|32|SliderQuant+|5.83|7.85|
> |Llama2-7B|64|DuQuant|6.29|7.88|
> |Llama2-7B|64|SliderQuant+|5.81|7.83|
> |Llama2-7B|128|DuQuant|6.28|7.90|
> |Llama2-7B|128 (default setting)|SliderQuant+|5.71|7.68|
> |Llama2-7B|256|DuQuant|6.23|7.88|
> |Llama2-7B|256|SliderQuant+|**5.64**|**7.53**|
>
> **Finally**, based on the constructive comments by you and the other three reviewers and our responses, **we carefully revised and updated the manuscript of our work**. Regarding more experiments and discussions that we made during the rebuttal phase, you are referred to our responses to the other three reviewers, and the revised manuscript.

---

> ### Comment · Reviewer_ZBQd · 2025-11-26
> **Reviewer Comment**
>
> Thank you for your answers, I've revised my score accordingly.

---

> ### Author Response · Authors · 2025-11-27
> **Thank You for the Recognition of Our Rebuttal and Work**
>
> Dear Reviewer ZBQd,
>
> We are glad to see that you are satisfied with our rebuttal and have increased your score to 8. We will continue to improve the quality of our work.
>
> Thanks again for your constructive comments, time and patience.
>
> The authors.

---

### Author Response · Authors · 2025-11-21
**Our rebuttal will be submitted on next Monday (November 24)**

Dear Reviewers, Area Chairs, Senior Area Chairs and Program Chairs,

We are trying our best to thoroughly conduct extra experiments and in-depth analysis requested by four reviewers. At the moment, it seems that we can finish all these experiments in two to three days, and can finalize and submit our rebuttal on next Monday (November 24).

Thanks for your constructive comments and great patience!

The authors.

---

### Author Response · Authors · 2025-11-30
**Summary of Our Rebuttal and the Author-Reviewer Discussion**

Dear ACs, SACs and PCs,

We are really shocked by your notification that an OpenReview bug leaked the names of authors, reviewers and area chairs for all conferences, posing a serious risk of misconduct on leaked confidential information. Before this notification, our paper already got scores of (8,6,6,6) even though Reviewer R2bM did not reply to our rebuttal yet. Although we are honestly frustrated by your decision “reverting reviews and scores to their state before the discussion period”, we understand that this incident deeply impacts the ICLR community and support your decisions. Here, we make a summary of our paper status.

**1. Declarations**: **1)** we have strictly followed the ICLR Code of Conduct from our first paper in ICLR 2017 to now; **2)** we did not know this OpenReview bug before your notification, and would never think about any violation of the Code of Conduct for all conferences, being an esteemed team in the field.

**2. Scores of our paper**: **before your notification, our paper already got (8,6,6,6) scores** as 3 Reviewers (ZBQd, R2bM, BEoY) replied that all their concerns had been well resolved by our detailed responses and increased their scores consistently, and Reviewer DdkN did not reply to our rebuttal yet. **We believe our thorough rebuttal also well address all concerns of Reviewer DdkN who is likely to increase his/her initial positive rating 6 further**.

**3. Initial reviews and rebuttal feedbacks: 1) before the discussion period**, the empirical observations/motivation, the method’s novelty, the good writing and the comprehensive experiments of our work had been well recognized by all 4 Reviewers except that Reviewer BEoY has two doubts about i) the novelty of our empirical observations and method; ii) the practical significance of some experiments. **Actually, these two doubts of Reviewer BEoY do not hold at all, mainly due to several facts regarding our original paper submission**: few misunderstandings of our work and its connections & differences to some related works and our organization of experiments aimed for more fair comparisons with existing PTQ methods, missed critical clarifications and discussions; **2) during the discussion period**, we politely clarified these facts at a very detailed level, and **Reviewer BEoY recognized our responses and increased his/her rating to 6**.

**4. Improvements in our rebuttal and revised paper**: **1)** 4 sets of ablative experiments to more sufficiently study the choices of schedule knobs/hyperparameters of our method (concerned by ZBQd, R2bM); **2)** a set of experiments to compare the training time & memory overhead of our method with fixed-window baseline CBQ (concerned by ZBQd, R2bM, DdkN) and OmniQuant (concerned by DdkN), validating the training efficiency of our method; **3)** a set of ablative experiments with 32/64/128(default)/256 calibration samples (including a smaller number concerned by ZBQd), validating consistent gains of our method to both non-rotation-based and rotation-based PTQ methods; **4)** a set of ablative experiments to compare our adaptive sliding with repeated optimization (R2bM's concern), validating the gain is mainly from adaptive sliding quantization; **5)** a discussion of our two potential ways to automatically estimate the window schedule (ZBQd's concern); **6)** a more detailed description of intra-layer sliding quantization (DdkN's concern); **7)** clarifications about few missing or abnormal values “- /NAN” for QuIP and AffineQuant in Table 1, which are based on their related papers (DdkN's concern); **8)** from a perspective of local and global layer roles of LLMs, we provide intuitive explanations for our empirical observation that the first and last layers are much more sensitive to quantization than the other layers (BEoY's concern); **9)** 2 sets of experiments to show superior performance of our method against mixed-precision PTQ methods, for weight-only (BEoY's concern) and weight-activation quantization; **10)** a limitation discussion of our method compared to layer-wise PTQ methods (suggested by DdkN) **11) some aspects missed by Reviewers**: our method vs. rotation-based PTQ methods (ZBQd's concern), even quantization frequency of middle layers (R2bM's concern), training details (loss function and supervision information) and the role of two low-rank matrices $A$ and $B$ (DdkN's concern), differences of our method to mixed-precision, fixed-sized sliding and layer-wise PTQ methods (BEoY's concern), **which were actually included in our original paper submission**. We also add more clarifications and experiments to further improve the Method and Experiments sections of our paper.

Considering the above aspects, we believe our paper well surpasses the acceptance bar of ICLR 2026. We look forward to your positive decision.

Finally, we sincerely thank 4 reviewers for their thorough reviews and your relentless efforts to ensure the success of ICLR 2026.

Thanks,

The authors.

---

### Meta-Review · Area_Chair_ttnK · 2025-12-16

**Summary:**

This paper receives mixed initial scores, and the authors provide detailed rebuttals to each question, and three reviewers timely respond to the rebuttal and increase the scores accordingly. The AC carefully checked the rebuttal process and recommended accepting this paper. The authors are highly encouraged to include these modifications for the final version.

**Reviewer Concerns:**

1. Reviewer ZBQd (concerns addressed)
2. Reviewer R2bM would keep the score at 6.
3. Reviewer DdkN (concern addressed) raised the score from 4 to 6.
4. Reviewer BEoY (most concerns addressed) raised the score from 4 to 6.

**Reviewer Scores:**

1. Reviewer ZBQd raised the score from 6 to 8.
2. Reviewer R2bM would keep the score 6.
3. Reviewer DdkN raised the score from 4 to 6.
4. Reviewer BEoY raised the score from 4 to 6.

---

### Decision · Program_Chairs · 2026-01-26

Accept (Poster)